# Temperature and acidity dependence of secondary organic aerosol formation from α-pinene ozonolysis with a compact chamber system

Yange Deng[1], Satoshi Inomata[1]*, Kei Sato[1], Sathiyamurthi Ramasamy[1], Yu Morino[1], Shinichi Enami[1], Hiroshi Tanimoto[1]

[1]National Institute for Environmental Studies, Tsukuba 305-8506, Japan

*Correspondence to*: Satoshi Inomata (ino@nies.go.jp)

**Abstract.** Secondary organic aerosols (SOAs) affect human health and climate change prediction; however, the factors (e.g., temperature, acidity of pre-existing particles, and oxidants) influencing their formation are not sufficiently resolved. Using a compact chamber, the temperature and acidity dependence of SOA yields and chemical components in SOA from α-pinene ozonolysis were systematically investigated under 278 K, 288 K, and 298 K temperatures using neutral ($(NH_4)_2SO_4$)/acidic ($H_2SO_4+((NH_4)_2SO_4)$) seed aerosols. SOA components with *m/z* less than 400 were analyzed using negative electrospray ionization liquid-chromatography time-of-flight mass spectrometry. Based on the slightly negative temperature dependence of the SOA yields, the enthalpies of vaporization under neutral and acidic seed conditions were estimated to be 25 and 44 kJ mol$^{-1}$, respectively. In addition, SOA yields increased with an increase in the acidity of seed particles (solid/near solid state) at low SOA mass loadings, when compared with the seed particle amounts. Acidity dependence analysis of the chemical formula, molecular mass, and O:C ratio of the detected compounds indicated the enhanced formation of multiple oligomers in the wide molecular mass range with a wide range of O:C ratios under acidic seed conditions. The abundance of some chemical compounds increased with an increase in the acidity of seed particles (e.g., m/z 197, 311, 313, 339, 355 and 383), while decreases in the abundance of some chemical compounds were observed (e.g., m/z 171, 185, 215, 343, and 357). The acidity dependence could be explained by acid-catalyzed heterogeneous reactions or acid-catalyzed decomposition of hydroperoxides. In addition, organosulfate (OS) formation was observed under acidic seed conditions. Six out of the eleven detected OS were potentially formed via the aldehyde + $HSO_4^-$ pathway.

## 1 Introduction

Secondary organic aerosol (SOA) in the atmosphere is a complex of organic compounds which are formed through oxidation of precursor volatile organic compounds (VOCs) of either biogenic (e.g., monoterpene and isoprene) or anthropogenic (e.g., alkanes and aromatics) origins, or both (Hallquist et al., 2009). SOA plays important roles in the aerosol effect on climate (Tilmes et al., 2019), air quality (Parrish et al., 2011), as well as human health (Shiraiwa et al., 2017). Nevertheless, it is noted in the IPCC Fifth Assessment Report (AR5) that the formation of SOA has not been included in the estimation of the radiative forcing from aerosols because the formation is influenced by a variety of factors not yet sufficiently quantified (Stocker et al., 2013). However, in association with the advance of research technologies, processes that influence the growth of SOA particles to sizes relevant for clouds and radiative forcing have been intensively investigated (Shrivastava et al., 2017).

The importance of the formation of low (and/or extremely low) volatility organic compounds (LVOCs) with saturation concentrations less than $10^{-0.5}$ μg m$^{-3}$ through heterogeneous/multiphase accretion processes has been highlighted in SOA formation mechanisms in recent studies (Ziemann and Atkinson, 2012; Shrivastava et al., 2017). Semi-volatile organic compounds (SVOCs) are generated from oxidation reactions of VOCs in the gas phase. Heterogenous/multiphase reactions of

SVOCs on particles are thought to contribute to the formation of LVOCs in SOA. Earlier acid-catalyzed heterogeneous reaction studies (Jang et al., 2002; Hallquist et al., 2009; Ziemann and Atkinson, 2012) proposed the formation of hemiacetals, aldol

products, organosulfates, among others, in the presence of acidic seed particles, which prompted the notion that the acidity of pre-existing particles is one of the key factors that influence SOA formation.

The influence of the acidity of pre-existing particles on SOA formation has been investigated in both chamber experiments and field measurements. From chamber experiments, Eddingsaas et al. (2012) observed the clear uptake of several SVOCs (e.g., α-pinene oxide and α-pinene hydroxy hydroperoxides) after the injection of acidic particles into α-pinene OH oxidation

system under low-NOx conditions (photooxidation for four hours, lights off and contents in the dark for two hours followed by the injection). However, no apparent uptake was observed after the injection of neutral particles. Shiraiwa et al. (2013a) observed the evident formation of peroxyhemiacetals after gaseous tridecanal was injected into the dodecane photooxidation system under dry, ammonium sulfate seed particles, and low-NOx conditions (photooxidation for four hours, lights off and contents in the dark for two hours followed by the injection), which was believed to have been catalyzed by the presence of

acids generated in the low-NOx dodecane mechanism. However, the influence of the acidity of pre-existing particles on SOA yields from chamber experiments is poorly understood. Previous studies have reported complex results. For example, Eddingsaas et al. (2012) reported greater SOA yields under acidic than under neutral seed conditions from photooxidation of α-pinene under high-NOx conditions, and no influence of seed particle acidity on SOA yields under low-NOx conditions. Field studies also reported inconsistent results on the influence of acidity on SOA formation. Some researchers reported that SOA

formation was enhanced under more acidic conditions (Chu et al., 2004; Lewandowski et al., 2007; Zhang et al., 2007; Hinkley et al., 2008; Renjarajan et al., 2011; Zhou et al., 2012), whereas others reported little or no enhancement under acidic conditions (Takahama et al., 2006; Peltier et al., 2007; Tanner et al., 2009). Other factors, such as temperature, humidity, $NO_x$ concentration level, and oxidation agents, might have also affected the results of the aforementioned studies in addition to the acidity of pre-existing particles (Jang et al., 2008), but they also are not well understood. For example, the relative importance

of the temperature-dependencies of the volatilities of oxidation products and of the gas-phase and multiphase chemical reactions, which could result in different temperature-dependence of SOA yields (Pathak et al., 2007b; von Hessberg et al., 2009), are still not well constrained. This motivated the current study to develop a new compact chamber system, in which SOA formation reactions under controlled temperature, humidity, oxidation agents, and seed particle acidity can be easily performed.

Monoterpenes are known to be a large source of SOA in the global atmosphere (Kelly et al., 2018). α-Pinene is the dominant monoterpene and the second-most emitted VOC following isoprene (Guenther et al., 2012; Messina et al., 2016). It can react rapidly with atmospheric oxidants including $O_3$, OH, and $NO_3$ radicals, and ozonolysis is the major atmospheric oxidation pathway, which is estimated to account for 46 % of reacted α-pinene (Capouet et al., 2008). Pre-existing studies regarding α-pinene ozonolysis indicate that both SOA yields and chemical compositions in SOA are influenced by the air temperature and

aerosol acidity (Czoschke et al., 2003; Gao et al., 2004; Iinuma et al., 2004, 2005; Czoschke and Jang, 2006; Jang et al., 2006; Northcross and Jang, 2007; Surratt et al., 2007, 2008; Hallquist et al., 2009; Saathoff et al., 2009; Kristensen et al., 2014, 2017). Pathak et al. (2007b) found that the SOA yields show a weak temperature dependence in the range of 15 to 40 °C and a stronger temperature dependence between 0 and 15 °C. Saathoff et al. (2009) parameterized the temperature dependence of the two-product SOA yield parameters using a dataset including several previous studies as well as their own comprehensive

measurements from 243 to 313 K. In general, a negative temperature dependence of the α-pinene ozonolysis SOA yields could be confirmed. While enhancements of SOA yields from α-pinene ozonolysis reactions under acidic seed aerosol conditions compared with neutral seed conditions have been generally observed in previous studies (Czoschke et al., 2003; Gao et al., 2004; Iinuma et al., 2004, 2005; Czoschke and Jang, 2006; Jang et al., 2006; Northcross and Jang, 2007), the degree of enhancement varied probably because of the different experimental settings among the studies. For example, the studies of

Czoschke et al. (2003), Iinuma et al. (2004), Czoschke and Jang (2006), Jang et al. (2006), and Northcross and Jang (2007) reported enhancement ranges of 21–87 % from weak acid seed conditions to high acid seed conditions. Interestingly, studies of Gao et al. (2004) reported smaller enhancement (8–15 %) of SOA yields when the initial α-pinene concentrations were high, and of Iinuma et al. (2005) reported both increases and decreases of SOA yields under acidic seed conditions. It is turned out that OH radical scavengers, which is known to play an important role in influencing SOA yields (Iinuma et al., 2005; Na et

al., 2007), were applied in the studies of Gao et al. (2004) and Iinuma et al. (2005), but not applied in the studies of others.

With respect to chemical compositions, Kristensen et al. (2017) compared the chemical compositions of α-pinene ozonolysis SOA formed at temperatures of 293 and 258 K and found that the mass fraction of carboxylic acids increased at 258 K compared to 293 K, while the formation of dimer esters was suppressed at the sub-zero reaction temperature. Compared with neutral seed conditions, enhanced formation of large molecules under acidic seed conditions has been reported by Gao et al.

(2004) and Iinuma et al. (2004). Gao et al. (2004) additionally reported less abundant small oligomers (e.g., a compound with molecular mass of 358) under acidic seed conditions. These researches indicate that systematic studies of α-pinene ozonolysis SOA formation under specific experimental settings are warranted to clarify temperature and acidity dependence.

Furthermore, enhanced formation of organosulfates (OS) from α-pinene oxidation under acidic seed conditions has been suggested by previous studies (Surratt et al., 2007, 2008; Iinuma et al., 2009; Duporté et al., 2020). OS has been regarded as

an important aerosol component, accounting for up to 30 % of organic mass in $PM_{10}$, and also as an anthropogenic pollution marker in the past two decades (Iinuma et al., 2007, 2009; Surratt et al., 2007, 2008; Riva et al., 2015, 2016; Duporté et al., 2016, 2020; Brüggemann et al., 2020). The formation mechanisms of OS from α-pinene oxidations have been studied under different experimental settings. For example, Surratt et al. (2007) proposed OS formation through esterification of hydroxyl or carbonyl groups in photooxidation experiments. Surratt et al. (2008) proposed nitrooxy organosulfates formation through

esterification of hydroxyl groups in nighttime oxidation (i.e., $NO_3$-initiated oxidation under dark conditions). Iinuma et al. (2009) proposed the acid-catalyzed ring-opening of epoxides mechanism through the α-pinene oxide/acidic sulfate particle experiment. Nozière et al. (2010) proposed a sulfate radical initiated OS formation process of α-pinene in irradiated sulfate solutions. For α-pinene ozonolysis experiments, the formation of OS through reactions between $SO_2$ and stabilized Criegee intermediates under dry conditions or organic peroxides in aqueous phase has been recently suggested (Ye et al., 2018; Stangl

et al., 2019; Wang et al., 2019). However, based on our knowledge, no study concerning OS formation from α-pinene ozonolysis in the presence of sulfate particles exists.

In this study, α-pinene ozonolysis experiments have been conducted under dark conditions in the presence of seed particles and an OH scavenger utilizing a self-made compact type chamber system. The study aimed to characterize the newly developed chamber as well as to study the temperature and acidity dependence of the yield and chemical composition of α-pinene

ozonolysis SOA. Moreover, OS compounds and its possible formation mechanisms have been targeted during the analysis.

## 2 Experimental

### 2.1 Chamber description and operation

A temperature-controllable chamber system has been developed for the simulation of SOA formation (Fig. S1). The chamber is a cuboid shape Teflon bag (FEP; 0.7 m$^3$ volume, 900 mm × 600 mm × 1300 mm; 50 μm thickness; Takesue, Japan) contained

in a constant temperature cabinet (HCLP-1240; W1200 mm × D703 mm × H1466 mm; NK System, Japan). The temperature inside the cabinet was measured using a thermocouple attached to the inside of the cabinet (T3 in Fig. S1). The achievable operating temperature range of the chamber was 5–40 ℃. A detailed evaluation of the thermostat capacity of the chamber under dark conditions is presented in Text S1, which indicates that the temperature inside the chamber was well controlled (varied within ±1 ℃). The chamber is collapsible and is operated at atmosphere pressure. For a typical experimental run, a

total volume of 0.6 standard cubic meters (sm$^3$) G3 pure air (CO$_2$ < 1 ppmv, CO < 1 ppmv, THC < 1 ppmv, and dew temperature < −70 °C) was introduced into the Teflon chamber at a flow rate of 20 standard L min$^{-1}$ for 30 min. The relative humidity (RH) of the chamber air was adjusted by passing the G3 pure air through MiliQ water (resistivity of 18.2 MΩ·cm, total organic carbon content ≤ 5 ppb) before it entered the Teflon bag. The RH of the chamber air was measured after the experimental run by pumping the remaining chamber air into a separate small TEFLON bag, to which a VAISALA RH&T probe (model,

HMP76) equipped with a measurement indicator (model, M170) was attached. Particle number concentration and VOC mixing ratio measurements indicate that the chamber background concentrations of particles and VOCs were negligible, and no extra contamination was observed by the humidification process of the G3 air (Text S2). α-Pinene liquid (Wako Chemicals, Japan) was injected into the G3 pure air line through a septum equipped in a Swagelock PFA Tee connector using a micro-syringe (ITO corporation, Japan) in the middle of the injection of dry pure G3 air. Diethyl ether in nitrogen gas (mass fraction of 0.4

%; Takachiho, Japan) used as an OH radical scavenger was introduced to the chamber in excess amounts (approximately 53 ppmv; 164–1963 times the initial concentration of α-pinene) after the introduction of pure air. Seed aerosol particles generated by a commercial atomizer (ATM220S, TOPAS GmbH, Germany) were subsequently introduced into the Teflon chamber after being dried by a diffusion dryer containing silica gel. Neutral seed aerosols were generated from a 0.6/0.3 mol L$^{-1}$ (NH$_4$)$_2$SO$_4$ solution and acidic seed aerosols were generated using a solution mixture of 0.25 mol L$^{-1}$ (NH$_4$)$_2$SO$_4$ and 0.25 mol L$^{-1}$ H$_2$SO$_4$.

After 30 min of stabilization, the initial concentrations of α-pinene and seed aerosol particles were measured with a quadrupole-type proton transfer reaction mass spectrometry instrument (PTR-QMS500, Ionicon Analytik Gesellschaft m.b.H., Innsbruck, Austria) and a scanning mobility particle sizer (SMPS, TSI classifier model 3082; differential mobility analyser model 3081; condensation particle counter model 3772; USA), respectively. The PTR-MS was operated at a flow rate of approximately 250 cm$^3$ s$^{-1}$ under a field strength (E/N, where E is the electric field strength (V cm$^{-1}$) and N is the buffer gas number density

(molecule cm$^{-3}$) of the drift tube) of 106 Td. The length of the drift tube was 9.2 cm. The drift voltage was set to 400 V. The temperatures of the inlet and drift tubes were set to 105°C and the pressure at the drift tube was set to 2.1 mbar. The signal intensities of ions with m/zs of 21, 30, 32, 37, 45, 46, 75, 81, and 137 were recorded approximately every 4.5 sec. The detection sensitivity of α-pinene was 3.3±0.6 ncps ppbv$^{-1}$ (ncps means normalized counts per second to 10$^6$ cps of H$_3$O$^+$). The SMPS was contained in a smaller constant temperature cabinet (LP-280-E, NK System, Japan), whose temperature was adjusted to

be the same as the cabinet containing the Teflon chamber. The sheath and sample flow rates of the SMPS were 3.0 and 0.3 L min$^{-1}$, respectively. The measured diameter range of the SMPS was 13.8–697.8 nm, and the data were collected every 5 min. After obtaining the initial concentration of α-pinene and seed particles, excess ozone produced by irradiation of pure O$_2$ with vacuum ultraviolet light from a low-pressure mercury lamp ozone generator (Model 600, Jelight Compony Inc., USA) was introduced into the chamber at a flow rate of 200 standard mL min$^{-1}$ for 1.5 min, to initiate the ozonolysis reactions. After the

ozone generator was turned off, the introduction of pure O$_2$ continued for another minute to purge all generated ozone into the Teflon bag. Subsequently, the G3 pure air was introduced for one minute to facilitate the mixing of the chamber air. The ozone concentration in the chamber was measured with an ozone monitor (Model 1200, Dylec, Japan) immediately after its introduction. For some experiments, the order of the introduction of α-pinene and O$_3$ was inverted (see Tables S1 and S2). In the experiments in which O$_3$ was first introduced, the introduction of G3 pure air was sustained for one more minute after the

injection of α-pinene to purge all α-pinene into the Teflon bag and to facilitate the mixing of the chamber air. In the latter case, the maximum α-pinene concentrations appeared within 55 s of its introduction, which indicated that the mixing by introducing air with a flow rate of 20 SLM was probably completed within 55 s. The concentrations of both the α-pinene and aerosol particles were continually measured until the end of the experiment, which is defined in this study as 90 min after the start of the α-pinene ozonolysis reaction. The concentration of ozone after 90 min of ozonolysis reactions was also measured. In total,

40 experimental runs were executed under neutral or acidic seed aerosol conditions at temperatures of 278, 288, or 298 K (Tables S1 and S2). Notably, when the chamber temperature was set to 278 K, the temperature in the small cabinet was set to 280 K, which is the lowest work temperature of the SMPS. In the present study, we did not investigate the influence of humidity

on SOA yield. Since very dry conditions are not realistic in ambient air, we carried out the experiments at medium humidity (26–55 % RH, Tables S1 and S2). The differences in RH among experiments in the present study would not influence SOA

formation significantly, as explained in the following section. First, nucleation would be negligible in all experiments because of the high concentrations of seed particles applied. Consequently, the influence of RH on SOA formation would be reflected in the particle phase (Kristensen et al., 2014). In addition, because the seed particles were dried into effloresced states before being introduced into the chamber, all particles would be in solid (neutral seed conditions) or near solid states (acidic seed conditions) (Tang and Munkelwitz, 1977). Therefore, the influence of water on the particle phase through physical partitioning

or chemical reactions would be minor (Faust et al., 2017). Before each experimental run, the Teflon bag was cleaned by filling it with pure G3 air and then evacuating all the air from the bag at least three times, which took approximately 40 min. The very low chamber background particle concentrations indicate that the bag was sufficiently cleaned (Text S2).

One Teflon filter (PF020, 47 mm diameter, Advantec MFS) aerosol sample was collected for each different combination of seed and temperature condition. The sample volume was 0.5 m$^3$ for each sample. In total, six aerosol samples were collected,

and they were subjected to negative electrospray ionization liquid-chromatography time-of-flight mass spectrometry analyses (Sect. 2.2). A blank filter was also analyzed using a procedure similar to that of the sample filters. The results confirmed no substantial contamination in the filter and the filter analysis procedure (Text S2). The acidity of the seed particles was measured in a separate experiment where the seed particles were sampled on a Teflon filter which was then extracted into 10 mL MiliQ water. The pH of the water solution was measured with a pH meter (FPH70, AS ONE, Japan). The H$^+$ concentration was ~220

nmol m$^{-3}$ under the acidic seed conditions.

## 2.2 (−) ESI LC-TOF-MS analysis

Chemical composition analysis of the Teflon filter samples was conducted using electrospray ionization liquid-chromatography time-of-flight mass spectrometry (ESI LC-TOF-MS) (Agilent Technologies, UK) similarly as in previous studies (Sato et al., 2018, 2019) except that negative-mode was used in this study whereas positive-mode was used in those

previous studies. The key configuration parameter settings were as follows: nebulizer pressure was 0.21 MPa; the voltage in the spray chamber was −3500 V; the drying nitrogen gas temperature was 325°C and flow rate was 5 L min$^{-1}$; and the fragmentor voltage was 175 V. The mass calibration and lock-mass correction were conducted using G1969-85000 and G1969-85001 tuning mixtures (Agilent Technologies, UK), respectively. The mass resolution of the mass spectrometer (full width at half maximum) was > 20000. For the analysis, the Teflon filter sample was sonicated in 5 mL methanol for 30 min after the

addition of internal standard (i.e., sodium ethyl-d5 sulfate methanol solution, Sect. 3.2). The filter extract was concentrated to near dryness under a stream of nitrogen (~1 L min$^{-1}$). A 1 mL formic-acid–methanol–water solution ($v/v/v$ = 0.05/100/99.95) was added to the concentrated extract to obtain the analytical sample. A 10 μL aliquot of the analytical sample was injected into the LC-TOF-MS instrument and separated with an octadecyl silica gel column (Inertsil ODS-3; GL Science, Japan; 0.5 μm × 3.0 mm × 150 mm). A formic-acid–water solution (0.05 % $v/v$) and methanol were used as mobile phases. The total

flow of the mobile phases was 0.4 mLmin$^{-1}$. The methanol fraction during each analysis was set at 10 % (0 min), 90 % (30 min), 90 % (40 min), 10 % (45 min), and 10 % (60 min). As reported previously (Sato et al., 2007), the recovery of malic acid, whose saturation concentration was estimated to be 157 μg m$^{-3}$, was determined to be >90 %, suggesting that evaporation loss during pre-treatment is negligible for molecules with saturation concentrations of ~10$^2$ μg m$^{-3}$ or less. We tentatively determined the molecular formulae and signal intensities of 362 products (including 11 organosulfates) with different m/z (Table S3) based

on retention times and interpretation of mass spectra. In addition, the tentatively determined molecular structures and compound names of some major products based on literature data and the results of this study (Sect. 4.2) are presented in Table S4.

**2.3 Evaluation of wall-loss**

The wall-loss rate of particles in the chamber was evaluated by measuring the time evolution of the volume-size distributions of seed-only particles using the SMPS. The measurements were carried out whenever a new Teflon bag was used or the experimental conditions (i.e., temperature or seed particle acidity) were changed under humid air conditions. The latest measured bulk wall-loss rate (Sect. 3.1) was applied for each SOA formation experiment.

In Fig. S4, size-resolved particle wall-loss rates, which were determined assuming first-order wall-loss constants (Wang et al., 2018a), were shown for seed particles of different size-distributions. Large wall-loss was observed for particles with mobility diameters less than 100 nm and larger than 200 nm. The size-distributions of the measured particle wall-loss rates presented shapes similar to that of a 0.83 $m^3$ Teflon chamber (Hu et al., 2014), whereas in the latter, the lowest wall-loss rates appeared in the smaller size end (~70–110 nm) and were greater (~0.2 $hr^{-1}$) than those in the present study. The large apparent wall-loss rates of sub-100 nm particles were also similar to those of a 1.5 $m^3$ Teflon reactor (Wang et al., 2018a). Model simulation (Text S3) and literature survey results revealed that the high wall-loss rates of sub-100 nm particles were mainly caused by particle coagulation (Nah et al., 2017; Wang et al., 2018a) and those of super-200 nm particles were likely the result of turbulent deposition (Lai and Nazaroff, 2000). Figure S4 also indicates that the wall-loss rates of super-200 nm particles were relatively high when the mean diameter of the seed particles was relatively small. Therefore, we waited for 30 min after the introduction of seed particles to start the ozonolysis reaction so that the size distribution of the seed particles could shift to the larger size end due to coagulation and loss of small particles. In addition, we used high concentration solutions for the generation of seed particles to produce larger particles in this study (Sect. 2.1).

Wall-loss of gas-phase organic compounds in the Teflon chamber could also cause the underestimation of SOA yields (Zhang et al., 2014; Krechmer et al., 2016). Although not experimentally determined in the present study, the influence of gas-phase wall-loss on SOA yields will be discussed based on the studies of Zhang et al. (2014) and Krechmer et al. (2016) in Sect. 4.1.

**3 Data analysis**

**3.1 Derivation of SOA yield**

SOA yield ($Y$) is defined as the ratio of the mass concentration of SOA ($m_{SOA}$, µg $cm^{-3}$) to that of the reacted α-pinene ($\Delta_{VOC}$, µg $cm^{-3}$) in each experimental run.

$$Y = \frac{m_{SOA}}{\Delta_{VOC}} \qquad (1)$$

where $m_{SOA}$ was calculated as the product of the increased volume of aerosol particles from the volume of seed particles and a SOA density of 1.34 g $cm^{-3}$ (Sato et al., 2018), and the arithmetic mean of the last three data of each experimental run was applied here. It was corrected for particle-phase wall loss using the bulk-volume wall-loss rate, assuming a first-order wall-loss constant which is independent of particle size and reaction time (Pathak et al., 2007b). The size-resolved wall-loss rates were not applied because the bulk wall-loss rates were very close to the size-resolved rates at approximately 300 nm (Fig. S4). The mode diameters of the volume-size distributions of the seed particles (Fig. S5) and of aerosol particles at the end of the ozonolysis reactions were also approximately 300 nm (Fig. S5). A detailed explanation of the derivation of $m_{SOA}$ is presented in Text S4. Note that when the mass loadings of SOA are low, the obtained $m_{SOA}$ and related yields retain greater uncertainties because the subtracted volume concentrations of seed particles from the measured volume concentrations are large (Mei et al., 2013). The influence of gas-phase wall-loss on SOA yield is discussed qualitatively in Sect. 4.1.

In addition, a four-product volatility basis-set (VBS) gas/particle partitioning absorption model (Eq. (2), Donahue et al., 2006; Lane et al., 2008) was applied to assist the interpretation of the observed responses of $Y$ to the chamber temperature and the acidity of the seed aerosol.

$$Y = \sum_i \alpha_i \left( \frac{1}{1 + c_i^*/m_{SOA}} \right) \qquad (2)$$

where $\alpha_i$ is the mass-based stoichiometric yield for product $i$, and $c_i^*$ is the effective saturation concentration of $i$ in µg m$^{-3}$. In this study, $\alpha_i$ is assumed to be temperature-independent (Pathak et al., 2007a) whereas the temperature dependence of $c_i^*$ is accounted using the Clausius–Clapeyron equation as follows:

$$c_i^* = c_{i,0}^* \frac{T_0}{T} \exp \left( \frac{\Delta H_{vap,i}}{R} \left( \frac{1}{T_0} - \frac{1}{T} \right) \right) \qquad (3)$$

where $T_0$ is the reference temperature, which is 298 K in this study; $c_{i,0}^*$ is the effective saturation concentration of $i$ at $T_0$; R is the ideal gas constant; and $\Delta H_{vap,i}$ is the effective enthalpy of vaporization. The temperature dependence of $Y$ is then represented by the substitution of Eq. (3) for $c_i^*$ in Eq. (2). We further assume a constant effective $\Delta H_{vap}$ for all condensable organic compounds. Thus, the four-product basis set has five free parameters: $\alpha_1$, $\alpha_2$, $\alpha_3$, $\alpha_4$, and $\Delta H_{vap}$. Here, $c_0^* = \{1, 10, 100, 1000\}$ µg m$^{-3}$, which was set based on the measured range of $m_{SOA}$ (Sect. 4.1) in this study. Microsoft Excel Solver GRG Nonlinear engine was used for the derivation of the five parameters under neutral or acidic seed particle conditions.

**3.2 Derivation of the ethyl-d5-sulfate equivalent (EDSeq.) yield of OS**

Before the extraction of filter samples, 20 µL sodium ethyl-d5 sulfate methanol solution (50 µg mL$^{-1}$) was added to each sample filter as an internal standard for the quantification of OS. The ethyl-d5-sulfate equivalent (EDSeq.) masses of OSs were determined by comparing the total chromatographic peak areas of OSs to that of the EDS standard with known mass. The EDSeq. masses of OSs were divided by the corresponding air volumes collected to obtain the EDSeq. concentrations of OSs. The EDSeq. molecular yield of OS is defined as the ratio between the estimated EDSeq. concentration of OS ($m_{OS}$, µg cm$^{-3}$) and the reacted mass concentration of α-pinene ($\Delta_{VOC}$, µg cm$^{-3}$). Note that the sensitivity of ESI mass spectrometry is compound specific; therefore, the calculated EDSeq. yield includes the uncertainties that result from compound specific sensitivities.

**3.3 Volatility distribution analysis**

SOA compounds identified from the six filter samples through LC-TOF-MS analysis were subjected to volatility distribution analysis. The saturation concentration ($C^*$) of each chemical compound was calculated and then ascribed to the volatility basis-set (Donahue et al., 2006). Again, 298 K was used as the reference temperature ($T_0$). For compounds whose chemical structures have been suggested by previous researchers, the SPARC online calculator (Hilal et al., 2003; Sato et al., 2018) was used for the derivation of their $C_0^*$. For other compounds, including organosulfates, the following equation from Li et al. (2016) was applied:

$$\log_{10} C_0^* = (n_C^0 - n_C)b_C - n_O b_O - 2\frac{n_C n_O}{n_C + n_O} b_{CO} - n_S b_S \qquad (4)$$

where $n_C^0$ is the reference carbon number; $n_C$, $n_O$, and $n_S$ are the numbers of carbon, oxygen, and sulfur atoms in the molecule, respectively; $b_C$, $b_O$, and $b_S$ are the respective contribution of each atom to $\log_{10} C_0^*$; and $b_{CO}$ is the carbon–oxygen nonideality. For compounds containing only C, H, and O atoms, the values for $n_C^0$, $b_C$, $b_O$, and $b_{CO}$ are 22.66, 0.4481, 1.656, and −0.7790, respectively. For OS compounds that contain C, H, O, and S atoms, the values for $n_C^0$, $b_C$, $b_O$, $b_{CO}$, and $b_S$ are 24.06, 0.3637, 1.327, −0.3988, and 0.7579, respectively. The $\log_{10} C_0^*$ of the 362 compounds determined by LC-TOF-MS analysis are presented in Table S3. As has been noted previously, the sensitivity of ESI mass spectrometry is compound specific, thus the calculated distribution includes the uncertainties that result from compound specific sensitivities. The estimated volatility distributions were further used to estimate the influence of gas-phase wall-loss on SOA yields following the method suggested by Krechmer et al. (2016) (Text S5).

**4 Results and discussion**

**4.1 Performance of the Teflon chamber**

An example experimental run of SOA formation from α-pinene ozonolysis is presented in Fig. 1 (Exp. No. 27). The initial concentrations of α-pinene and ozone were 145 and >824 ppbv, respectively. The number concentration of seed aerosol particles was $9.6 \times 10^3$ cm$^{-3}$ and they were concentrated in the diameter range of 80–200 nm. SOA was formed while α-pinene was consumed immediately at the introduction of excess $O_3$. The mass concentration of SOA reached its maximum while α-pinene was almost totally consumed approximately 50 min after the introduction of ozone. The time variation of α-pinene conforms to an α-pinene limited first-order chemical reaction ($5\tau$ = 47 min). With the particle-phase wall-loss correction, the SOA loading at the end of this experiment was calculated to be 200±34 μg m$^{-3}$, which resulted in a final SOA yield of 26±7 % (Table S2). As the corrected SOA particle concentration was constant after 50 min, we consider the wall-loss correction applied here was reasonable (Ng et al., 2006). Without particle-phase wall-loss correction, the concentration of SOA particles at 90 minutes would be underestimated by approximately 40 %. The observed particle number-size distribution shifted to a much greater but narrower size range of 200–300 nm at the end of the experiment. Evolution of the particle number-size distribution is presented in Fig. S6.

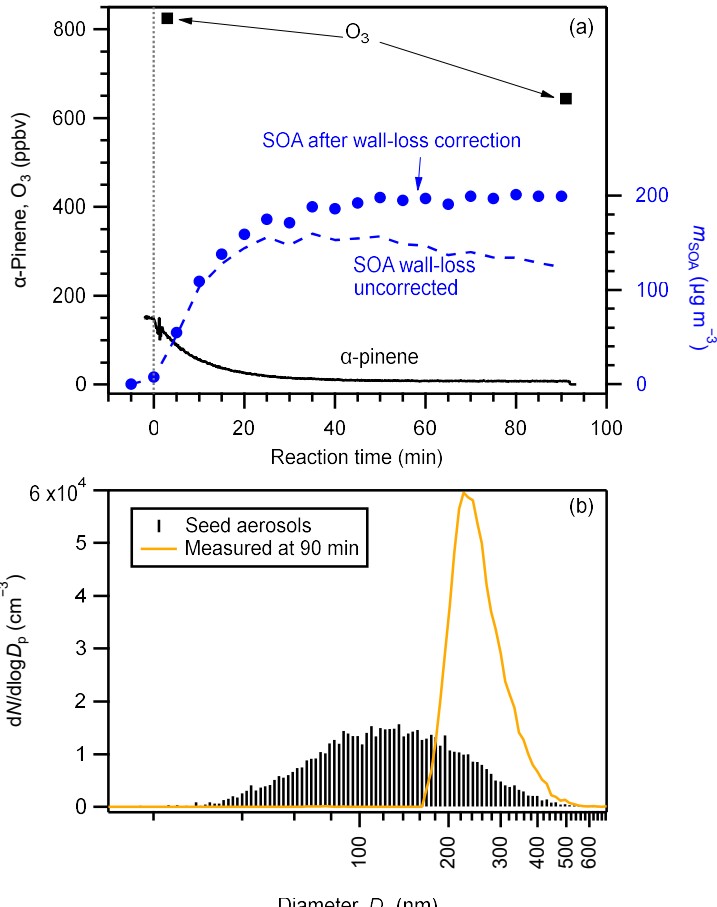

**Figure 1: Example experimental run: (a) concentrations of α-pinene (black solid curve), $O_3$ (square markers), and SOA before (blue dashed line) and after (circle markers) particle-phase wall-loss correction compared with reaction time; and (b) number-size distributions of seed aerosol particles (black sticks) and aerosol particles at the end of the experimental run (brown curve). The vertical dotted line in panel (a) indicates the starting time that $O_3$ was injected. Data shown in this figure are from Exp. No. 27 (Table S2).**

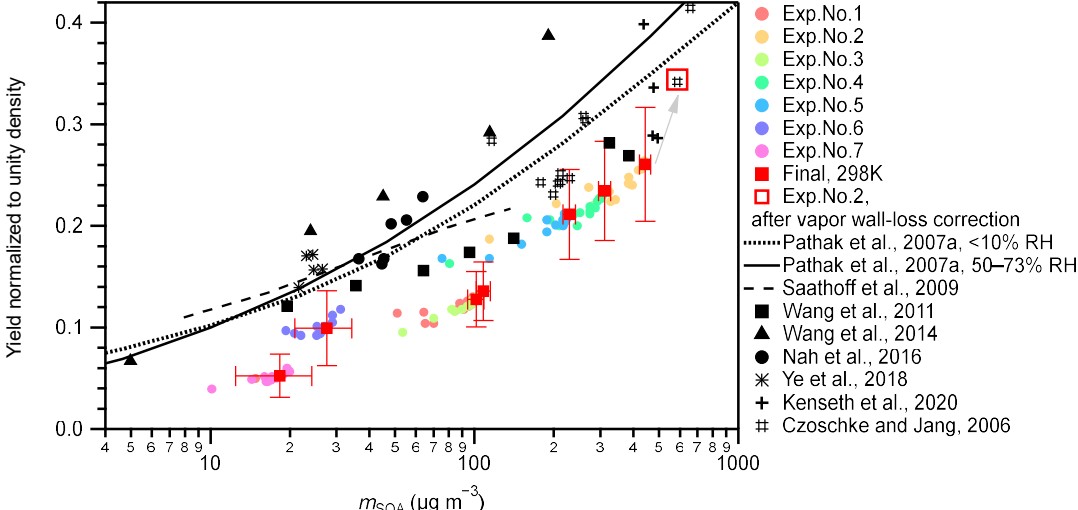

**Figure 2: Yield comparison.** SOA mass yields measured at 298 K under neutral seed conditions in the present study were compared to those of previous studies. Colored markers represent the results of this study. Colored circular markers represent the real time SOA yields, i.e., the SOA yields along with the α-pinene ozonolysis reactions from 0 to 90 min. Different experimental runs are differentiated by colors. Red solid square markers represent the final SOA yields of the seven experiments. Horizontal error bars indicate the uncertainties of the final SOA concentrations; vertical error bars indicate the uncertainties of the final SOA yields.
However, the systematic errors from vapor wall-loss are not included. The open square indicates the result of Exp. No. 2 after gas-phase wall-loss correction (Text S5). Black markers and curves represent results of previous studies. Black markers represent experimental results. The solid and dotted black curves represent the parameterized results from the four-product volatility basis-set fittings of previous α-pinene ozonolysis experiments under low NOx and dark conditions summarized in Pathak et al. (2007a). The solid curve represents results under a 50–73 % RH range and the dotted curve represents results under RH<10 %. The dashed
curve represents the results calculated using Eqs. 1, 6, 7, 10, and 11 at 303 K in Saathoff et al., 2009. The experiments of Saathoff et al. (2009) were carried out at 303 K, 48–37 and 0.02 % RH, without or with OH scavenger (cyclohexane or 2-butanol); experiments of Wang et al. (2011) were carried out at 295 K, < 1 % RH, without OH scavenger; experiments of Wang et al. (2014) were carried out at 295 K, < 5 % RH, without OH scavenger; experiments of Nah et al. (2016) were carried out at 298 K, < 5 % RH, with cyclohexane as OH scavenger; experiments of Ye et al. (2018) were carried out at 296 K, 12–14 and 48–49 % RH, with cyclohexane
as OH scavenger; experiments of Kenseth et al. (2020) were carried out at 295 K, <5 % RH, without OH scavenger; and experiments of Czoschke and Jang (2006) were carried out at 294–300 K, 14–67 %, without OH scavenger. Note that all data presented in this figure are normalized to unity density (1 g cm$^{-3}$).

The measured SOA mass yields at 298 K under neutral seed conditions were compared with that of Pathak et al. (2007a) and other experimental studies (Fig. 2). In this study, seven experiments with different initial α-pinene concentrations (54–323
ppbv) at 298 K and neutral seeds were conducted under a RH condition of approximately 26–27 % (Table S1). In Pathak et al. (2007a), the SOA mass yields from pre-existing studies under low NOx dark ozonolysis conditions were summarized and categorized into two groups according to experimental RH conditions: RH<10 % and RH=50–73 %. They further fitted the data in each group using the multiple product basis-set approach (Pathak et al., 2007a). The four-product basis-set fitting was adopted to compare with the experiment results of this study (Fig. 2). The SOA yields in this study were 25–60 % lower than
that of Pathak et al. (2007a). Possible reasons may include lack of consideration of the wall-loss of oxidized organic vapors because the surface to volume ratio of the chamber used in this study (7.1 m$^{-1}$) was much larger than those of previous studies (<3 m$^{-1}$; Pathak et al., 2007a and references therein). According to Zhang et al. (2014), the vapor wall-loss bias factor, $R_{wall}$ (defined as the ratio of the SOA mass when the vapor wall loss was assumed to be zero, to the SOA mass when the optimal vapor wall loss rate was applied), was reported to be ~4 at the initial seed surface area of ~2 × 10$^3$ µm$^2$ cm$^{-3}$ (seed-to-chamber
surface area ratio ~1 × 10$^{-3}$) and ~2 at the initial seed surface area of >6 × 10$^3$ µm$^2$ cm$^{-3}$ (seed-to-chamber surface area ratio >3 × 10$^{-3}$) during the photooxidation of toluene. As the initial seed surface area in the present study was in the range of (1−3) × 10$^3$ µm$^2$ cm$^{-3}$ (seed-to-chamber surface area ratio (1−4) × 10$^{-4}$), $R_{wall}$ in the Teflon bag might be at least twice that of the large chambers. This leads to the underestimation of the SOA yield of 50 % compared with the values obtained from the large chambers. A low limit correction of the gas-phase wall-loss influence for Exp. No. 2 and other experimental runs in which
chemical composition analysis had been conducted based on the method of Krechmer et al. (2016) (Text S5, Fig. 2), which confirmed that gas-phase wall-loss is one reason for the lower SOA yields in the present study compared with Pathak et al.

(2007a) and other previous studies presented in Fig. 2. We note that this is a shortcoming of the compact chamber with 0.7 m³ volume used in this study. We also note that the chamber is aimed for exploratory research, where multiple experiments under different temperature, seed particle, relative humidity, oxidant, and radiation conditions can be executed within relatively short

periods. Furthermore, the vapor wall-loss correction factors of the SOA mass presented no obvious temperature-dependence (Text S5). Therefore, the temperature dependence of SOA yields will be discussed assuming that the underestimation of the SOA yield due to the wall-loss of oxidized organic vapors does not affect the temperature dependence.

**4.2 Temperature and acidity dependence of SOA yield**

The yields of SOA from α-pinene ozonolysis under different experimental conditions in this study are summarized in Fig. 3.

Results under neutral and acidic seed conditions are presented separately in panels (a) and (b). In each panel, the measured SOA yields as a function of SOA mass loadings are indicated by markers and the four-product VBS model fitting results are shown by curves. The fitted parameters with the four-product VBS model under neutral and acidic seed conditions are summarized in Table 1. Under neutral seed conditions, the fitted temperature independent stoichiometric yields $\alpha$ = {0.00, 0.09, 0.09, 0.52} and $\Delta H_{vap}$ is 25 kJ mol⁻¹. Under acidic seed conditions, $\alpha$ = {0.00, 0.14, 0.05, 0.43} and $\Delta H_{vap}$ is 44 kJ mol⁻¹.

The fitting results pointed out that most of the detected SOA compounds are of relatively high saturation concentration (i.e., $c^*$ at 298 K of 1000 µg m⁻³) under both neutral and acidic seed conditions. Weak increases of SOA yields with decreases of the chamber temperature can be observed under both seed conditions. This is consistent with the result in Pahtak et al. (2007b), which found a weak dependence of SOA yields on temperature in the range of 288–303 K. As temperature decreases, SVOCs tend to partition into the particle phase because of the lowering of their volatilities.

The effective $\Delta H_{vap}$ of α-pinene ozonolysis SOA derived from the four-product VBS fitting in this study was compared with previous studies (Table 2, Fig. S7). The $\Delta H_{vap}$ values derived in this study were comparable to those in Saathoff et al. (2009) and Pathak et al. (2007a) where the experiments were executed in large chambers of 10–200 m³. It may support our assumption that the temperature dependence of SOA yields was not influenced by vapor wall-loss. It is also in agreement with the $\Delta H_{vap}$ of 40 kJ mol⁻¹ applied in the CMAQv4.7 model (Carlton et al., 2010). However, they are lower than those of Saha et al. (2016)

and much lower than those of Epstein et al. (2010). While the $\Delta H_{vap}$ in Saha et al. (2016) was derived based on thermodeneuder measurements which attributed 20–40 % of SOA mass to low-volatility material ($C^* < 0.3$ µg m⁻³), most of the measured SOA in this study are of relatively high volatility as aforementioned. The differences in the volatility of SOA and the derived $\Delta H_{vap}$ between Saha et al. (2016) and this study are consistent with the general phenomenon that $\Delta H_{vap}$ is conversely related with volatility (Epstein et al., 2010). The $\Delta H_{vap}$ in Epstein et al. (2010) was derived from published experimental vapor pressure

data of organic compounds. The reason for the difference in $\Delta H_{vap}$ between this study and Epstein et al. (2010) cannot be currently explained. Notably, sensitivity analyses achieved by fixing the stoichiometric yields $\alpha_i$ while changing $\Delta H_{vap}$ and comparing the resulting VBS curves with measured data (Fig. S8) indicated that the effective $\Delta H_{vap}$ could be in the 0 to 70 and 0 to 80 kJ mol⁻¹ ranges for neutral and acidic seed conditions, respectively.

Figure S9 presents the comparisons of the SOA yields under neutral and acidic seed conditions at different temperatures. It

indicates that the SOA yields were enhanced under acidic seed conditions when the SOA loadings were low. When the SOA loadings were high, the enhancement disappeared. This is consistent with the results of Gao et al. (2004), which reported obvious initial α-pinene concentration dependence of the enhancement of SOA yields under acidic conditions when compared with neutral seed conditions. For the initial α-pinene concentrations of 12, 25, 48, 52, 96, 120, and 135 ppbv, the relative enhancements of SOA yields were 37, 34, 26, 24, 15, 10, and 8 %, respectively (Gao et al., 2004). This is probably because

the SOA components can be of high viscosity under conditions where RH is smaller than around 50 %, and if high SOA mass loadings coated the seed particles, the acid-catalyzed heterogeneous SOA formation reactions could be impeded (Shiraiwa et al., 2013b; Zhou et al., 2013). In this study, the initial concentrations of α-pinene were 54–323 ppbv at 298 K, suggesting that

the enhancement could be less than 24 %. When the SOA volume loading was 50 μm³ cm⁻³, the fitted SOA yields under acidic conditions were enhanced by 11, 17, and 25 % when compared to the neutral seed conditions under 298, 288, and 278 K, respectively. This is consistent with the findings of Gao et al. (2004) and is also comparable to the results of Iinuma et al. (2005). In Iinuma et al. (2005), the experiment with 2-Butanol as an OH radical scavenger under room temperature (294–298 K) reported an enhancement of 19 %, with a final SOA volume concentration of approximately 50 μm³ cm⁻³. Furthermore, the degree of acidity of the seed aerosols could have also influenced the enhancement (Gao et al., 2004; Czoschke and Jang, 2006). Further comprehensive studies are warranted (including the consideration of the particle viscosity and phase separation) on the influence of seed aerosol acidity on α-pinene ozonolysis SOA formation.

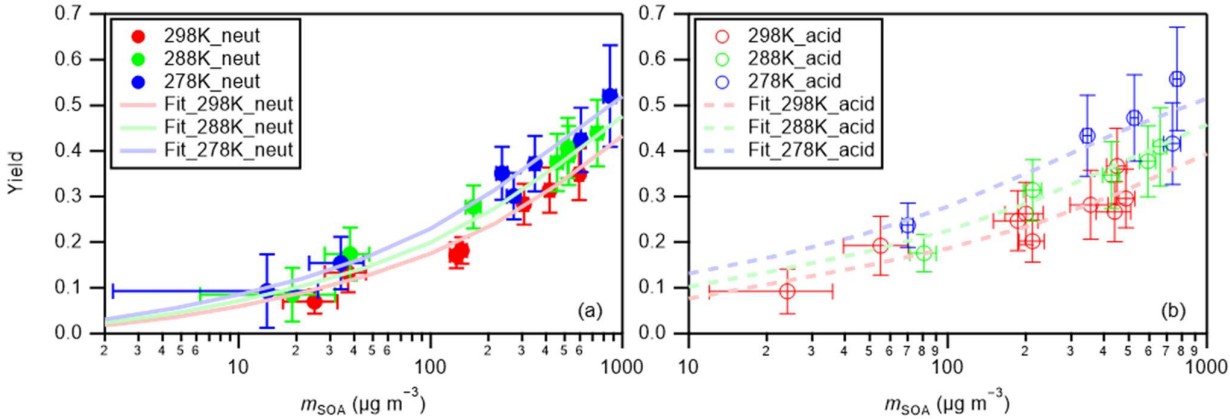

**Figure 3: Mass yields of SOA for increasing SOA mass loadings. Markers and whiskers are measured data and their uncertainties (Text S4; the uncertainties of the data during the reaction time of 85–90 min are presented); curves are fitting of measured data using a four-product VBS model (Donahue et al., 2006; Lane et al., 2008). Panel (a) presents results under neutral seed conditions, and panel (b) presents results under acidic seed conditions.**

**Table 1: Four-product VBS model fitting results.**

| Seed particles | RH (%) | Temperature (K) | $c_1^*$ | $c_2^*$ | $c_3^*$ | $c_4^*$ | $\alpha_1$ | $\alpha_2$ | $\alpha_3$ | $\alpha_4$ | $\Delta H_{vap}$ (kJ mol⁻¹) |
|---|---|---|---|---|---|---|---|---|---|---|---|
| | | | \multicolumn{4}{c}{$c^*$ values (μg m⁻³)} | \multicolumn{4}{c}{$\alpha$ (Stoichiometric Yields) values} | |
| **Neutral** | ~26–27 | 298 | 1.000 | 10.00 | 100.0 | 1000 | | | | | |
| | ~32–34 | 288 | 0.7292 | 7.292 | 72.92 | 729.2 | 0 | 0.092 | 0.087 | 0.52 | 25 |
| | ~45–55 | 278 | 0.5191 | 5.191 | 51.91 | 519.1 | | | | | |
| **Acidic** | ~26–27 | 298 | 1.000 | 10.00 | 100.0 | 1000 | | | | | |
| | ~32–34 | 288 | 0.5550 | 5.550 | 55.50 | 555.0 | $2.8 \times 10^{-5}$ | 0.14 | 0.048 | 0.43 | 44 |
| | ~45–55 | 278 | 0.2949 | 2.949 | 29.49 | 294.9 | | | | | |

**Table 2: Comparison of $\Delta H_{vap}$ in this study with previous studies.**

| References | $\Delta H_{vap}$ (kJ mol⁻¹) | $C^*$ ranges (μg m⁻³) | Temp ranges (°C) |
|---|---|---|---|
| This work; ozonolysis | 25–44 | $10^1$–$10^3$ | 5–25 |
| Saha et al., 2016; ozonolysis | $80-11 \log_{10}C^*$ | $10^{-2}$–$10^4$ | 30–120 |
| Saathoff et al., 2009; ozonolysis | 24–59 | $2.1 \times 10^{-3}$–56 | −30–40 |
| Pathak et al., 2007a; ozonolysis | 30, 70 | $10^{-2}$–$10^4$ | 0–49 |
| Epstein et al., 2010; semiempirical correlation-based fit | $129-11 \log_{10}C^*$ | $10^{-2}$–$10^{10}$ | 27 |
| Carlton et al., 2010; CMAQv4.7 SOA module | 40 | 15, 134 | 40 |

## 4.3 Temperature and acidity dependence of SOA composition

Among the 362 compounds identified through LC-ToF-MS analysis in this study (Table S3), 331 compounds were ascribed to VBS bin ranges of −8–3. The other 31 compounds were ascribed to higher VBS bin ranges of 4–6. Only the former 331 compounds are targeted in the following discussions for two reasons. First, less than a half of the compounds that belong to

VBS bins four or greater could exist in the particle phase (Donahue et al., 2006), which would introduce large uncertainties for the estimation of the mass concentrations of the compounds in gas-phase from the particle phase. In addition, LC-ToF-MS analysis of pure compounds indicated that fragmentation of high molecular compounds during the ionization could occur, e.g., pinic acid ($C_9H_{14}O_4$) could be fragmented into $C_8H_{14}O_2$, and the latter was assigned to VBS bin 6 (Fig. S10). Note that due to the potentially high viscosity of the newly formed SOA, high-volatility compounds formed inside the aerosol particle could have been wrapped into the particle phase and detected (Shiraiwa et al., 2013b).

Figure 4 presents the volatility distributions of the identified compounds. The measured intensities of particle phase compounds were normalized by their total intensity for each experiment and are presented in Fig. 4a. Compounds that were attributed to VBS bins between 0 and 3 are known as SVOC (Li et al., 2016). Those in bins 2 and 3 generally presented a decreasing tendency with the increase of experimental temperatures, and there was no obvious temperature dependence for those in bins 0 and 1 (Fig. 4a). The corresponding gas phase concentrations of each compound were derived assuming gas-particle partitioning equilibrium (Odum et al., 1996). The $\Delta H_{vap}$ values derived in this study (Table 1) were used to calculate the saturation concentration under 278 and 288 K following Eq. (3). The intensities of both the particle and gas phases were then normalized by their total amount and are presented in Fig. 4b. With the inclusion of the corresponding gaseous phase compounds, the temperature dependence of compounds in VBS bin 3 changed to positive (i.e., increased with temperature) whereas the tendency in bin 2 remained negative (Fig. 4b). As the α-pinene ozonolysis rate constant at the temperature range of 278–298 K does not vary considerably (within 15 %; IUPAC Task Group on Atmospheric Chemical Kinetic Data Evaluation, (http://iupac.pole-ether.fr, last access: 1 February 2021)), α-pinene was completely consumed at the reaction time of 90 min, and the temperature-dependence of gas-phase wall-loss was considered insignificant (Text S5), the total amounts of SVOCs formed should be similar at the three temperatures. Hence, the total amounts of gas and particle phase compounds in each VBS bin should be independent of experimental temperatures. If a larger $\Delta H_{vap}$ were applied for the derivation of the intensity of gas phase compounds (e.g., the semiempirical equation in Epstein et al. (2010)), the total amount in VBS bin 3 at 298 K would be much higher than at other temperatures, which is unreasonable. This suggested the appropriateness of the $\Delta H_{vap}$ values derived from the four-product VBS fitting of the SOA yields in Sect. 4.1. In addition, we note that the volatility distribution pattern derived in this study is similar to that of the experimental runs 1 and 6 (derived using the same method) of Sato et al. (2018), although positive electrospray ionization analysis was used and the α-pinene ozonolysis experiments were carried out under dry conditions in the latter. According to Morino et al. (2020), both the root-mean-square errors between the observed and simulated SOA concentrations for the formation experiments and between the observed and simulated volume fraction remainings for the heating experiments were minimized in the case of the $C^*$ distribution reported by Sato et al. (2018). In Fig. 4b, the volatility distributions determined by normalizing the stoichiometric SOA yields (Table 2) based on the total mass fractions of compounds in VBS bins 0 to 3 are also presented. The mass fractions determined from the LC-TOF-MS data at VBS bins 1 and 3, and 0 and 2, were smaller and larger, respectively, than those determined from the SOA stoichiometric yields. These differences are probably caused by uncertainties in saturation concentration and sensitivity parameterization as well as the existence of undetected molecules in the LC-TOF-MS analysis (Sato et al., 2018).

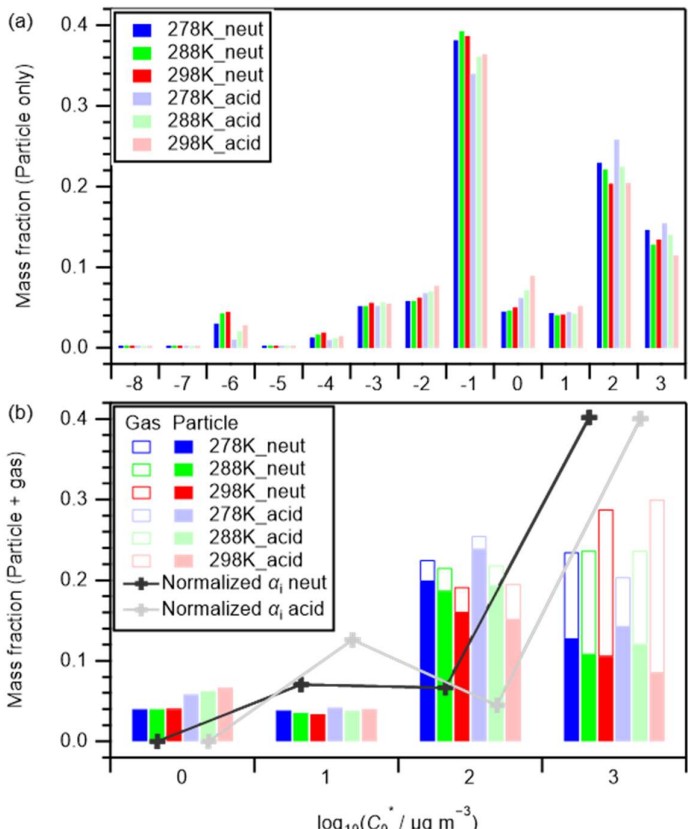

**Figure 4: Volatility distributions of chemical components in (a) particle only phase and (b) particle + gas phases. In panel (a), data were normalized by the total signal intensity of the particle phase of each experiment; in panel (b), data were normalized by the total intensity of the particle phase and its equilibrium gas phase in each experiment. Curves and markers imposed in panel (b) are the volatility distributions under neutral (black) and acidic (grey) seed conditions determined by normalizing the stoichiometric SOA yields $\alpha_i$ by the total mass fraction of compounds in VBS bins 0 to 3 (Sato et al., 2018).**

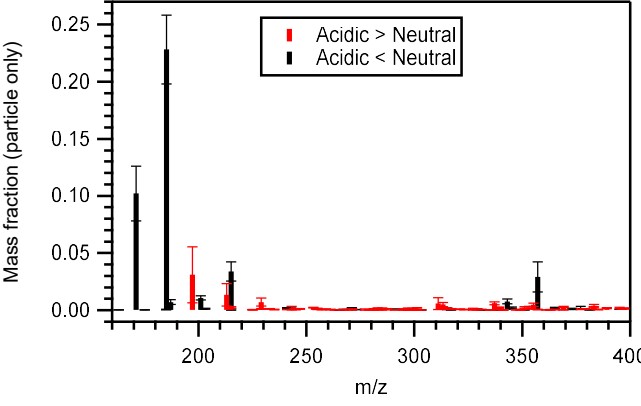

**Figure 5: Mean particle phase mass fraction distributions of compounds whose particle phase mass fractions under acidic seed conditions were more than 1.1 (red symbols) or less than 0.9 (black symbols) times those of neutral seed conditions. Whiskers represent the standard deviations of the three experiments at different temperatures.**

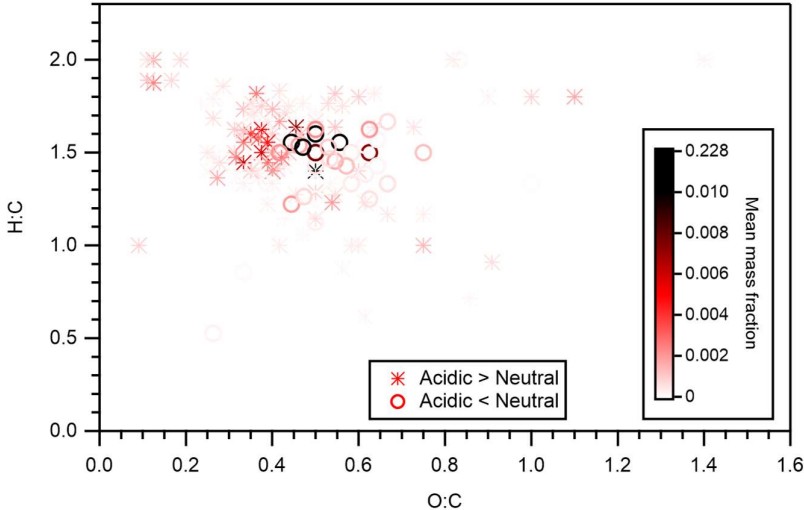

Figure 6: Atomic H to C ratio (H:C) versus O to C ratio (O:C). Star and Circle markers respectively indicate compounds whose particle phase mass fractions under acidic seed conditions were more than 1.1 and less than 0.9 times those of neutral seed conditions. Color scale indicates the mean mass fraction of identified compounds across all six experiments. Notably, organosulfates were not presented.

To gain more insights into the acidity dependence of the α-pinene ozonolysis SOA, the relative intensities of the compounds identified by LC-TOF-MS under acidic and neutral seed conditions were compared. For compounds whose intensity under acidic seed conditions was more than 1.1 times that of neutral seed conditions, monomers with a chemical formula of $C_{10}H_{9-18}O_{4-14}$ accounted for 32 % and oligomers with a chemical formula of $C_{11-22}H_{7-35}O_{2-12}$ accounted for 68 % of the total intensity. Conversely, for compounds whose intensity under acidic conditions was less than 0.9 times that of neutral seed conditions, monomers with a chemical formula of $C_{6-10}H_{7-15}O_{4-6}$ accounted for 87 % and oligomers with a chemical formula of $C_{11-21}H_{9-27}O_{5-10}$ accounted for 13 % of the total intensity.

Figure 5 presents the particle phase mass distributions of compounds whose mass fractions under acidic seed conditions were more than 1.1 (or less than 0.9) times those of neutral seed conditions. The molecular mass of those compounds with greater mass fractions under acidic seed conditions was generally distributed in the higher mass end compared with those with greater mass fractions under neutral seed conditions. The mass fraction weighted mean molecular mass of those compounds presented greater intensities under acidic seed conditions was $284 \pm 14$ (mean ± standard deviation) g mol$^{-1}$, whereas of those presented greater intensities under neutral seed conditions was $204 \pm 4$ g mol$^{-1}$.

Figure 6 presents the identified compounds whose intensity under acidic seed conditions was more than 1.1 (or less than 0.9) times that of neutral seed conditions in the atomic H to C ratio (H:C) versus O to C ratio (O:C) space. The color scale indicates the mean mass fraction of identified compounds across all six experiments. It is suggested that compounds with a lower intensity under acidic seed conditions were concentrated in the O:C ratio range of 0.4–0.75 and H:C ratio range of 1.1–1.7, whereas those with higher intensity under acidic seed conditions were more broadly distributed in the H:C versus O:C space. Compounds with O:C ratios less than 0.4 were oligomers with a chemical formula of $C_{11-22}H_{19-35}O_{2-7}$ and were distributed in the VBS bins of −3 to 3. Compounds with O:C ratios greater than 0.75 were likely to be highly oxidized molecules with a chemical formula of $C_{10-14}H_{9-21}O_{9-14}$ and were attributed to VBS bins −8 to −2. Furthermore, oligomers with O:C ratios of less than 0.4 accounted for 61 % of those oligomers with high relative intensity under acidic conditions, whereas those with O:C ratios of greater than 0.75 accounted for only 1 %. This, together with the aforementioned chemical formula and molecular mass distributions, indicated that the formation of many oligomers, especially with small O:C ratios, was enhanced under acidic seed conditions.

Figure 7 presents the compounds whose normalized intensity under acidic seed conditions was less than 0.9 (panel a) or more than 1.1 (panel b) times that of neutral seed conditions in the VBS space. The acidity dependence of the major compounds in

VBS bins was tentatively explained from the viewpoints of acid-catalyzed decomposition reactions or acid-catalyzed heterogenous reactions. Compounds that presented lower intensities under acidic than neutral seed conditions were mainly distributed in VBS bins 3, 2, −1, −4, and −6 (Fig. 7a). The respective compounds that presented the highest intensity in VBS bins 3, 2, −1, −4, and −6 were m/z 215.091 ($C_{10}H_{15}O_5$), 171.065 ($C_8H_{11}O_4$), 185.081 ($C_9H_{13}O_4$), 343.139 ($C_{16}H_{23}O_8$), and 357.154 ($C_{17}H_{25}O_8$), and they accounted for on average 25, 46, 62, 54, and 99 % of the total intensity of the compounds determined in the respective bins. The structures of m/z 215 have been proposed by Zhang et al. (2017) as hydroperoxides from α-pinene ozonolysis reactions. They can be decomposed under acidic seed conditions to m/z 197.081 ($C_{10}H_{13}O_4$, VBS bin 2) (Scheme 1a) as well as m/z 155.010 ($C_8H_{11}O_3$, VBS bin 5) (Scheme 1b) (acid-catalyzed decomposition of hydroperoxides; Seubold and Vaughan, 1953). Next, previous studies indicate that m/z 171 and 185 were monomer precursors of dimers m/z 343 and 357, and all four products exist in both laboratory α-pinene ozonolysis SOA samples and field samples (Gao et al., 2004; Yasmeen et al., 2010; Kristensen et al., 2014; Zhang et al., 2015). The chemical structure of m/z 185 has been proposed as pinic acid (Gao et al., 2004; Yasmeen et al., 2010; Kristensen et al., 2014), and m/z 171 as terpenylic acid (Yasmeen et al., 2010; Kristensen et al., 2014) and norpinic acid (Gao et al., 2004). In this study, excess OH scavengers could have minimized the formation of an OH functional group in the α-pinene ozonolysis products whereas the aldehydes should dominate the products (Gaona-Colmán et al., 2017). Thus, it is possible that the acid-catalyzed heterogenous esterification between acid products such as m/z 171 and 185, and aldehyde products such as pinonaldehyde (Hallquist et al., 2009), led to the decreased intensity of m/z 171 and 185 under acidic seed conditions, which further led to the decreased intensity of the related dimers of m/z 343 and 357. Scheme 2 presents a possible esterification reaction between terpenylic acid (a possible isomer of m/z 171) and pinonaldehyde. The product, m/z 339.180 ($C_{18}H_{27}O_6$), ascribed to VBS bin −1, presented higher relative intensity under acidic seed conditions.

The compounds with higher contributions under acidic than neutral seed conditions were mainly distributed in VBS bins −3, −2, 0 and 2 (Fig. 7b). The most abundant compounds of those whose intensity under acidic seed conditions was more than 1.1 times that of neutral conditions in bins 2, −2, and −3 were m/z 197 ($C_{10}H_{13}O_4$), 355.175 ($C_{18}H_{27}O_7$), and 383.170 ($C_{19}H_{27}O_8$), respectively. Their contributions to the total signal of their respective bins were 14, 7, and 5 %, and the ratios of their intensities under acidic seed conditions to those under neutral seed conditions were 5.4, 1.8, and 1.3, respectively. The higher intensity of m/z 197 under acidic than neutral conditions might be explained partly by the acid-catalyzed decomposition of m/z 215 (Scheme 1a). Zhang et al. (2015) proposed m/z 355 as a dimer ester with a hydroperoxide function. Another possible assignment of m/z 355 is an ester from the reaction of a Criegee intermediate ($C_{10}H_{16}O_3$) with terpenylic acid and/or norpinic acid ($C_8H_{12}O_4$) (Kristensen et al., 2017). The latter assignment of the product also has a hydroperoxide group. The formation of hydroperoxides involving Criegee intermediates has been observed in the gas phase (Sekimoto et al., 2020) and on the surface of aqueous droplets (Enami et al., 2017). The relatively high intensity of m/z 355 under acidic seed conditions might be explained by the fact that the hydrolysis of ester hydroperoxides becomes slower under acidic conditions (Zhao et al., 2018). Kristensen et al. (2017) proposed m/z 383 as a dimer ester, whereas the molecular structure is to be identified. In bin 0, the intensities of m/z 311.149 ($C_{16}H_{23}O_6$) and m/z 313.165 ($C_{16}H_{25}O_6$) were the highest (contributing on average 10 and 9 % of the total intensity of the bin, respectively), and under acidic seed conditions were 4.8 and 1.5 times those of neutral seed conditions, respectively. Although the structure of m/z 311 is yet to be identified, m/z 313 was proposed as a dimer ester by Zhang et al. (2015). These dimer esters at m/z 383 and 313 might be formed through acid-catalyzed heterogeneous reactions.

In addition, the eleven identified organosulfate compounds were ascribed to the VBS bins of −3 to 0 (Table 3). All of them presented greater intensity on average under acidic conditions than under neutral seed conditions (Fig. S11). Four of the OS were in VBS bin 0, which together contributed 9.2 % of the intensity of that bin. The contributions of the two, three, and two OS compounds (which respectively belongs to VBS bins of −1, −2, and −3) to their respective bins were 0.4, 4.7, and 2.6 %. The formation mechanisms of those OS compounds will be discussed in the next section. It should be noted that the intensities

of OS compounds in this study might be underestimated because methanol, which could react with carbonyl and carboxylic compounds in the SOA (Bateman et al., 2008), was used as the extraction solution. Acetonitrile should be used as the extraction solution in future studies to confirm the influence (Bateman et al., 2008).


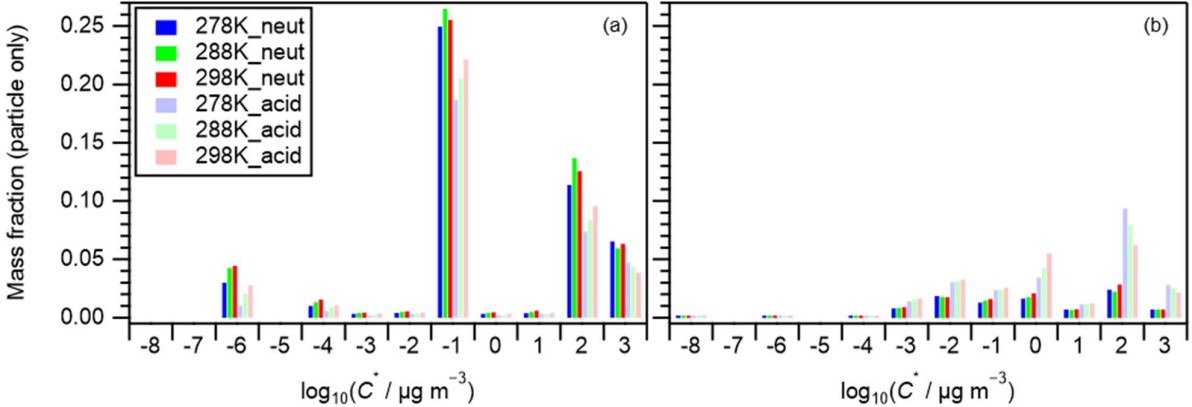

Figure 7: Volatility distribution of SOA compounds whose particle phase mass fractions under acidic seed conditions were (a) less than 0.9 and (b) more than 1.1 times those of neutral seed conditions. Note the mass fractions presented here are same as those in Fig. 4a.

Scheme 1: Proposed acid-catalyzed decomposition reactions of hydroperoxides at m/z 215 in this study.

Scheme 2: Possible acid-catalyzed esterification reaction of Terpenylic acid (m/z 171) with Pinonaldehyde.

### 4.4 OS formation under acidic conditions

In this study, significant signals were identified for eleven organosulfates under acidic seed conditions (Fig. S11). Figure 8 presents the ethyl-d5-sulfate equivalent (EDSeq.) molecular yields of those OS compounds estimated from the LC-TOF-MS analysis. m/z 223 ($C_7H_{12}O_6S$) was the most abundant OS identified at 298 k, followed by m/z 279 ($C_{10}H_{16}O_7S$), 281 ($C_9H_{14}O_8S$), 269 ($C_8H_{14}O_8S$), 283 ($C_9H_{16}O_8S$), 265 ($C_{10}H_{18}O_6S$), 253 ($C_8H_{14}O_7S$), 267 ($C_8H_{12}O_8S$), 251 ($C_9H_{16}O_6S$), 249 ($C_{10}H_{18}O_5S$), and 247 ($C_{10}H_{16}O_5S$).

The potential formation mechanisms of the OS compounds were proposed based on literature data in combination with the results of the experiments of this study. To our knowledge, while the formation of OS through α-pinene ozonolysis in the presence of an OH radical scavenger has been reported very recently (Ye et al., 2018; Stangl et al., 2019; Wang et al., 2019),

these studies focused on the reactions between α-pinene ozonolysis products with $SO_2$ rather the identification of OS compounds. Earlier studies on the identification and formation mechanisms of OS with α-pinene as the precursor VOC related to photooxidation (i.e., OH-initiated oxidation) or nighttime oxidation (i.e., $NO_3$-initiated oxidation under dark conditions) (Surratt et al., 2007, 2008). In this study, the formation of OS from α-pinene ozonolysis in the presence of an OH radical scavenger and acidic seed particles was investigated.

Table 3 summarizes the possible chemical structures of the eleven detected OS compounds and their precursor α-pinene oxidation products. Among the eleven OS compounds detected in this study, eight of them have been detected in previous laboratory and/or field studies of α-pinene oxidations, and the other three (m/z 251, 253, and 269) were detected for the first time in this study (Table 3). While all eleven OS compounds detected in this study are formed from α-pinene ozonolysis due to the presence of excess OH radical scavenger, five of the previously identified OS compounds (m/z 223, 247, 249, 265,

and 279) were detected in α-pinene photooxidation and/or nighttime oxidation experiments (Surratt et al., 2008) and the other three (m/z 267, 281, and 283) were only identified in ambient aerosols.

The chemical structures of five of the OS compounds (i.e., m/z 223, 247, 249, 265, and 279) have been proposed in previous studies. Four of them (m/z 223, 247, 249, and 279) were likely to be sulfate esters formed from alcohols and sulfate (Surratt et al., 2007, 2008; Zhang et al., 2015; Hettiyadura et al., 2019), and one (m/z 265) was said to be from the sulfation of

pinonaldehyde (Liggio and Li, 2006; Surratt et al., 2007). The former is referred hereafter as the alcohol pathway, and the latter the aldehyde pathway (aldehyde + $HSO_4^-$; Surratt et al., 2007) (Scheme S1). Because aldehydes should dominate the products from α-pinene ozonolysis in the presence of an OH scavenger (Gaona-Colmán et al., 2017) and pinonaldehyde has been detected in previous α-pinene ozonolysis studies with an OH scavenger (Jackson et al., 2017), we propose that the aldehyde pathway could be one of the dominant OS formation pathways in this study. In addition to m/z 265, we propose

possible aldehyde precursors formed in α-pinene ozonolysis experiments for five other OS compounds identified in this study (i.e., m/z 251, 253, 269, 279, and 283). Except for the study where the aldehyde precursor of m/z 269 was found (Reinnig et al., 2009), an OH scavenger was used in all other studies where the aldehyde precursors were found (Yu et al., 1999; Ma et al., 2008; Jackson et al., 2017). Presently, we suppose that the other five OS compounds were possibly formed through the alcohol pathway although no relevant precursor information has been found for m/z 223 and 247. Notably,

another possible precursor with a hydroxyl function, which is formed in α-pinene ozonolysis experiments with an OH scavenger, has also been found for the OS at m/z 269 (Kristensen et al., 2014). Additionally, the proposed precursor for m/z 283 (Table 3) also contains a hydroxyl function, which can form another OS of m/z 265.038 ($C_9H_{13}O_7S$) through the alcohol pathway. A weak signal of this OS was observed (not presented in Fig. S11).

As shown in Fig. 8, the ethyl-d5-sulfate equivalent (EDSeq.) yields of m/z 247, 249, and 265 decreased with the increase of

temperature. In fact, the OS at m/z 247 and 249 could only be observed at the lower temperatures. Conversely, the EDSeq. yields of other OS compounds increased with the temperature, or at least the yields at 278 K were the lowest among the three temperature conditions. The temperature dependence of OS EDSeq. yields seems not to be directly related to the formation mechanisms of either the alcohol pathway or the aldehyde pathway. Nevertheless, the mean EDSeq. yields of the eleven OS compounds, which were dominated by mz223 and mz279, increased with the increase of reaction temperature. As eleven OS

(including three unreported) were observed in the α-pinene ozonolysis reactions with an OH scavenger and acidic seed particles, the identification of the structures of the OSs using high-resolution ion mobility mass spectrometry is planned as the next step. When the structures of the OSs can be determined and the precursor compounds can be assumed, the formation mechanisms of OSs in α-pinene ozonolysis with acidic seeds may be confirmed.

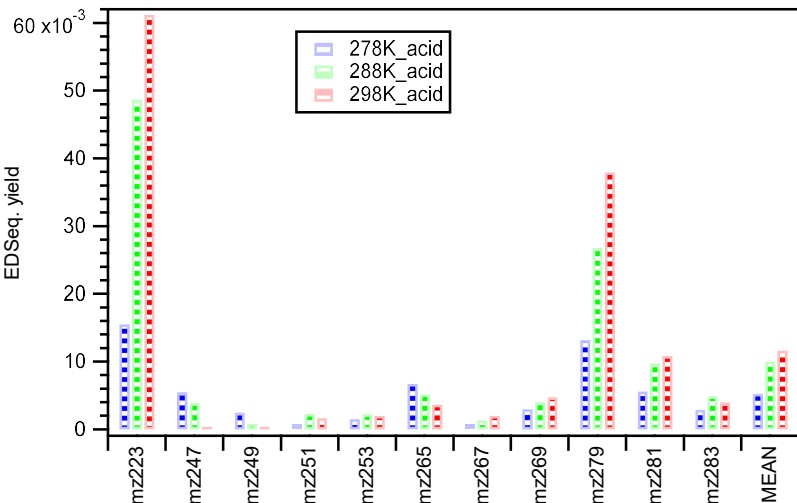


**Figure 8: The ethyl-d5-sulfate equivalent (EDSeq.) yield of the eleven OS compounds and their arithmetic mean under acidic seed conditions at different temperatures. The retention time range used for the calculation of the intensity of each OS compound is indicated in Fig. S11.**


**Table 3: Organosulfates identified in this study, their respective m/z and formula of [M−H]⁻, assigned VBS bin number, proposed OS structure, and proposed precursor oxidation product structure.**

| m/z [M−H]⁻ | Formula [M−H]⁻ | VBS bin | Proposed OS structure | | Proposed precursor | | Ref.[a] |
|---|---|---|---|---|---|---|---|
| | | | Aldehyde + HSO₄⁻ | Esterification | Aldehyde + HSO₄⁻ | Esterification | |
| 223.027 | $C_7H_{11}O_6S^-$ | 0 | - | [structure b] | - | - | bcd |
| 247.063 | $C_{10}H_{15}O_5S^-$ | 0 | - | [structure e] | - | - | cfg |
| 249.079 | $C_{10}H_{17}O_5S^-$ | 0 | - | [structure c] | - | [structure c] | cgh |
| 251.060 | $C_9H_{15}O_6S^-$ | 0 | [structure] | - | [structure i] | - | This study |
| 253.039 | $C_8H_{13}O_7S^-$ | −1 | [structure] | - | [structure i] | - | This study |
| 265.074 | $C_{10}H_{17}O_6S^-$ | −1 | [structure j] | - | [structure jk] | - | cgl |
| 267.017 | $C_8H_{11}O_8S^-$ | −2 | - | [structure] | - | [structure mn] | o |
| 269.033 | $C_8H_{13}O_8S^-$ | −2 | [structure] | [structure] | [structure p] | [structure q] | This study |
| 279.053 | $C_{10}H_{15}O_7S^-$ | −2 | [structure] | [structure j or / j or / structure] | [structure i] | [structure jr or / jr or / c] | cgj |
| 281.033 | $C_9H_{13}O_8S^-$ | −3 | - | [structure] | - | [structure n] | stu |
| 283.048 | $C_9H_{15}O_8S^-$ | −3 | [structure] | [structure (m/z=265)] | [structure v] | - | cs |

[a]Selected laboratory and/or field studies where the corresponding OS has been identified; [b]Hettiyadura et al., 2019; [c]Surratt et al., 2008; [d]Yassine et al., 2012; [e]Zhang et al., 2015; [f]Tao et al., 2014; [g]Ma et al., 2014; [h]Wang et al., 2018b; [i]Yu et al., 1999; [j]Surratt et al., 2007; [k]Jackson et al., 2017; [l]Liggio and Li, 2006; [m]Kahnt et al., 2014a; [n]Kahnt et al., 2014b; [o]Meade et al., 2016; [p]Reinnig et al., 2009; [q]Kristensen et al., 2014; [r]Aschmann et al., 1998, 2002; [s]Brüggemann et al., 2019; [t]Wang et al., 2016; [u]Brüggemann et al., 2017; [v]Ma et al., 2008.


## 5 Summary and conclusions

Using the compact chamber system, SOA formation from α-pinene ozonolysis was studied with diethyl ether as an OH radical scavenger at temperatures of 278, 288, and 298 K, with acidic and neutral seed aerosol particles. The SOA yields and compounds with a molecular mass of less than 400 Da determined using a LC-TOF-MS were analyzed from the perspectives of temperature and seed particle acidity dependence.

The SOA yield increased slightly with the decrease of chamber temperature. The enthalpies of vaporization under neutral and acidic seed conditions was estimated to be 25 and 44 kJ mol$^{-1}$, respectively. The acidity dependence of the SOA yields at low SOA loadings were comparable to those reported by Gao et al. (2004) and Iinuma et al. (2005). However, the enhancement of the SOA yields under acidic conditions would be limited if the SOA mass loadings are much greater than the amounts of pre-existing particles.

Among the 362 compounds identified, the volatility of 331 was distributed in the VBS bins between $-8$ and 3. The temperature dependence of the volatility distribution of those identified compounds (particle phase + gas phase) could be consistently explained by the enthalpies of vaporization derived in this study.

The compounds whose intensities under acidic seed conditions were less than 0.9 times those under neutral seed conditions were dominated by monomers, whereas the compounds whose intensities under acidic seed conditions were more than 1.1 times those under neutral seed conditions were dominated by oligomers. The O:C ratios of the former were concentrated in the range of 0.4–0.75. The O:C ratios of the latter were broadly distributed. The compounds with O:C ratios less than 0.4 were all oligomers, which accounted for 61 % of the oligomers with high relative intensity under acidic conditions, whereas those with O:C ratios of greater than 0.75 were highly oxidized molecules, and only contributed to 1 % of the oligomers. In addition, the mean molecular mass of the former compounds ($204 \pm 4$ g mol$^{-1}$) were evidently lower than those of the latter ($284 \pm 14$ g mol$^{-1}$). The differences indicated that the formation of many oligomers, especially with low O:C ratios, was enhanced under acidic seed conditions. The acidity-dependence of certain major compounds could be explained by acid-catalyzed heterogenous reactions (e.g., m/z 171, 185, 343, and 357) or acid-catalyzed decomposition reactions (e.g., m/z 215 and 197), which suggests that little or no enhancement of SOA under acidic conditions in field observations could occur when acid-catalyzed decomposition is dominant.

For the first time, organosulfate compounds were studied for α-pinene ozonolysis reaction in the presence of an OH scavenger and acidic seed particles. Eleven OS compounds were determined from LC-TOF-MS analysis. All of them on average presented higher yields under acidic than under neutral seed conditions. Six of the OS compounds were potentially formed via the aldehyde + HSO$_4^-$ pathway, which should be confirmed in future studies through high resolution mass spectrometry analyses.

Finally, the organosulfates and the oligomers that increased with an increase in acidity of the seed particles could be indicators of the acidity of pre-existing particles in the field, and the new findings obtained from this study should be confirmed using more complex and larger chambers.

**Data availability.**

All the final data supporting the findings of this study are available in the manuscript or in the supplement. Raw data used to derive the final data are available on request to the corresponding author.

**Author contributions**

SI, YD, SE, and HT did chamber experiments; KS and SR did LC-TOF-MS analyses; SI, YD, SE, and MY did data analyses; YD, SI, and HT made the manuscript; and all authors contributed to the revisions of the manuscript.

**Competing interests.**

The authors declare no competing interests.

**Acknowledgements**

We thank Ms. Sumiko Komori, Dr. Yoshikatsu Takazawa, and Dr. Tomoharu Sano for the technical support. This work was supported by NIES research funding (Type A) and JSPS KAKENHI Grant No. 19H01154.

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
