# Peer review of "Temperature and acidity dependence of secondary organic aerosol formation from $\alpha$ -pinene ozonolysis with a compact chamber system"

_Atmospheric Chemistry and Physics, 2020_

## Referee Comment (RC1) · Anonymous Referee #1 · 12 Oct 2020

In this paper, Temperature and acidity dependence of secondary organic aerosol formation from $\alpha$-pinene ozonolysis with a compact chamber system have been investigated. This is a nice piece of work where the temperature and acidity dependence of SOA yields and chemical components were investigated. In addition, the formation of organosulfates (OS) was observed under acidic seed conditions. The data presented are of significant importance; however some modifications would be good to improve the article. The article should be published after major modifications. Some characteristics of the new chamber are missing. We have developed a new chamber, it would be good to have more specifications. The temperature effect on the SOA formation has been relatively little studied, mainly due to the low number of temperature controlled

chamber. Give more precision on the temperature controlled chamber. There is no uncertainties, it would be good to add it in the text, in the figure and in the table. For example, no uncertainties has given for the temperature, the yield, the concentration . . .. It would be good also to correct the SOA yield obtained by all wall losses (particle and also oxidized organic vapors) to improve the article. It would be good to add also some comments and reference (list of some references given at the end of this pdf file) about the relative humidity (RH) because it's not the same in each experiment and the humidity can also influence the SOA yields and chemical compounds. The effect of RH on aerosol formation has yet to be optimized. Some experiments performed in flow reactor have shown that the aerosol yield was dependent on RH and other studies performed in simulation chamber have reported no effect. For example, the study of Saathoff [2009] on the ozonolysis of $\alpha$-pinene have shown that water had little influence on aerosol yields at room temperature, however, a significant positive effect of water is observed for lower temperatures. Relative humidity is dependent on temperature.

For more explanation, see specific comments.

Specific comments: Introduction Line 41-59: The role of acidity is really good explain. However, the role of the other parameters as the temperature is very succinct, just one sentence. It would be good to explain more the influence of the other parameters on the SOA yields and chemical composition. Line 63-81: The dependence of humidity is not explained for the specific case of $\alpha$-pinene ozonolysis. It would be good to add also some sentences and references (list of some references given at the end of this pdf file) about the relative humidity (RH) because it's not the same in each experiment and the humidity can also influence the SOA yields and chemical compounds. The effect of RH on aerosol formation has yet to be optimized. Some experiments for ïАą-pinene ozonolysis performed in flow reactor have shown that the aerosol yield was dependent on RH and other studies performed in simulation chamber have reported no effect. For example, the study of Saathoff [2009] on the ozonolysis of ïАą-pinene have shown that water had little influence on aerosol yields at room temperature, however, a

significant positive effect of water is observed for lower temperatures. Relative humidity is dependent on temperature.

Experimental Line 109-140: A new chamber is developed. It would be good to have more specifications of this chamber and the analytical tools. Which instrument allow to control and to measure the temperature and the humidity in the chamber? What is the range possible in the chamber? What is the uncertainties and the precision of each measurement? The temperature is homogeneous in all chamber?

Line 125: Add more precision on the PTRMS analysis: flow of sampling, time resolution, m/z range, E/N . . ..

Line 136: Add the uncertainties for each temperature

Line 137: Modify the sentence "For experiments with the same temperature setting, the RH settings were also similar (Tables S1 and S2). Âż There is more of 10% of difference for RH for a same temperature, it's not really similar. Add the range of humidity in the text for each temperature. Furthermore, the humidity is really variable between 26 and 55% for the three temperature.

Line 141-146: I would like to know if you have perform some blank filter and some blank of chamber. If yes, it would be good to add a sentence as Âń These control runs led to no detectable SOA production, hence confirming that no "memory effects" of the chamber were contaminating our results Âż.

Line 150: It would be good to have the parameters for the LC-MS (at least in Supporting Information). For example, what is the configuration of the ESI source in this analysis in negative mode.

Line 162: 362 products have been identified. I would like to know if some standard has been analyzed in LCMS to confirm the compounds in this study. If it's not the case, it would be good to say that the identification method was based on retention times and on mass spectra interpretation. Note that in absence of authentic standard, the

identification should be regarded as tentative. A recent article Kenseth et al, EST 2000, shows that the use of unrepresentative surrogates can lead to substantial systematic errors in quantitative LC/ESI-MS analyses of SOA.

Line 164-176: What type of seed (neutral, acidic?) is used to determine the aerosol wall loss in the chamber? In which conditions (RH, T) this experiment of aerosol wall loss has been performed? In SI, it's written that this experiment has been performed under an RH range of 17–23 %, but this range is different during the experiment (26-55%). Can this difference cause different wall losses? There are not only aerosol wall loss but there is also the wall-loss of oxidized organic vapors. I think it would be good to add some sentence on this wall loss even if no experiment would be performed. It's already mentioned in this article in line 250 but it's too late. To improve the article, I would correct the yield by all the wall losses.

Line 165: When a new Teflon bag is used? After a blank experiment?

Line 168-176: It would be good to model (i.e. Lai and Nazaroff) of the wall loss performed with seed. This well-known process depends on chamber geometry, static charge build-up on the walls, air flow and particle size. Since aerosol deposition was measured for (NH4)2SO4 seed with d = 1.77 g/cm3, due to the gravitational effect the wall loss rates for SOA (with different density) would be different. For more explanation, see the article of Lai and Nazaroff 2000.

Data analysis

Line 179-201: It would be good to correct also the SOA yield by the wall loss of oxidized organic vapors. Maybe give a range of yield with the particle wall loss correction for the smallest value and the vapor wall loss correction for the largest value.

Line 193: Âń $\alpha i$ is assumed to be temperature-independent Âż. Do you have a reference to assume that $\alpha i$ is assumed to be temperature-independent? If yes, it would be good to add it.

Result and discussion

Line 230 figure 1: It would be good to homogenize the axe of the figure, for example use the same nomenclature in the figure and also in the text. For example, in this figure it would be good to modify SOA ($\mu$g/m3) by mSOA ($\mu$g/m3). In figure 1, it would be good to add a banana plot for the aerosol. I think it would be a good idea to correct the data by also the wall loss of oxidized organic vapors.

Line 235 figure 2: It would be good to compare the yield corrected by all wall loss and not only by the particle wall loss. A comparison is performed with Pathak et al. 2007, it's good to do this comparison but it's difficult to understand why a comparison is performed with two curves (less of 10% RH and another with a range between 50 and 73%) because in our data the experiment are performed at 26% RH. I would like to know if no literature data exists for this range of humidity (26%). If yes, it would be good to compare the data with this reference. If not, it would be good to say that this data are new and it's difficult to compare with literature because no data have been performed at 298K and 26% RH in presence of seed.

Line 267: Thanks to add the uncertainties on the $\Delta$Hvap obtained in this study

Table 1: Add the humidity in the table and give the same number of significant number

Table 2: It would be good to compare the data on a graphs, the comparison would be more visual. Why the comparison with CMAQv4.7 model in the table? It would be good also to perform it.

Line 321-324: Âń Among the 362 compounds identified through LC-ToF-MS analysis in this study (Table S3), 331 compounds were ascribed to VBS bin ranges of $-8$–3. The other 31 compounds were ascribed to higher VBS bin ranges of 4–6, which is unrealistic for the compounds in the aerosol phase. Additionally, those latter compounds only accounted for an average of 12 % of the total mass of identified compounds. Thus, only the former 331 compounds are targeted in the following discussions. Âż I think

that the conclusion is very rapid. The VBS is not good consequently you remove 31 compounds. It would be good to add more comments on this compounds even if you decide to follow the discussion with 331 compounds. 31 compounds were ascribed to higher VBS bin ranges of 4–6, which is unrealistic for the compounds in the aerosol phase; it's not the first time this observation is performed. This observation strongly supports the resistance to diffusion imposed by the viscosity of the aerosol to the products formed inside the aerosol itself. As previously mentioned in the literature (Shiraiwa et al 2013) this observation indicates that volatility can be strongly minimized by viscosity and that relying only on gas-phase equilibria to represent partitioning can be misleading.

Line 332: How is obtained the total amount? By SMPS? It would be good to add a sentence to explain that.

Line 350, Figure 4: There is no axes for the yield in the graph B; it would be good to add it.

Line 448-449: Replace Âń M/z 223 was the most abundant OS identified at 298 k, followed by m/z 279, 281, 269, 283, 265, 253, 267, 251, 249, and 247. Âż by Âń m/z 223 was the most abundant OS identified at 298 K, followed by m/z 279, 281, 269, 283, 265, 253, 267, 251, 249, and 247. Âż

Line 500, Table 3: it's difficult to follow the table 3, it would be good to simplify to know what is really proposed by this study. Pleased find some title of some column of the new table : m/z $[M-H]-$; Formula $[M-H]-$; VBS bin; Tentative structure proposed in this study and Literature

SUPPORTING information

Table S1 Remove the C near O3 in table S1. I would add if it's possible a column with $\Delta\alpha$–pinene. It would be good to add the uncertainties in the table. For the yield, it would be good to correct it by all the wall losses. Replace also SOA by mSOA

Table S2 I would add if it's possible a column with $\Delta\alpha$-pinene in the table S2. It would be good to add the uncertainties in the table. For the yield, it would be good to correct it by all the wall losses. Replace also SOA by mSOA

Figure S2: It would be good to model these experiment of wall loss and add the model in this figure

REFERENCES

Bonn, B., G. Schuster, and G.K. Moortgat, Influence of water vapor on the process of new particle formation during monoterpene ozonolysis. Journal of Physical Chemistry A, 2002. 106(12): p. 2869-2881. Cocker, D.R., et al., The effect of water on gas-particle partitioning of secondary organic aerosol. Part I: alpha-pinene/ozone system. Atmospheric Environment, 2001. 35(35): p. 6049-6072. Glasius, M., et al., Carboxylic acids in secondary aerosols from oxidation of cyclic monoterpenes by ozone. Environmental Science & Technology, 2000. 34(6): p. 1001-1010. Hakola, H., et al., Product Formation from the Gas-Phase Reactions of Oh Radicals and O3 with a Series of Monoterpenes. Journal of Atmospheric Chemistry, 1994. 18(1): p. 75-102. Jonsson, A.M., M. Hallquist, and E. Ljungstrom, Impact of humidity on the ozone initiated oxidation of limonene, Delta(3)-carene, and alpha-pinene. Environmental Science & Technology, 2006. 40(1): p. 188-194. Jonsson, A.M., M. Hallquist, and E. Ljungstrom, The effect of temperature and water on secondary organic aerosol formation from ozonolysis of limonene, Delta(3)-carene and alpha-pinene. Atmospheric Chemistry and Physics, 2008. 8(21): p. 6541-6549. Jonsson, A.M., M. Hallquist, and H. Saathoff, Volatility of secondary organic aerosols from the ozone initiated oxidation of alpha-pinene and limonene. Journal of Aerosol Science, 2007. 38(8): p. 843-852. Kenseth Christopher M., Nicholas J. Hafeman, Yuanlong Huang, Nathan F. Dalleska, Brian M. Stoltz, and John H. Seinfeld Synthesis of Carboxylic Acid and Dimer Ester Surrogates to Constrain the Abundance and Distribution of Molecular Products in $\alpha$-Pinene and $\beta$-Pinene Secondary Organic Aerosol Environmental Science & Technology Article ASAP DOI: 10.1021/acs.est.0c01566 Lai K., A. C.; Nazaroff, W.

W. Modeling indoor particle deposition from turbulent flow onto smooth surfaces. J. Aerosol Sci. 2000, 31 (4), 463−476. Shiraiwa, M.; Zuend, A.; Bertram, A. K.; Seinfeld, J. H., Gas-particle partitioning of atmospheric aerosols: interplay of physical state, non-ideal mixing and morphology. Physical Chemistry Chemical Physics 2013, 15 (27), 11441-11453. Winterhalter, R., et al., LC-MS analysis of aerosol particles from the oxidation of $\alpha$-pinene by ozone and OH radicals. Atmospheric Chemistry and Physics Discussions, 2003. 3: p. 1-39. Yu, J.Z., et al., Gas-phase ozone oxidation of monoterpenes: Gaseous and particulate products. Journal of Atmospheric Chemistry, 1999. 34(2): p. 207-258. Yu, Y., et al., Nitrate ion photochemistry at interfaces: a new mechanism for oxidation of alpha-pinene. Physical Chemistry Chemical Physics, 2008. 10(21): p. 3063-3071.

Please also note the supplement to this comment:
https://acp.copernicus.org/preprints/acp-2020-798/acp-2020-798-RC1-supplement.pdf

---

## Referee Comment (RC2) · Anonymous Referee #2 · 28 Oct 2020

**Review of the manuscript acp-2020-798 with title: "Temperature and acidity dependence of secondary organic aerosol formation from α-pinene ozonolysis with a compact chamber system" by Deng et al.**

**General comments:**

This manuscript describes a new atmospheric simulation chamber and its application to study the secondary organic aerosol from the oxidation of alpha-pinene by ozone at different temperatures and for seed aerosol particles of different acidity. Therefore, it fits well in the scope of the journal of atmospheric chemistry and physics and focusses on an interesting scientific topic which has been subject of many previous studies. The scientific results are presented in a clear and well-structured way and are based on a reasonable scientific approach and valid methods. However, this manuscript represents only a limited contribution to scientific progress in this field.

The description of the new simulation chamber is quite limited and rewards a more detailed discussion of its capabilities and limitations compared to other simulation chambers as well as how suitable it is for this kind of studies. Especially, the impact of the chamber limitations on the uncertainties for the major scientific results should be addressed and quantified.

The scientific results lack a reasonable representation of their uncertainties as well as a suitable comparison with data from the large number of previous studies on this subject. This is especially the case for the yields and VBS distributions. Some of the most interesting findings in this study are the analysis and identification of SOA compounds for which also potential formation mechanisms are discussed. This discussion may be restructured and focused on new findings.

Most of the scientific results presented don't go much beyond current knowledge and some of them can be expected to have higher uncertainties than those of previous studies. Therefore, the manuscript contains only a limited contribution to the scientific understanding of formation and composition of secondary organic aerosol in our atmosphere. Hence, this manuscript should only be accepted for publication after major improvements and focusing on new scientific findings.

**Specific comments**

Try to include all major results in the abstract.

Page 1 line 10: Actually, you did not describe the development of a chamber in this study. Please reformulate.

Page 1 line 25: SOA is not only formed by photo oxidation. Please reformulate.

Page 1 line 34: Please use "saturation concentrations".

Page 1 line 35: Notify the importance of ELVOCs and that ELVOCs and LVOCs are also formed in the gas phase (see e.g. Tröstl et al., Nature, 2016).

Page 2 line 50-54: Reformulate to shorter sentences.

Page 2 line 54: Do you mean contradictory results?

Page 3 line 89-93: You may cite Brüggemann et al., EST, 54, 3767, 2020 here as well.

Page 3 line 112: It should read: "at atmospheric pressure".

Page 3 line 114-115: Explain how the pure air is humidified (bubbling through water?) and how this affects the purity of the G3 pure air regarding gases (e.g. VOC) and small particles.

Page 3 line 131: Explain how ozone was added to the chamber and if you can estimate a mixing time for the reactants. How is mixing achieved in your chamber?

Page 4 line 134: Here it should read: "aerosol particles". Generally, distinguish between aerosols = airborne particles and gases, aerosol particles = airborne particles, and gases throughout the manuscript.

Page 4 line 139-140: Explain the criteria for achieving a sufficient cleanliness of your chamber by flushing.

Page 5 line 161: It should read: "saturation concentrations".

Page 5 line 161: It should read: "molecular formulae" while chemical formulae would already contain some structural information. It would be very useful if you would add the names of those compounds identified in table S3. Explain how the C* values have been calculated.

Page 5 line 169-170: Coagulation is not a wall loss. Please rephrase. Specify if you have any indications of electrostatic particle losses in your chamber especially for new Teflon foil.

Page 5 line 169-172: Please fit the particle losses in your chamber e.g. using the formulation given by Lai and Nazaroff, J. Aerosol Sci., 31, 436, 2000. Discuss how the wall loss parameters of your chamber compare to others and how suitable your chamber is for different kind of applications. Did you also measure the wall losses for solid particles?

Page 5 line 173: How can the seed particle size affect wall losses like sedimentation for super 200 nm particles?

Page 5 line 176: It should read: "larger particles".

Page 5 line 182: Please explain how you separated the SOA mass from the seed particle mass and what uncertainty this means for the determination of the SOA mass.

Page 5 line 182-186: Please discuss the uncertainties caused by the different approximations applied in this section. Did you take direct losses of semi volatile gases to the wall into account (Zhang et al., PNAS, 2014)?

Page 6 line 204: It should read "….was calculated and then ascribed…".

Page 6 line 219: I think it would be helpful to add the evolution of measured particle number size distribution e.g. as a contour plot in Figure 1 as well as the evolution of the particle number.

Page 6 line 224: Add the uncertainty of the SOA yield. It should read

Page 6 line 224-225: It should read: "As the corrected SOA particle concentration was constant after 50 min,…".

Page 6 line 226: It should read: "…concentration of SOA particle mass at 90 minutes would be underestimated…".

Page 8 line 243: It should read: "…seven experiments with different initial $\alpha$-pinene concentrations (54–323 ppbv) at 298 K and neutral seeds were conducted at a relative humidity (RH) of approximately 26–27 % (Table S1).".

Page 8 line 244/Table S1: Indicate how much aerosol volume or mass was formed after adding ozone to your chamber but before adding pinene.

Page 8 line 248: Add the error bars for your data in Figure 2 and enlarge the symbols. Discuss if the uncertainties are larger for the lower mass loadings. Note that you have only 5 minutes time resolution of the SMPS data. Please change the caption indicating that the lines represent a parameterization by Pathak et al. 2007a and not measured yields. Discuss how representative a comparison to only the data of Pathak et al. is considering the large amount of data available in the literature as well as more recent studies.

Page 8 line 250: Do you mean: "…used in this study was significantly larger than for most previous studies…"?

Page 8 line 258-261: Given the methods you applied to correct for potential artifacts what uncertainty remains for SOA yields for different conditions and what are e.g. resulting concentration limit for your chamber.

Page 8 line 266-267: Give the uncertainties for these parameters.

Page 8 line 266-267: Do you mean: "…lowering of their vapor pressures."?

Page 8 line 277: It should read: "…lower than those of Saha et al….".

Page 9 line 285: Indicate if the dependence of SOA yields on the acidity of seed particles is significant or not.

Page 9 line 299: Indicate if the SOA yields were significantly enhanced comparing acidic to neutral seeds or not.

Page 9 line 306-308: Wich measurement accuracy would be needed to achieve significant results?

Page 9 line 310: Add uncertainties to your data points in Figure 3 and compare to literature data.

Page 10 line 316: Add the uncertainties to the VBS parameters in Table 1 and compare to literature values.

Page 10 line 325: It should read: "…measured intensities of particle phase compounds…".

Page 10 line 331: It should read: "The intensities of both the particle and gas phases were…".

Page 10 line 334: Give how much the rate coefficients varies with temperature. Please cite original references like Tillmann et al., PCCP, 11, 2323, 2009.

Page 10 line 335-336: How valid is this assumption as the total amount of SVOCs may be influenced by temperature dependent wall losses, changing product branching ratios, etc..

Page 11 Figure 4: Replace Aerosol by Particle on the y-axis label and caption. Indicate the uncertainty of your data.

Page 12 line 380: It should read: "…were concentrated in the O:C ratio range…"

Page 13: I think it would be useful to add a table of major products, their abundances and their tentative chemical identification (compound, structure) either separately or in table S3.

Page 13 line 429: Please explain the unspecific chromatograms in Figure S5.

Page 14 line 433: Can you give an estimate of the uncertainties for the OS intensities?

Page 14 Figure 7: Add uncertainties and discuss which differences or trends are significant.

Page 14 line 448-449: For the mechanistic discussion it would be more useful to give here and in the following either the compound name or molecular formulae for each compound identified. State the uncertainties of the molecular yields. Consider adding potential reactions mechanisms e.g. in the supplement.

Page 14 line 450-451: Consider revising: "The possible formation mechanisms of the OS compounds were proposed based on literature data combined with the experimental results of this study."

Page 15 line 461: It should read: "While all eleven OS compounds detected in this study are formed from α-pinene ozonolysis due to the presence of an excess of the OH radical scavenger,…"

Page 15 line 467: It should read: "…and one (m/z 265) was said to be from the sulfacation of pinonaldehyde…".

Page 15 line 485: "…could only be observed at the lower temperatures."

Page 15 line 488-489: Consider revising: "When the structure of the OS can be determined and the precursor compound can be assumed, the formation mechanism of the OS in α-pinene ozonolysis with acidic seeds may be confirmed."

Page 16 Figure 8: Indicate to what the yield is related to. You may add total OS yield vs. temperature which will be positive and dominated by $C_7H_{11}O_6S$ and $C_{10}H_{15}O_7S$.

Page 17 Table 3: Try to enlarge the structures as much as possible as well as the letters indicating the various references.

Page 18 line 508: Consider revising: "…with acidic and neutral seed aerosol particles.".

Page 18 line 513: Consider revising: "Among the 362 compounds identified,…".

Page 18: In section 5 you should go beyond a summary and add conclusions for our understanding of atmospheric SOA and OS. You may also discuss the relevance of your findings vs. existing literature data.

Tables S1 and S2: Why is there a letter c at the head of the column with ozone concentrations? Add uncertainties for your data. Give only significant digits.

Table S3: Please add compound names were possible and mention that C* values were calculated according to Li et al., 2016.

Figure S3: Add a legend.

Figure S4: Are the differences between the yield parameterizations significant? Is it correct that the temperature dependence of the yields is more significant than the acidity dependence?

Figure S5: How do you explain the rather unspecific chromatograms e.g. for m/z 249, 253, 267, 283 ?

**References:**

Brüggemann, M., Xu, R., Tilgner, A., Kai Kwong, C., Mutzel, A., Poon, H.Y., Otto, T., Schaefer, T., Poulain, L., Chan, M.N., and Herrmann, H., Organosulfates in Ambient Aerosol: State of Knowledge and Future Research Directions on Formation, Abundance, Fate, and Importance, *Environmental Science & Technology 54* (7), 3767-3782 (2020). DOI: 10.1021/acs.est.9b06751

Lai A.C.K., and Nazaroff, W.W. , MODELING INDOOR PARTICLE DEPOSITION FROMTURBULENT FLOW ONTO SMOOTH SURFACES, *J. Aerosol Sci*., 31, 463-467 (2000) doi.org/10.1016/S0021-8502(99)00536-4

Tillmann, R., Saathoff, H., Brauers, T., Kiendler-Scharr, A., Mentel, T.F., Temperature dependence of the rate coefficient for the α-pinene reaction with ozone in the range between 243 K and 303 K, *Phys. Chem. Chem. Phys.,* 11, 2323-2328 (2009) DOI: 10.1039/B813407C

Tröstl, J., Chuang, W., Gordon, H. *et al.* The role of low-volatility organic compounds in initial particle growth in the atmosphere. *Nature* 533**,** 527–531 (2016). doi.org/10.1038/nature18271

---

## Author Comment (AC1) · 25 Feb 2021

**Response to Anonymous Referee #1**

We appreciate the invaluable comments. Our answers to the comments are provided below. The reviewer comments are written in italic.

*General remarks:*

*In this paper, Temperature and acidity dependence of secondary organic aerosol formation from α-pinene ozonolysis with a compact chamber system have been investigated. This is a nice piece of work where the temperature and acidity dependence of SOA yields and chemical components were investigated. In addition, the formation of organosulfates (OS) was observed under acidic seed conditions. The data presented are of significant importance; however some modifications would be good to improve the article. The article should be published after major modifications.*

*Some characteristics of the new chamber are missing. We have developed a new chamber, it would be good to have more specifications. The temperature effect on the SOA formation has been relatively little studied, mainly due to the low number of temperature controlled chamber. Give more precision on the temperature controlled chamber.*

*There is no uncertainties, it would be good to add it in the text, in the figure and in the table. For example, no uncertainties has given for the temperature, the yield, the concentration ….*

*It would be good also to correct the SOA yield obtained by all wall losses (particle and also oxidized organic vapors) to improve the article.*

*It would be good to add also some comments and reference (list of some references given at the end of this pdf file) about the relative humidity (RH) because it's not the same in each experiment and the humidity can also influence the SOA yields and chemical compounds. The effect of RH on aerosol formation has yet to be optimized. Some experiments performed in flow reactor have shown that the aerosol yield was dependent on RH and other studies performed in simulation chamber have reported no effect. For example, the study of Saathoff [2009] on the ozonolysis of α-pinene have shown that water had little influence on aerosol yields at room temperature, however, a significant positive effect of water is observed for lower temperatures. Relative humidity is dependent on temperature.*

*For more explanation, see specific comments.*

**Reply>** We respond to the general comments within the following responses to the specific comments.

*Specific comments:*

*Introduction*

*Line 41-59: The role of acidity is really good explain. However, the role of the other parameters as the temperature is very succinct, just one sentence. It would be good to explain more the influence of the other parameters on the SOA yields and chemical composition.*

**Reply>** Examples of the temperature-dependence of SOA formation have been added to the revised manuscript as follows: "For example, the relative importance of the temperature-dependencies of the volatilities of oxidation products and of the gas-phase and multiphase chemical reactions, which could result in different temperature-dependence of SOA yields (Pathak et al., 2007b; von Hessberg et al., 2009), are still not well constrained." (Page 2 lines 59–62)

*Line 63-81: The dependence of humidity is not explained for the specific case of α-pinene ozonolysis. It would be good to add also some sentences and references (list of some references given at the end of this pdf file) about the relative humidity (RH) because it's not the same in each experiment and the humidity can also influence the SOA yields and chemical compounds. The effect of RH on aerosol formation has yet to be optimized. Some experiments for α-pinene ozonolysis performed in flow reactor have shown that the aerosol yield was dependent on RH and other studies performed in simulation chamber have reported no effect. For example, the study of Saathoff [2009] on the ozonolysis of α-pinene have shown that water had little influence on aerosol yields at room temperature, however, a significant positive effect of water is observed for lower temperatures. Relative humidity is dependent on temperature.*

**Reply>** We agree with the reviewer. The SOA yields and chemical compounds can be influenced by the humidity in the chamber. However, we did not investigate the influence of humidity on SOA yield in the present study. Since very dry conditions are not realistic in ambient air, we carried out the experiments at medium humidity (~26–27 % RH at 298 K, ~32–34 % RH at 288 K, and ~45–55 % RH at 278 K), which did not vary considerably at each temperature. We believe that the differences in RH among our experiments would not influence SOA formation significantly, as explained in the following section. First, nucleation was negligible in all the experiments because of the high concentrations of seed particles applied; therefore, the influence of RH on SOA formation would be reflected in the particle phase (Kristensen et al., 2014). Second, because the seed particles were dried into effloresced states before being introduced into the chamber, all particles in the chamber were in solid (neutral seed conditions) or near solid state (acidic seed conditions) (Tang and Munkelwitz, 1977). Consequently, the influence of water on the particle phase through physical partitioning or chemical reactions should be minor (Faust et al., 2017). We have mentioned this in the revised manuscript as follows:

"In the present study, we did not investigate the influence of humidity on SOA yield. Since very dry conditions are not realistic in ambient air, we carried out the experiments at medium humidity (26–55 % RH, Tables S1 and S2). The differences in RH among experiments in the present study would not influence SOA formation significantly, as explained in the following section. First, nucleation would be negligible in all experiments because of the high concentrations of seed particles applied. Consequently, the influence of RH on SOA formation would be reflected in the particle phase (Kristensen et al., 2014). In addition, because the seed particles were dried into effloresced states before being introduced into the chamber, all particles would be in solid (neutral seed conditions) or near solid states (acidic seed conditions) (Tang and Munkelwitz, 1977). Therefore, the influence of water on the particle phase through physical partitioning or chemical reactions would be minor (Faust et al., 2017)." (Pages 4–5 lines 162–170).

*Experimental*

*Line 109-140: A new chamber is developed. It would be good to have more specifications of this chamber and the analytical tools. Which instrument allow to control and to measure the temperature and the humidity in the chamber? What is the range possible in the chamber? What is the uncertainties and the precision of each measurement? The temperature is homogeneous in all chamber?*

**Reply>** The RH of the chamber air was not measured during but after the experimental run by pumping the remaining chamber air into a separate small TEFLON bag, to which a VAISALA RH&T probe (model, HMP76) equipped with a measurement indicator (model, M170) was attached. In experimental runs with the same temperature setting, similar flow rates and filling times were applied for both humid and dry G3 air, the RH was only measured for 4, 3, and 4 of the experimental runs for temperature settings of 298, 288, and 278 K, respectively. The measured RH ranges are presented in Tables S1, S2, and Table 1. The measurement of the RH has been included in the revised manuscript as follows: "The RH of the chamber air was measured after the experimental run by pumping the remaining chamber air into a separate small TEFLON bag, to which a VAISALA RH&T probe (model, HMP76) equipped with a measurement indicator (model, M170) was attached.". (Page 4 lines 123–125)

The temperature inside the cabinet was measured using a thermocouple attached to the inside of the cabinet (T3 in Fig. S1). The potential temperature range in the cabinet is 5–40 ℃. The evaluation of the thermostat capacity of the cabinet is presented in Text S1 and Fig. S2. Under initial temperature settings of 4.6, 14.6, and 24.7 °C, the temperature inside the cabinet within an hour varied within the 3.9–5.5, 14.0–15.3, and 24.6–25.3 °C ranges, respectively, and the standard deviations are 0.36, 0.27, and 0.18 °C, respectively. The temperature ranges inside the Teflon chamber within one hour were 4.8–5.2, 14.8–15.1, and 24.9–25.2 °C, respectively, and the standard deviations were 0.15, 0.10, and 0.11 °C, respectively. Besides, we waited for 30 min before the start of chemical reactions. Therefore, we think that the temperature was well controlled (uncertainties were < 1 °C) during the SOA formation reactions. This has been added to the revised manuscript, as follows:

"The temperature inside the cabinet was measured using a thermocouple attached to the inside of the cabinet (T3 in Fig. S1). The achievable operating temperature range of the chamber was 5–40 ℃. A detailed evaluation of the thermostat capacity of the chamber under dark conditions is presented in Text S1, which indicates that the temperature inside the chamber was well controlled (varied within ±1 ℃).". (Page 3 lines 115–119)

*Line 125: Add more precision on the PTRMS analysis: flow of sampling, time resolution, m/z range, E/N ….*

**Reply>** The following details have been included in the revised manuscript: "The PTR-MS was operated at a flow rate of approximately 250 $cm^3$ $s^{-1}$ under a field strength (E/N, where E is the electric field strength (V $cm^{-1}$) and N is the buffer gas number density (molecule $cm^{-3}$) of the drift tube) of 106 Td. The length of the drift tube was 9.2 cm. The drift voltage was set to 400 V. The temperatures of the inlet and drift tubes were set to 105°C and the pressure at the drift tube was set to 2.1 mbar. The signal intensities of ions with m/zs of 21, 30, 32, 37, 45, 46, 75, 81, and 137 were recorded approximately every 4.5 sec. The detection sensitivity of α-pinene was 3.3±0.6 ncps $ppbv^{-1}$ (ncps means normalized counts per second to $10^6$ cps of $H_3O^+$).". (Page 4 lines 138–143)

*Line 136: Add the uncertainties for each temperature*

**Reply>** As stated previously, evaluation of the thermostat capacity presented in Text S1 indicates that the temperature inside the chamber was well controlled with uncertainties < 1 °C. This point is presented in the revised manuscript as follows: "A detailed evaluation of the thermostat capacity of the chamber under dark conditions is presented in Text S1, which indicates that the temperature inside the chamber was well controlled (varied within ±1 °C)." (Page 3 lines 117–119). Hereafter, we don't show the uncertainties of the temperature in the text.

*Line 137: Modify the sentence "For experiments with the same temperature setting, the RH settings were also similar (Tables S1 and S2). » There is more of 10% of difference for RH for a same temperature, it's not really similar. Add the range of humidity in the text for each temperature. Furthermore, the humidity is really variable between 26 and 55% for the three temperature.*

**Reply>** The sentence "For experiments with the same temperature setting, the RH settings were also similar" has been deleted in the revised manuscript. Instead, the RH range is displayed using the following sentence "In the present study, we did not investigate the influence of humidity on SOA yield. Since very dry conditions are not realistic in ambient air, we carried out the experiments at medium humidity (26–55 % RH, Tables S1 and S2)." (Pages 4–5 lines 162–164)

As has been explained earlier, we think the differences in RH among the experiments do not influence SOA formation in our study considerably. We have added the following explanations in the revised manuscript: "The differences in RH among experiments in the present study would not influence SOA formation significantly, as explained in the following section. First, nucleation would be negligible in all experiments because of the high concentrations of seed particles applied. Consequently, the influence of RH on SOA formation would be reflected in the particle phase (Kristensen et al., 2014). In addition, because the seed particles were dried into effloresced states before being introduced into the chamber, all particles would be in solid (neutral seed conditions) or near solid states (acidic seed conditions) (Tang and Munkelwitz, 1977). Therefore, the influence of water on the particle phase through physical partitioning or chemical reactions would be minor (Faust et al., 2017)." (Page 5 lines 164–170).

*Line 141-146: I would like to know if you have perform some blank filter and some blank of chamber. If yes, it would be good to add a sentence as « These control runs led to no detectable SOA production, hence confirming that no "memory effects" of the chamber were contaminating our results ».*

**Reply>** Information concerning blank filter and blank of chamber has been included in the manuscript and Text S2.

With regard to the chamber blank, a sentence "Particle number concentration and VOC mixing ratio measurements indicate that the chamber background concentrations of particles and VOCs were negligible, and no extra contamination was observed by the humidification process of the G3 air (Text S2)." has been added to the revised manuscript (Page 4 lines 125–127).

About the filter blank, a sentence "A blank filter was also analyzed using a procedure similar to that of the sample filters. The results confirmed no substantial contamination in the filter and the filter analysis procedure (Text S2)." has been added to the revised manuscript (Page 5 lines 176–177).

*Line 150: It would be good to have the parameters for the LC-MS (at least in Supporting Information). For example, what is the configuration of the ESI source in this analysis in negative mode.*

**Reply>** The relative configuration parameters have been included in the revised manuscript as follows:

"The key configuration parameter settings were as follows: nebulizer pressure was 0.21 MPa; the voltage in the spray chamber was –3500 V; the drying nitrogen gas temperature was 325°C and flow rate was 5 L min$^{-1}$; and the fragmentor voltage was 175 V." (Page 5 lines 185–187).

*Line 162: 362 products have been identified. I would like to know if some standard has been analyzed in LCMS to confirm the compounds in this study. If it's not the case, it would be good to say that the identification method was based on retention times and on mass spectra interpretation. Note that in absence of authentic standard, the identification should be regarded as tentative. A recent article Kenseth et al, EST 2020, shows that the use of unrepresentative surrogates can lead to substantial systematic errors in quantitative LC/ESI-MS analyses of SOA.*

**Reply>** The original sentence "We determined the chemical formulae and signal intensities for 362 products (including eleven organosulfates) of different m/z (Table S3)" has been changed to "We tentatively determined the molecular formulae and signal intensities of 362 products (including 11 organosulfates) with different m/z (Table S3) based on retention times and interpretation of mass spectra." (Page 5 lines 198–200).

*Line 164-176: What type of seed (neutral, acidic?) is used to determine the aerosol wall loss in the chamber? In which conditions (RH, T) this experiment of aerosol wall loss has been performed? In SI, it's written that this experiment has been performed under an RH range of 17–23 %, but this range is different during the experiment (26-55%). Can this difference cause different wall losses?*

**Reply>** Fig. S2 (currently Fig. S4) is for checking the size-dependent wall-loss rate. However, the bulk wall-loss rate has been measured under different seed particle and temperature conditions. As was indicated in Sect. 2.3 "The measurements were carried out whenever a new Teflon bag was used or the experimental conditions (i.e., temperature or seed particle acidity) were changed under humid air conditions. The latest measured bulk wall-loss rate (Sect. 3.1) was applied for each SOA formation experiment." (Currently Page 6 lines 205–207).

*There are not only aerosol wall loss but there is also the wall-loss of oxidized organic vapors. I think it would be good to add some sentence on this wall loss even if no experiment would be performed. It's already mentioned in this article in line 250 but it's too late. To improve the article, I would correct the yield by all the wall losses.*

**Reply>** A qualitative discussion of the influence of gas-phase wall-loss on SOA yield has been included in Sect. 4.1. In addition, an associated introduction has been included at the end of Sect. 2.3, as follows: "Wall-loss of gas-phase organic compounds in the Teflon chamber could also cause the underestimation of SOA yields (Zhang et al., 2014; Krechmer et al., 2016). Although not experimentally determined in the present study, the influence of gas-phase wall-loss on SOA yields will be discussed based on the studies of Zhang et al. (2014) and Krechmer et al. (2016) in Sect. 4.1." (Page 6 lines 221–223).

*Line 165: When a new Teflon bag is used? After a blank experiment?*

**Reply>** A new Teflon bag was used whenever the old Teflon bag had a leak problem.

*Line 168-176: It would be good to model (i.e. Lai and Nazaroff) of the wall loss performed with seed. This well-known process depends on chamber geometry, static charge build-up on the walls, air flow and particle size. Since aerosol deposition was measured for $(NH_4)_2SO_4$ seed with d = 1.77 g/cm3, due to the gravitational effect the wall loss rates for SOA (with different density) would be different. For more explanation, see the article of Lai and Nazaroff 2000.*

**Reply>** Simulations of wall-loss performed with seed particles have been performed and presented in Text S3 and referred to in Sect. 2.3 as: "Model simulation (Text S3) and literature survey results revealed that the high wall-loss rates of sub-100 nm particles were mainly caused by particle coagulation (Nah et al., 2017; Wang et al., 2018a) and those of super-200 nm particles were likely the result of turbulent deposition (Lai and Nazaroff, 2000)." (Page 6 lines 213–216).

The models of Lai and Nazaroff (2000), which focused on turbulent deposition, ignoring Brownian diffusion, but considering the gravitational settling deposition, and that of Hinds (1999), which considered only Brownian diffusivity deposition or settling deposition, were applied. The simulation results (Fig. S4) revealed that pure gravitational settling deposition played a minor role in the total observed particle loss. The influence of Brownian diffusivity deposition was also minimal after 30 min of stabilization. Whereas the large wall-loss of super-200 nm particles can be explained by the turbulent deposition, that of sub-100 nm particles cannot be explained by any of the simulated mechanisms. According to the study of Wang et al. (2018a), the changes in particle number-size distributions caused by coagulation could be the major reason for the large apparent wall-loss of sub-100 nm particles. In addition, electrostatic particle losses were small even for a new Teflon bag in our experiment, because the wall-loss of a new bag was often lower than that of an old bag. Besides, humid air was applied during the experiment, which may prevent electrostatic particle losses.

*Data analysis*

*Line 179-201: It would be good to correct also the SOA yield by the wall loss of oxidized organic vapors. Maybe give a range of yield with the particle wall loss correction for the smallest value and the vapor wall loss correction for the largest value.*

**Reply>** The influence of the gas-phase wall-loss on SOA yield is qualitatively discussed in Sect. 4.1. A sentence indicating this point has been included in this section in the revised manuscript, as follows: "The influence of gas-phase wall-loss on SOA yield is discussed qualitatively in Sect. 4.1." (Page 6 lines 238).

*Line 193: « αi is assumed to be temperature-independent ». Do you have a reference to assume that αi is assumed to be temperature-independent? If yes, it would be good to add it.*

**Reply>** Yes. A reference (Pathak et al., 2007a) has been added (Page 7 line 244).

*Result and discussion*

*Line 230 figure 1: It would be good to homogenize the axe of the figure, for example use the same nomenclature in the figure and also in the text. For example, in this figure it would be good to modify SOA (µg/m3) by mSOA (µg/m3). In figure 1, it would be good to add a banana plot for the aerosol. I think it would be a good idea to correct the data by also the wall loss of oxidized organic vapors.*

**Reply>** The name of the left axis in panel (a) has been changed from "Concentrations of α-pinene, $O_3$ (ppbv)" to "α-Pinene, $O_3$ (ppbv)", and the name of the right axis has been changed from "SOA ($\mu g\ m^{-3}$)" to "$m_{SOA}$ ($\mu g\ m^{-3}$)". A banana plot for the aerosol has been included as a supplementary figure (Fig. S6). A relevant explanation has been included in the manuscript: "Evolution of the particle number-size distribution is presented in Fig. S6." (Page 8 lines 291–292).

Qualitative discussions of the influence of the wall loss of oxidized organic vapors based on the results of Fig. 2 have been included in the revised manuscript as follows: "The SOA yields in this study were 25–60 % lower than those of Pathak et al. (2007a). Possible reasons may include lack of consideration of the wall-loss of oxidized organic vapors because the surface to volume ratio of the chamber used in this study (7.1 $m^{-1}$) was much larger than those of previous studies (<3 $m^{-1}$; Pathak et al., 2007a and references therein). According to Zhang et al. (2014), the vapor wall-loss bias factor, $R_{wall}$ (defined as the ratio of the SOA mass when the vapor wall loss is assumed to be zero, to the SOA mass when the optimal vapor wall loss rate is applied), was reported to be ~4 at the initial seed surface area of ~2 × $10^3$ $\mu m^2$ $cm^{-3}$ (seed-to-chamber surface area ratio ~1 × $10^{-3}$) and ~2 at the initial seed surface area of >6 × $10^3$ $\mu m^2$ $cm^{-3}$ (seed-to-chamber surface area ratio >3 × $10^{-3}$) during the photooxidation of toluene. As the initial seed surface area in the present study was in the (1–3) × $10^3$ $\mu m^2$ $cm^{-3}$ range (seed-to-chamber surface area ratio (1–4) × $10^{-4}$), $R_{wall}$ in the Teflon bag might be at least twice those of the large chambers. This leads to the underestimation of the SOA yield of 50 % when compared with the values obtained from the large chambers. A low limit correction of the gas-phase wall-loss influence for Exp. No. 2 and other experimental runs in which chemical composition analysis had been conducted based on the method of Krechmer et al. (2016) (Text S5, Fig. 2), which confirmed that gas-phase wall-loss is one reason for the lower SOA yields in the present study compared with Pathak et al. (2007a) and other previous studies presented in Fig. 2." (Pages 9–10 lines 324–337).

*Line 235 figure 2: It would be good to compare the yield corrected by all wall loss and not only by the particle wall loss. A comparison is performed with Pathak et al. 2007, it's good to do this comparison but*

*it's difficult to understand why a comparison is performed with two curves (less of 10% RH and another with a range between 50 and 73%) because in our data the experiment are performed at 26% RH. I would like to know if no literature data exists for this range of humidity (26%). If yes, it would be good to compare the data with this reference. If not, it would be good to say that this data are new and it's difficult to compare with literature because no data have been performed at 298K and 26% RH in presence of seed.*

**Reply>** As mentioned above, we have estimated the amount of the wall-loss of oxidized organic vapors for Exp. No. 2 and the result is shown in Fig. 2. In the figure, we have added some yield data, which were obtained under different RH conditions. As has been explained previously (Pages 4–5 lines 162–170), we think that the differences in RH among the experiments do not influence the SOA yields substantially, and that the present data are categorized in the data with medium RH conditions.

*Line 267: Thanks to add the uncertainties on the ΔHvap obtained in this study*

**Reply>** Estimating the uncertainties of the VBS parameters arithmetically is challenging. Instead, obtaining sensitivity analyses of the four-product VBS fitting curves by changing $\Delta H_{vap}$ while fixing the stoichiometric yields $\alpha_i$ revealed that the effective $\Delta H_{vap}$ could be in the 0 to 70 and 0 to 80 kJ mol$^{-1}$ ranges under neutral and acidic seed conditions, respectively. This has been stated in the revised manuscript, as follows: "Notably, sensitivity analyses achieved by fixing the stoichiometric yields $\alpha_i$ while changing $\Delta H_{vap}$ and comparing the resulting VBS curves with measured data (Fig. S8) indicated that the effective $\Delta H_{vap}$ could be in the 0 to 70 and 0 to 80 kJ mol$^{-1}$ ranges for neutral and acidic seed conditions, respectively." (Page 10 lines 366–368).

*Table 1: Add the humidity in the table and give the same number of significant number*

**Reply>** The RH has been included in Table 1. The significant number of $\alpha_i$ has been unified.

*Table 2: It would be good to compare the data on a graphs, the comparison would be more visual. Why the comparison with CMAQv4.7 model in the table? It would be good also to perform it.*

**Reply>** Information on the CMAQv4.7 SOA module (Carlton et al., 2010) has been included in Table 2. A supplementary figure (Fig. S7) has been illustrated for comparison of $\Delta H_{vap}$ in the present study with those in previous studies based on $\Delta H_{vap}$ versus $C^*$ plots.

*Line 321-324: « Among the 362 compounds identified through LC-ToF-MS analysis in this study (Table S3), 331 compounds were ascribed to VBS bin ranges of −8–3. The other 31 compounds were ascribed to higher VBS bin ranges of 4–6, which is unrealistic for the compounds in the aerosol phase. Additionally, those latter compounds only accounted for an average of 12 % of the total mass of identified compounds. Thus, only the former 331 compounds are targeted in the following discussions. » I think that the conclusion is very rapid. The VBS is not good consequently you remove 31 compounds. It would be good to add more comments on this compounds even if you decide to follow the discussion with 331 compounds. 31 compounds were ascribed to higher VBS bin ranges of 4–6, which is unrealistic for the compounds in the*

*aerosol phase; it's not the first time this observation is performed. This observation strongly supports the resistance to diffusion imposed by the viscosity of the aerosol to the products formed inside the aerosol itself. As previously mentioned in the literature (Shiraiwa et al 2013) this observation indicates that volatility can be strongly minimized by viscosity and that relying only on gas-phase equilibria to represent partitioning can be misleading.*

**Reply>** We agree with the reviewer's suggestion that high-volatility compounds produced inside the aerosol particles might have been wrapped into the particle phase because of the high viscosity of the BSOA. We further note that only less than a half of the compounds in VBS bin ranges greater than four are in the particle phase (Donahue et al., 2006), and that larger compounds might have been fragmented into small molecules during the ESI MS measurements (Fig. S10).

Accordingly, the sentence ", which is unrealistic for the compounds in the aerosol phase. Additionally, those latter compounds only accounted for an average of 12 % of the total mass of identified compounds. Thus, only the former 331 compounds are targeted in the following discussions." has been modified to: ". Only the former 331 compounds are targeted in the following discussions for two reasons. First, less than a half of the compounds that belong to VBS bins four or greater could exist in the particle phase (Donahue et al., 2006), which would introduce large uncertainties for the estimation of the mass concentrations of the compounds in gas-phase from the particle phase. In addition, LC-ToF-MS analysis of pure compounds indicated that fragmentation of high molecular compounds during the ionization could occur, e.g., pinic acid ($C_9H_{14}O_4$) could be fragmented into $C_8H_{14}O_2$, and the latter was assigned to VBS bin 6 (Fig. S10). Note that due to the potentially high viscosity of the newly formed SOA, high-volatility compounds formed inside the aerosol particle could have been wrapped into the particle phase and detected (Shiraiwa et al., 2013b)." (Pages 11–12 lines 397–404).

*Line 332: How is obtained the total amount? By SMPS? It would be good to add a sentence to explain that.*

**Reply>** For each tentatively identified compound, the amount of the aerosol phase is the sum of the area of extracted ion chromatography. The gas phase was derived assuming gas-particle partitioning equilibrium. The total amount in *Line 332* (currently Page 12 line 412) is the sum of the aerosol phase and gaseous phase. The derivation of the quantity of the gas-phase was explained in the two sentences before *Line 332*: "The corresponding gas phase concentrations of each compound were derived assuming gas-particle partitioning equilibrium (Odum et al., 1996). The $\Delta H_{vap}$ values derived in this study (Table 1) were used to calculate the saturation concentration under 278 and 288 K following Eq. (3)." (currently Page 12 lines 409–411) .

*Line 350, Figure 4: There is no axes for the yield in the graph B; it would be good to add it.*

**Reply>** The curve does not represent the yields, but the $\alpha_i$ obtained from the four-product VBS fitting normalized based on the total mass fraction of compounds in VBS bins 0 to 3. This is explained in the current version of the text by changing the original expression "In Fig. 4b, the volatility distributions determined from SOA yield curves were also presented." (original manuscript *Page 10 lines 343–344*) to "In Fig. 4b, the volatility distributions determined by normalizing the stoichiometric SOA yields (Table 2)

based on the total mass fractions of compounds in VBS bins 0 to 3 are also presented." (Page 12 lines 428–429).

*Line 448-449: Replace « M/z 223 was the most abundant OS identified at 298 k, followed by m/z 279, 281, 269, 283, 265, 253, 267, 251, 249, and 247. » by « m/z 223 was the most abundant OS identified at 298 K, followed by m/z 279, 281, 269, 283, 265, 253, 267, 251, 249, and 247. »*

**Reply>** Modification has been made accordingly. In addition, the chemical formulas of each m/z are also presented (Page 16 lines 534–536).

*Line 500, Table 3: it's difficult to follow the table 3, it would be good to simplify to know what is really proposed by this study. Pleased find an example of table.*

| m/z

[M−H]− | Formula

[M−H]− | VBS bin | Tentative structure proposed in this study | Literature |
|---|---|---|---|---|
|  |  |  |  |  |

**Reply>** Modifications have been made to Table 3. (Page 19).

*SUPPORTING information*

*Table S1 Remove the C near O3 in table S1. I would add if it's possible a column with Δ α -pinene IN THE TABLE S1. It would be good to add the uncertainties in the table. For the yield, it would be good to correct it by all the wall losses. Replace also SOA by $m_{SOA}$*

**Reply>** In Table S1, the $^c$ near $O_3$ in has been removed; a column with Δα-Pinene (µg m$^{-3}$) has been added; uncertainties of the SOA mass and yields have been added; SOA has been replaced with $m_{SOA}$. However, the correction of gas-phase wall-loss hasn't been reflected in Table S1. It is only qualitatively discussed in Sect. 4.1.

*Table S2 I would add if it's possible a column with Δ α -pinene in the table S2. It would be good to add the uncertainties in the table. For the yield, it would be good to correct it by all the wall losses. Replace also SOA by $m_{SOA}$*

**Reply>** In Table S2, a column with Δα-Pinene (µg m$^{-3}$) has been added; uncertainties of the SOA mass and yields have been added; SOA has been replaced with $m_{SOA}$. However, the correction of gas-phase wall-loss hasn't been reflected in Table S2. It is only qualitatively discussed in Sect. 4.1.

*Figure S2: It would be good to model these experiment of wall loss and add the model in this figure*

**Reply>** The original Figure S2 is currently Figure S4. Model simulation of the particle wall-loss has been included in the figure. A discussion of those simulation results is now presented in Text S3.

**References:**

Carlton, A. G., Bhave, P. V., Napelenok, S. L., Edney, E. D., Sarwar, G., Pinder, R. W., Pouliot, G. A., and Houyoux, M.: Model Representation of Secondary Organic Aerosol in CMAQv4.7, Environmental Science & Technology, 44, 8553-8560, 10.1021/es100636q, 2010.

Donahue, N. M., Robinson, A. L., Stanier, C. O., and Pandis, S. N.: Coupled partitioning, dilution, and chemical aging of semivolatile organics, Environmental Science & Technology, 40, 2635-2643, 10.1021/es052297c, 2006.

Faust, J. A., Wong, J. P. S., Lee, A. K. Y., and Abbatt, J. P. D.: Role of Aerosol Liquid Water in Secondary Organic Aerosol Formation from Volatile Organic Compounds, Environmental Science & Technology, 51, 1405-1413, 10.1021/acs.est.6b04700, 2017.

Hinds, W. C.: Aerosol Technology: Properties, Behavior, and Measurement of Airborne Particles, Wiley-Interscience, United States, 483 pp., 1999.

Krechmer, J. E., Pagonis, D., Ziemann, P. J., and Jimenez, J. L.: Quantification of Gas-Wall Partitioning in Teflon Environmental Chambers Using Rapid Bursts of Low-Volatility Oxidized Species Generated in Situ, Environmental Science & Technology, 50, 5757-5765, 10.1021/acs.est.6b00606, 2016.

Kristensen, K., Cui, T., Zhang, H., Gold, A., Glasius, M., and Surratt, J. D.: Dimers in alpha-pinene secondary organic aerosol: effect of hydroxyl radical, ozone, relative humidity and aerosol acidity, Atmospheric Chemistry and Physics, 14, 4201-4218, 10.5194/acp-14-4201-2014, 2014.

Lai, A. C. K., and Nazaroff, W. W.: Modeling indoor particle deposition from turbulent flow onto smooth surfaces, Journal of Aerosol Science, 31, 463-476, 2000.

Nah, T., McVay, R. C., Pierce, J. R., Seinfeld, J. H., and Ng, N. L.: Constraining uncertainties in particle-wall deposition correction during SOA formation in chamber experiments, Atmospheric Chemistry and Physics, 17, 2297-2310, 10.5194/acp-17-2297-2017, 2017.

Odum, J. R., Hoffmann, T., Bowman, F., Collins, D., Flagan, R. C., and Seinfeld, J. H.: Gas/particle partitioning and secondary organic aerosol yields, Environmental Science & Technology, 30, 2580-2585, 10.1021/es950943+, 1996.

Pathak, R. K., Presto, A. A., Lane, T. E., Stanier, C. O., Donahue, N. M., and Pandis, S. N.: Ozonolysis of alpha-pinene: parameterization of secondary organic aerosol mass fraction, Atmospheric Chemistry and Physics, 7, 3811-3821, 10.5194/acp-7-3811-2007, 2007a.

Pathak, R. K., Stanier, C. O., Donahue, N. M., and Pandis, S. N.: Ozonolysis of alpha-pinene at atmospherically relevant concentrations: Temperature dependence of aerosol mass fractions (yields), Journal of Geophysical Research-Atmospheres, 112, 10.1029/2006jd007436, 2007b.

Shiraiwa, M., Zuend, A., Bertram, A. K., and Seinfeld, J. H.: Gas-particle partitioning of atmospheric aerosols: interplay of physical state, non-ideal mixing and morphology, Physical Chemistry Chemical Physics, 15, 11441-11453, 10.1039/c3cp51595h, 2013b.

Tang, I. N., and Munkelwitz, H. R.: Aerosol growth studies—III ammonium bisulfate aerosols in a moist atmosphere, Journal of Aerosol Science, 8, 321-330, 10.1016/0021-8502(77)90019-2, 1977.

von Hessberg, C., von Hessberg, P., Poschl, U., Bilde, M., Nielsen, O. J., and Moortgat, G. K.: Temperature and humidity dependence of secondary organic aerosol yield from the ozonolysis of beta-pinene, Atmospheric Chemistry and Physics, 9, 3583-3599, 10.5194/acp-9-3583-2009, 2009.

Wang, N. X., Jorga, S. D., Pierce, J. R., Donahue, N. M., and Pandis, S. N.: Particle wall-loss correction methods in smog chamber experiments, Atmospheric Measurement Techniques, 11, 6577-6588, 10.5194/amt-11-6577-2018, 2018a.

Zhang, X., Cappa, C. D., Jathar, S. H., McVay, R. C., Ensberg, J. J., Kleeman, M. J., and Seinfeld, J. H.: Influence of vapor wall loss in laboratory chambers on yields of secondary organic aerosol, Proceedings of the National Academy of Sciences of the United States of America, 111, 5802-5807, 10.1073/pnas.1404727111, 2014.

**References:**

*Bonn, B., G. Schuster, and G.K. Moortgat, Influence of water vapor on the process of new particle formation during monoterpene ozonolysis. Journal of Physical Chemistry A, 2002. **106**(12): p. 2869-2881.*

*Cocker, D.R., et al., The effect of water on gas-particle partitioning of secondary organic aerosol. Part I: alpha-pinene/ozone system. Atmospheric Environment, 2001. **35**(35): p. 6049-6072.*

*Glasius, M., et al., Carboxylic acids in secondary aerosols from oxidation of cyclic monoterpenes by ozone. Environmental Science & Technology, 2000. **34**(6): p. 1001-1010.*

*Hakola, H., et al., Product Formation from the Gas-Phase Reactions of Oh Radicals and O3 with a Series of Monoterpenes. Journal of Atmospheric Chemistry, 1994. **18**(1): p. 75-102.*

*Jonsson, A.M., M. Hallquist, and E. Ljungstrom, Impact of humidity on the ozone initiated oxidation of limonene, Delta(3)-carene, and alpha-pinene. Environmental Science & Technology, 2006. **40**(1): p. 188-194.*

*Jonsson, A.M., M. Hallquist, and E. Ljungstrom, The effect of temperature and water on secondary organic aerosol formation from ozonolysis of limonene, Delta(3)-carene and alpha-pinene. Atmospheric Chemistry and Physics, 2008. **8**(21): p. 6541-6549.*

*Jonsson, A.M., M. Hallquist, and H. Saathoff, Volatility of secondary organic aerosols from the ozone initiated oxidation of alpha-pinene and limonene. Journal of Aerosol Science, 2007. **38**(8): p. 843-852.*

*Kenseth Christopher M., Nicholas J. Hafeman, Yuanlong Huang, Nathan F. Dalleska, Brian M. Stoltz, and John H. Seinfeld Synthesis of Carboxylic Acid and Dimer Ester Surrogates to Constrain the Abundance and Distribution of Molecular Products in α-Pinene and β-Pinene Secondary Organic Aerosol Environmental Science & Technology **Article ASAP** DOI: 10.1021/acs.est.0c01566*

*Lai K., A. C.; Nazaroff, W. W. Modeling indoor particle deposition from turbulent flow onto smooth surfaces. J. Aerosol Sci. 2000, 31 (4), 463–476.*

*Shiraiwa, M.; Zuend, A.; Bertram, A. K.; Seinfeld, J. H., Gas-particle partitioning of atmospheric aerosols: interplay of physical state, non-ideal mixing and morphology. Physical Chemistry Chemical Physics **2013,** 15 (27), 11441-11453.*

*Winterhalter, R., et al., LC-MS analysis of aerosol particles from the oxidation of -pinene by ozone and OH radicals. Atmospheric Chemistry and Physics Discussions, 2003. **3**: p. 1-39.*

*Yu, J.Z., et al., Gas-phase ozone oxidation of monoterpenes: Gaseous and particulate products. Journal of Atmospheric Chemistry, 1999. **34**(2): p. 207-258.*

Yu, Y., et al., Nitrate ion photochemistry at interfaces: a new mechanism for oxidation of alpha-pinene. Physical Chemistry Chemical Physics, 2008. **10**(21): p. 3063-3071.

---

## Author Comment (AC2) · 25 Feb 2021

**Response to Anonymous Referee #2**

We appreciate the invaluable comments. Our answers to the comments are provided below. The reviewer comments are written in italics.

*General comments:*

*This manuscript describes a new atmospheric simulation chamber and its application to study the secondary organic aerosol from the oxidation of alpha-pinene by ozone at different temperatures and for seed aerosol particles of different acidity. Therefore, it fits well in the scope of the journal of atmospheric chemistry and physics and focusses on an interesting scientific topic which has been subject of many previous studies. The scientific results are presented in a clear and well-structured way and are based on a reasonable scientific approach and valid methods. However, this manuscript represents only a limited contribution to scientific progress in this field.*

*The description of the new simulation chamber is quite limited and rewards a more detailed discussion of its capabilities and limitations compared to other simulation chambers as well as how suitable it is for this kind of studies. Especially, the impact of the chamber limitations on the uncertainties for the major scientific results should be addressed and quantified.*

**Reply>** We have added more explanations on the features of the chamber. The chamber is aimed for exploratory research, especially, to investigate the influence of the acidity of pre-exiting particles on SOA formation at different temperatures. Using the present compact chamber, multiple experiments under different temperature, seed particle, relative humidity, oxidant, and radiation conditions can be executed in relatively short time. The major limitations of the present chamber are wall-loss of particles and oxidized vapors are high because of the high surface-to-volume ratio of the chamber. We have carefully revised the entire manuscript, while considering both the merits and limitations of the present chamber.

A brief explanation of the capabilities of the compact chamber after the discussion of its limitation in Sect. 4.1 has been included into the revised manuscript as follows: "We also note that the chamber is aimed for exploratory research, where multiple experiments under different temperature, seed particle, relative humidity, oxidant, and radiation conditions can be executed within relatively short periods." (Page 10 lines 338–341).

*The scientific results lack a reasonable representation of their uncertainties as well as a suitable comparison with data from the large number of previous studies on this subject. This is especially the case for the yields and VBS distributions. Some of the most interesting findings in this study are the analysis and identification of SOA compounds for which also potential formation mechanisms are discussed. This discussion may be restructured and focused on new findings.*

**Reply>** We have carefully estimated the uncertainties of the values reported in this study. Regarding the comparison with SOA yields of previous studies, we recognize that our values are largely underestimated when compared to those from the large chambers, mainly due to the wall-loss of oxidized vapors (Sect. 4.1). We focus on the influence of the acidity of seed particles on the SOA yields and chemical compositions, in addition to the influence of the chamber temperature (Sects. 4.2, 4.3). With regard to

the VBS distributions, we have referred to a recent work by Morino et al. (2020), where the most potential VBS distributions of the SOA from α-pinene ozonolysis among the reported ones were suggested based on model simulations of SOA concentrations from SOA formation experiments and VFRs (volume fraction remaining) from heating experiments. From the analysis of SOA compositions (Sec. 4.3), we attempted to find an indicator of the acidity of pre-existing particles in ambient aerosol, which we have highlighted in the revised manuscript.

*Most of the scientific results presented don't go much beyond current knowledge and some of them can be expected to have higher uncertainties than those of previous studies. Therefore, the manuscript contains only a limited contribution to the scientific understanding of formation and composition of secondary organic aerosol in our atmosphere. Hence, this manuscript should only be accepted for publication after major improvements and focusing on new scientific findings.*

**Reply>** From the systematic experiments regarding the temperature and acidity dependence of SOA yields and chemical compositions from α-pinene ozonolysis, we claim the following findings: (1) The SOA yields increased with an increase in the acidity of seed particles (solid/near solid state) in the 278–298 K temperature range at low SOA mass loadings. If the SOA mass loading is too high compared to the amounts of pre-existing particles, the enhancement of the SOA yield under acidic conditions would be limited. (2) Whereas the abundance of some chemical compounds such as organosulfates and oligomers increased with an increase of the acidity of seed particles, the acid-catalyzed decomposition of some chemical compounds was also observed , which suggests that little or no enhancement of SOA could occur under acidic conditions in field observations when the acid-catalyzed decomposition is dominant. (3) The organosulfates and the oligomers that increased with an increase in the acidity of the seed particles could be indicators of the acidity of pre-existing particles in the field. The findings are highlighted in the revised manuscript.

*Specific comments*

*Try to include all major results in the abstract.*

**Reply>** We have revised the abstract carefully as follows:

"Secondary organic aerosols (SOAs) affect human health and climate change prediction; however, the factors (e.g., temperature, acidity of pre-existing particles, and oxidants) influencing their formation are not sufficiently resolved. Using a compact chamber, the temperature and acidity dependence of SOA yields and chemical components in SOA from α-pinene ozonolysis were systematically investigated under 278 K, 288 K, and 298 K temperatures using neutral ($(NH_4)_2SO_4$)/acidic ($H_2SO_4+((NH_4)_2SO_4)$) seed aerosols. SOA components with *m/z* less than 400 were analyzed using negative electrospray ionization liquid-chromatography time-of-flight mass spectrometry. Based on the slightly negative temperature dependence of the SOA yields, the enthalpies of vaporization under neutral and acidic seed conditions were estimated to be 25 and 44 kJ mol$^{-1}$, respectively. In addition, SOA yields increased with an increase in the acidity of seed particles (solid/near solid state) at low SOA mass loadings, when compared with the seed particle amounts. Acidity dependence analysis of the chemical formula, molecular mass, and O:C ratio of the detected compounds indicated the enhanced formation of multiple oligomers in the wide

molecular mass range with a wide range of O:C ratios under acidic seed conditions. The abundance of some chemical compounds increased with an increase in the acidity of seed particles (e.g., m/z 197, 311, 313, 339, 355 and 383), while decreases in the abundance of some chemical compounds were observed (e.g., m/z 171, 185, 215, 343, and 357). The acidity dependence could be explained by acid-catalyzed heterogeneous reactions or acid-catalyzed decomposition of hydroperoxides. In addition, organosulfate (OS) formation was observed under acidic seed conditions. Six out of the eleven detected OS were potentially formed via the aldehyde + $HSO_4^-$ pathway."

*Page 1 line 10: Actually, you did not describe the development of a chamber in this study. Please reformulate.*

**Reply>** We have revised the abstract. We have added more explanations on the characteristics of the chamber.

*Page 1 line 25: SOA is not only formed by photo oxidation. Please reformulate.*

**Reply>** The word "photooxidation" has been changed to "oxidation" (page 1 line 26).

*Page 1 line 34: Please use "saturation concentrations".*

**Reply>** The phrase "saturate concentrations" has been changed to "saturation concentrations" (Page 1 line 34).

*Page 1 line 35: Notify the importance of ELVOCs and that ELVOCs and LVOCs are also formed in the gas phase (see e.g. Tröstl et al., Nature, 2016).*

**Reply>** The LVOCs here indicate all low volatility organic compounds, including ELVOCs, with saturation concentrations less than $10^{-0.5}$ µg m$^{-3}$. Regrettably, we noticed that the original description was not clear, and the reference was not appropriate. Hence, the original expression "The importance of the formation of low volatility organic compounds (LVOCs) with saturate concentrations of less than $10^{-0.5}$ µg m$^{-3}$ has been highlighted in the SOA formation mechanisms of recent studies (Ehn et al., 2014; Shrivastava et al., 2017)." has been modified to

"The importance of the formation of low (and/or extremely low) volatility organic compounds (LVOCs) with saturation concentrations less than $10^{-0.5}$ µg m$^{-3}$ through heterogeneous/multiphase accretion processes has been highlighted in SOA formation mechanisms in recent studies (Ziemann and Atkinson, 2012; Shrivastava et al., 2017)." (Page 1 lines 34–36).

Page 2 line 50-54: Reformulate to shorter sentences.

**Reply>** The sentence "However, contrary results have been reported for the influence of the acidity of pre-existing particles on SOA yields from chamber experiments. For example, whereas Offenberg et al. (2009) reported a positive relationship between the acidity of seed particles and the ratio of SOA concentrations from the photooxidation of α-pinene with NOx at an elevated acidity relative to neutral seed conditions, Eddingsaas et al. (2012) reported greater SOA yields under acidic than neutral seed conditions from photooxidation of α-pinene under high-NOx conditions, and no influence of seed particle acidity under low-NOx conditions." has been modified to:

"However, the influence of the acidity of pre-existing particles on SOA yields from chamber experiments is poorly understood. Previous studies have reported complex results. For example, Eddingsaas et al. (2012) reported greater SOA yields under acidic than under neutral seed conditions from photooxidation of α-pinene under high-NOx conditions, and no influence of seed particle acidity on SOA yields under low-NOx conditions." (Page 2 line 50–53).

*Page 2 line 54: Do you mean contradictory results?*

**Reply>** The phrase "contrary" has been changed to "inconsistent" (Page 2 line 54).

*Page 3 line 89-93: You may cite Brüggemann et al., EST, 54, 3767, 2020 here as well.*

**Reply>** The citation has been added (Page 3 line 97).

*Page 3 line 112: It should read: "at atmospheric pressure".*

**Reply>** The phrase "in atmospheric pressure" has been changed to "at atmospheric pressure" (Page 3 line 119).

*Page 3 line 114-115: Explain how the pure air is humidified (bubbling through water?) and how this affects the purity of the G3 pure air regarding gases (e.g. VOC) and small particles.*

**Reply>** The humidification method, which was explained in the original manuscript as: "The RH of the air was adjusted by passing the G3 pure air through MiliQ water (resistivity of 18.2 MΩ·cm) before entering the Teflon bag.", has now been updated to "The RH of the air was adjusted by passing the G3 pure air through MiliQ water (resistivity of 18.2 MΩ·cm, total organic carbon content ≤ 5 ppb) before it entered the Teflon bag.". (Page 4 line 121–123) No evident contamination from the humidification process of G3 air was observed according to results of blank chamber experiments (see Text S2).

*Page 3 line 131: Explain how ozone was added to the chamber and if you can estimate a mixing time for the reactants. How is mixing achieved in your chamber?*

**Reply>** Ozone produced by irradiation of pure $O_2$ with vacuum ultraviolet light from a low-pressure mercury lamp ozone generator (Model 600, Jelight Compony Inc., USA) was introduced into the chamber at a flow rate of 200 standard mL min$^{-1}$, for 1.5 min. After the ozone generator was turned off, the introduction of pure $O_2$ continued for another one minute to purge all generated ozone into the Teflon bag. The mixing in the chamber was achieved through the introduction of G3 pure dry air at a flow rate of 20 standard L min$^{-1}$ for 1 min after the introduction of ozone. These have been explained in the manuscript as follows: "After obtaining the initial concentrations of α-pinene and seed particles, excess ozone produced by irradiation of pure $O_2$ with vacuum ultraviolet light from a low-pressure mercury lamp ozone generator (Model 600, Jelight Compony Inc., USA) was introduced into the chamber at a flow rate of 200 standard mL min$^{-1}$ for 1.5 min, to initiate the ozonolysis reactions. After the ozone generator was turned off, the introduction of pure $O_2$ continued for another minute to purge all generated ozone into the Teflon bag. Subsequently, the G3 pure air was introduced for one minute to facilitate the mixing of the chamber air." (Page 4 line 147–151)

The mixing by introducing air at a flow rate of 20 SLM was probably completed within 55 s from the time profile of α-pinene after its introduction in the experiments where ozone was introduced into the chamber before α-pinene. This has been explained in the manuscript as follows: "In the experiments in which $O_3$ was first introduced, the introduction of G3 pure air was sustained for one more minute after the injection of α-pinene to purge all α-pinene into the Teflon bag and to facilitate the mixing of the chamber air. In the latter case, the maximum α-pinene concentrations appeared within 55 s of its introduction, which indicated that the mixing by introducing air with a flow rate of 20 SLM was probably completed within 55 s." (Page 4 line 153–157).

*Page 4 line 134: Here it should read: "aerosol particles". Generally, distinguish between aerosols = airborne particles and gases, aerosol particles = airborne particles, and gases throughout the manuscript.*

**Reply>** The phrase "aerosols" has been modified to "aerosol particles" or "particles" throughout the manuscript when necessary.

*Page 4 line 139-140: Explain the criteria for achieving a sufficient cleanliness of your chamber by flushing.*

**Reply>** At preliminary runs, we observed that the particle number and mass concentrations became sufficiently low after the clean process mentioned in the text. Indeed, the particle number and mass concentrations were reduced to less than 20 cm$^{-3}$ and 0.02 μg m$^{-3}$ (density = 1 g cm$^{-3}$), respectively. This, in addition to a detailed explanation of the clean process, have been included in the manuscript by modifying the original expression in *Page 4 lines 139-140* "The Teflon bag was flushed with G3 pure air at least three times between two successive experimental runs, which took approximately 40 min." to "Before each experimental run, the Teflon bag was cleaned by filling it with pure G3 air and then evacuating all the air from the bag at least three times, which took approximately 40 min. The very low chamber background particle concentrations indicate that the bag was sufficiently cleaned (Text S2)." (Page 5 lines 170–172).

*Page 5 line 161: It should read: "saturation concentrations".*

**Reply>** The phrase "saturated concentrations" has been changed to "saturation concentrations" (Page 5 line 198).

*Page 5 line 161: It should read: "molecular formulae" while chemical formulae would already contain some structural information. It would be very useful if you would add the names of those compounds identified in table S3. Explain how the C\* values have been calculated.*

**Reply>** The phrase "chemical formulae" has been changed to "molecular formulae" (Page 5 line 199). The compound names and molecular structures of the tentatively identified major products are now presented in Table S4. The calculation of *C\** was explained in Sect. 3.2 of the preprint manuscript, which corresponds to Sect. 3.3 in the revised manuscript (Page 7 lines 263–275).

*Page 5 line 169-170: Coagulation is not a wall loss. Please rephrase. Specify if you have any indications of electrostatic particle losses in your chamber especially for new Teflon foil.*

**Reply>** Although coagulation does not affect the total volume/mass of the aerosol particles, it influences the size distributions of the particles and the apparent wall-loss of sub-100 nm particles.

We think that electrostatic particle losses were low even for a new Teflon bag in our experiment, because the wall-loss of a new bag was often lower than that of an old bag. Besides, humid air was applied during the experiment, which may prevent electrostatic particle losses.

*Page 5 line 169-172: Please fit the particle losses in your chamber e.g. using the formulation given by Lai and Nazaroff, J. Aerosol Sci., 31, 436, 2000. Discuss how the wall loss parameters of your chamber compare to others and how suitable your chamber is for different kind of applications. Did you also measure the wall losses for solid particles?*

**Reply>** Simulations of wall-loss performed with seed particles have been carried out and presented in Text S3, and are referred to in Sect. 2.3 as: "Model simulation (Text S3) and literature survey results revealed that the high wall-loss rates of sub-100 nm particles were mainly caused by particle coagulation (Nah et al., 2017; Wang et al., 2018a) and those of super-200 nm particles were likely the result of turbulent deposition (Lai and Nazaroff, 2000)." (Page 6 lines 213–216).

Comparisons of the reported size-resolved wall-loss rates with the results of Hu et al. (2014) and Wang et al. (2018a) have been included in the revised manuscript as follows: "The size-distributions of the measured particle wall-loss rates presented shapes similar to that of a 0.83 $m^3$ Teflon chamber (Hu et al., 2014), whereas in the latter, the lowest wall-loss rates appeared in the smaller size end (~70–110 nm) and were greater (~0.2 $hr^{-1}$) than those in the present study. The large apparent wall-loss rates of sub-100 nm particles were also similar to those of a 1.5 $m^3$ Teflon reactor (Wang et al., 2018a).". (Page 6 lines 210–213)

The capacity of the current chamber has been described in Sect. 4.1 in the revised manuscript as follows: "We also note that the chamber is aimed for exploratory research, where multiple experiments under different temperature, seed particle, relative humidity, oxidant, and radiation conditions can be executed within relatively short periods." (Page 10 lines 338–340)

All seed particles used for the wall-loss rate measurements were in solid (neutral seed conditions) or near solid states (acidic seed conditions) in the studied 26–55 % RH range because they were dried into effloresced states before being introduced into the chamber (Tang and Munkelwitz, 1977).

*Page 5 line 173: How can the seed particle size affect wall losses like sedimentation for super 200 nm particles?*

**Reply>** Simulation results in Text S3 assuming pure sedimentation deposition indicates that it does not contribute much to the overall particle loss. Instead, model simulations indicate that turbulent deposition were likely the major reason for the deposition of super-200 nm particles. Therefore, the original expression "Large wall-loss was observed for particles with mobility diameters of less than 100 nm, caused mainly by coagulation (Wang et al., 2018a). Large wall-loss for particles with mobility diameters of larger than 200 nm was also observed. This is a shortcoming of this chamber and could be explained by the shorter sedimentation time in the compact space compared with large chambers." has been changed to: "Large wall-loss was observed for particles with mobility diameters less than 100 nm and larger than 200 nm. The size-distributions of the measured particle wall-loss rates presented shapes similar to that of a 0.83 $m^3$ Teflon chamber (Hu et al., 2014), whereas in the latter, the lowest wall-loss rates appeared in the smaller size end (~70–110 nm) and were greater (~0.2 $hr^{-1}$) than those in the present study. The large apparent wall-loss rates of sub-100 nm particles were also similar to those of a 1.5 $m^3$ Teflon reactor (Wang et al., 2018a). Model simulation (Text S3) and literature survey results revealed that the high wall-loss rates of sub-100 nm particles were mainly caused by particle coagulation (Nah et al., 2017; Wang et al., 2018a) and those of super-200 nm particles were likely the result of turbulent deposition (Lai and Nazaroff, 2000).". (Page 6 lines 209–216)

In addition, according to the measurement data (see new Fig. S4), the wall-loss rates of super 200 nm particles tend to be greater for seed particles with smaller mean diameters, leading to an increase in bulk wall-loss rates. We cannot clearly explain it, but we decided to wait for 30 min after the introduction of seed particles before initiating the reaction so that the size distributions of the seed particles could shift to the larger size end in our study. This was also explained in the text (currently Page 6 lines 216–220).

*Page 5 line 176: It should read: "larger particles".*

**Reply>** The word "large" has been changed to "larger" (page 6 line 220).

*Page 5 line 182: Please explain how you separated the SOA mass from the seed particle mass and what uncertainty this means for the determination of the SOA mass.*

**Reply>** The SOA mass was derived from its volume concentration, which was derived by the subtraction the volume of seed particles with the correction of the wall loss of particles (Text S4). This method would introduce larger uncertainty to the determined SOA mass if the increase in the volume concentration of particles as a result of SOA formation was low compared with the volume concentration of seed particles. As can be observed from Tables. S1 and S2, the relative uncertainty tends to be greater when the SOA mass concentration or yield is smaller. This point is now explained in the revised manuscript as follows: "A detailed explanation of the derivation of $m_{SOA}$ is presented in Text S4. Note that when the mass loadings of SOA are low, the obtained $m_{SOA}$ and related yields retain greater uncertainties because the subtracted volume concentrations of seed particles from the measured volume concentrations are large (Mei et al., 2013).". (page 6 line 235–238)

*Page 5 line 182-186: Please discuss the uncertainties caused by the different approximations applied in this section. Did you take direct losses of semi volatile gases to the wall into account (Zhang et al., PNAS, 2014)?*

**Reply>** The uncertainties of the wall-loss correction method can be estimated as the uncertainty (2σ) of the residual of the first-order bulk-volume wall-loss constant fitting. The derived relative uncertainties (2σ divided by the arithmetic mean of the measured bulk volume of seed particles) were in the 2.7–13 % range (Text S4). The loss of semi volatile gases was only qualitatively discussed in Sect. 4.1.

*Page 6 line 204: It should read "….was calculated and then ascribed…".*

**Reply>** The word "and" was added to the sentence (Page 7 line 264).

*Page 6 line 219: I think it would be helpful to add the evolution of measured particle number size distribution e.g. as a contour plot in Figure 1 as well as the evolution of the particle number.*

**Reply>** The evolutions of the measured particle number-size distributions and total particle number concentrations are now presented in Fig. S6.

*Page 6 line 224: Add the uncertainty of the SOA yield. It should read*

**Reply>** The uncertainties of the SOA mass and yield have been included in the manuscript, as well as the associated SI (Text S4, Tables S1 and S2, Fig. 3, and Page 8 line 287).

*Page 6 line 224-225: It should read: "As the corrected SOA particle concentration was constant after 50 min,…".*

**Reply>** The original expression "As the SOA concentration was kept constant after 50 min, …" has been modified to "As the corrected SOA particle concentration was constant after 50 min, …" (Page 8 line 288).

*Page 6 line 226: It should read: "…concentration of SOA particle mass at 90 minutes would be underestimated…".*

**Reply>** The original expression "… concentration of SOA at 90 min could be underestimated ..." has been modified to "… concentration of SOA particles at 90 minutes would be underestimated …" (Page 8 lines 289–290).

*Page 8 line 243: It should read: "…seven experiments with different initial α-pinene concentrations (54–323 ppbv) at 298 K and neutral seeds were conducted at a relative humidity (RH) of approximately 26–27 % (Table S1).".*

**Reply>** The original sentence "In this study, seven experiments with varied initial α-pinene concentrations (54–323 ppbv) at 298 K under neutral seed conditions were conducted under the RH conditions of approximately 26–27 % (Table S1)." has been modified to "In this study, seven experiments with different initial α-pinene concentrations (54–323 ppbv) at 298 K and neutral seeds were conducted under a RH condition of approximately 26–27 % (Table S1)." (Page 9 line 319–320).

*Page 8 line 244/Table S1: Indicate how much aerosol volume or mass was formed after adding ozone to your chamber but before adding pinene.*

**Reply>** SMPS measurements with only ozone and OH scavenger in the chamber filled with G3 pure air (Text S2) indicated that the aerosol mass before the addition of seed particles and α-pinene was 0.037 µg cm$^{-3}$, which was very low compared to the large mass of seed particles (33.6–120 µg cm$^{-3}$). (Unity density was assumed here.)

*Page 8 line 248: Add the error bars for your data in Figure 2 and enlarge the symbols. Discuss if the uncertainties are larger for the lower mass loadings. Note that you have only 5 minutes time resolution of the SMPS data. Please change the caption indicating that the lines represent a parameterization by Pathak et al. 2007a and not measured yields. Discuss how representative a comparison to only the data of Pathak et al. is considering the large amount of data available in the literature as well as more recent studies.*

**Reply>** The uncertainties of the final SOA mass concentration and yield have been included as error bars in Fig. 2. Note that although it is possible, the uncertainties of the real time SOA mass concentration and yield haven't been presented to maintain the clarity of the figure. For lower mass loadings, the relative uncertainties of both SOA loadings and yields tended to be larger because of the uncertainty that originated from subtraction of the volume concentration of seed particles was high (Mei et al., 2013). This point has been included in Sect. 3.1. The caption of Fig. 2 has been modified to clarify that the lines represent parameterization results by Pathak et al. (2007a). Only the results of Pathak et al. (2007a) were compared with the results of the present study because Pathak et al. (2007a) summarized the previous studies. Data recently reported by Saathoff et al. (2009), Wang et al. (2011), Wang et al. (2014), Nah et al. (2016), Ye et al. (2018), Kenseth et al. (2020), and Czoschke and Jang (2006) have been added in Fig. 2 in the revised manuscript, as follows.

[Figure]

**Figure 2: Yield comparison. SOA mass yields measured at 298 K under neutral seed conditions in the present study were compared to those of previous studies. Colored markers represent the results of this study. Colored circular markers represent the real-time SOA yields, i.e., the SOA yields along with the α-pinene ozonolysis reactions from 0 to 90 min. Different experimental runs are differentiated by colors. Red solid square markers represent the final SOA yields of the seven experiments. Horizontal error bars indicate the uncertainties of the final SOA concentrations; vertical error bars indicate the uncertainties of the final SOA yields. However, the systematic errors from vapor wall-loss are not included. The open square indicates the result of Exp. No. 2 after gas-phase wall-loss correction (Text S5). Black markers and curves represent results of previous studies. Black markers represent experimental results. The solid and dotted black curves represent the parameterized results from the four-product volatility basis-set fittings of previous α-pinene ozonolysis experiments under low NOx and dark conditions summarized in Pathak et al. (2007a). The solid curve represents results under a 50–73 % RH range and the dotted curve represents results under RH<10 %. The dashed curve represents the results calculated using Eqs. 1, 6, 7, 10, and 11 at 303 K in Saathoff et al., 2009. The experiments of Saathoff et al. (2009) were carried out at 303 K, 48–37 and 0.02 % RH, without or with OH scavenger (cyclohexane or 2-butanol); experiments of Wang et al. (2011) were carried out at 295 K, < 1 % RH, without OH scavenger; experiments of Wang et al. (2014) were carried out at 295 K, < 5 % RH, without OH scavenger; experiments of Nah et al. (2016) were carried out at 298 K, < 5 % RH, with cyclohexane as OH scavenger; experiments of Ye et al. (2018) were carried out at 296 K, 12–14 and 48–49 % RH, with cyclohexane as OH scavenger; experiments of Kenseth et al. (2020) were carried out at 295 K, <5 % RH, without OH scavenger; and experiments of Czoschke and Jang (2006) were carried out at 294–300 K, 14–67 %, without OH scavenger. Note that all data presented in this figure are normalized to unity density (1 g cm⁻³).**

(Page 10 lines 299–317)

*Page 8 line 250: Do you mean: "…used in this study was significantly larger than for most previous studies…"?*

**Reply>** Yes. The surface to volume ratio of the current compact 0.7 m³ chamber is 7.1 m⁻¹, while those of most previous studies were less than 3 m⁻¹. The whole sentence "Possible reasons may include no consideration of the wall-loss of oxidized organic vapors because the surface to volume ratio of the chamber used in this study could be much greater than those of previous studies (the volume of the chamber used in this study is 0.7 m³, whereas those of previous studies are 10–200 m³ (Pathak et al., 2007a and references therein))" has been modified to:

"Possible reasons may include lack of consideration of the wall-loss of oxidized organic vapors because the surface to volume ratio of the chamber used in this study (7.1 m$^{-1}$) was much larger than those of previous studies (<3 m$^{-1}$; Pathak et al., 2007a and references therein)." (Page 9 lines 325–327)

*Page 8 line 258-261: Given the methods you applied to correct for potential artifacts what uncertainty remains for SOA yields for different conditions and what are e.g. resulting concentration limit for your chamber.*

**Reply>** We did not make any correction when we discussed the temperature dependence of SOA yields, i.e., when we derived the $\Delta H_{vap}$ values. We have estimated the wall-loss of the organic vapors for the experimental runs of which the volatility distributions were obtained from the chemical analysis (Text S5). We found that the ratios of the corrected SOA masses to the uncorrected ones had no substantial temperature-dependence and we have added the following sentences in the revised manuscript: "Furthermore, the vapor wall-loss correction factors of the SOA mass presented no obvious temperature-dependence (Text S5). Therefore, the temperature dependence of SOA yields will be discussed assuming that the underestimation of the SOA yield due to the wall-loss of oxidized organic vapors does not affect the temperature dependence." (Page 10 lines 340–342).

*Page 8 line 266-267: Give the uncertainties for these parameters.*

**Reply>** We think it is difficult to estimate the uncertainties of the VBS parameters arithmetically. Instead, sensitivity analyses of the four-product VBS fitting curves achieved by changing $\Delta H_{vap}$ while fixing the stoichiometric yields $\alpha_i$ revealed that the effective $\Delta H_{vap}$ could be in the 0 to 70 and 0 to 80 kJ mol$^{-1}$ ranges for neutral and acidic seed conditions, respectively. This is now presented in the revised manuscript as follows: "Notably, sensitivity analyses achieved by fixing the stoichiometric yields $\alpha_i$ while changing $\Delta H_{vap}$ and comparing the resulting VBS curves with measured data (Fig. S8) indicated that the effective $\Delta H_{vap}$ could be in the 0 to 70 and 0 to 80 kJ mol$^{-1}$ ranges for neutral and acidic seed conditions, respectively." (Page 10 lines 366–368).

*Page 8 line  272: Do you mean: "…lowering of their vapor pressures."?*

**Reply>** The expression "…lowering their volatilities" means "…lowering of their saturation vapor pressures." (Page 10 lines 354).

*Page 8 line 277: It should read: "…lower than those of Saha et al….".*

**Reply>** The expression "…lower than that of Saha et al. (2016) and much lower than that of Epstein et al. (2010)." has been modified to "…lower than those of Saha et al. (2016) and much lower than those of Epstein et al. (2010)."(Page 10 line 359–360).

*Page 9 line 285: Indicate if the dependence of SOA yields on the acidity of seed particles is significant or not.*

**Reply>** The results of the present study indicated that when SOA concentrations were high, the acidity dependence could not be observed; however, it could be observed when SOA concentration was low. This is reasonable because the acidity effect decreases after the seed particles are covered with viscous SOA components (Shiraiwa et al., 2013b). The structure and part of the whole paragraph has been modified to as follows:

"Figure S9 presents the comparisons of the SOA yields under neutral and acidic seed conditions at different temperatures. It indicates that the SOA yields were enhanced under acidic seed conditions when the SOA loadings were low. When the SOA loadings were high, the enhancement disappeared. This is consistent with the results of Gao et al. (2004), which reported obvious initial $\alpha$-pinene concentration dependence of the enhancement of SOA yields under acidic conditions when compared with neutral seed conditions. For the initial $\alpha$-pinene concentrations of 12, 25, 48, 52, 96, 120, and 135 ppbv, the relative enhancements of SOA yields were 37, 34, 26, 24, 15, 10, and 8 %, respectively (Gao et al., 2004). This is probably because the SOA components can be of high viscosity under conditions where RH is smaller than around 50 %, and if high SOA mass loadings coated the seed particles, the acid-catalyzed heterogeneous SOA formation reactions could be impeded (Shiraiwa et al., 2013b; Zhou et al., 2013). In this study, the initial concentrations of $\alpha$-pinene were 54–323 ppbv at 298 K, suggesting that the enhancement could be less than 24 %. When the SOA volume loading was 50 $\mu m^3$ $cm^{-3}$, the fitted SOA yields under acidic conditions were enhanced by 11, 17, and 25 % when compared to the neutral seed conditions under 298, 288, and 278 K, respectively. This is consistent with the findings of Gao et al. (2004) and is also comparable to the results of Iinuma et al. (2005). In Iinuma et al. (2005), the experiment with 2-Butanol as an OH radical scavenger under room temperature (294–298 K) reported an enhancement of 19 %, with a final SOA volume concentration of approximately 50 $\mu m^3$ $cm^{-3}$. Furthermore, the degree of acidity of the seed aerosols could have also influenced the enhancement (Gao et al., 2004; Czoschke and Jang, 2006). Further comprehensive studies are warranted (including the consideration of the particle viscosity and phase separation) on the influence of seed aerosol acidity on $\alpha$-pinene ozonolysis SOA formation." (Pages 10–11 lines 369–385).

*Page 9 line 299: Indicate if the SOA yields were significantly enhanced comparing acidic to neutral seeds or not.*

**Reply>** Please refer to the responses to the previous comment (Pages 10–11 lines 369–385).

*Page 9 line 306-308: Which measurement accuracy would be needed to achieve significant results?*

**Reply>** We suggest that the acidity dependence is more significant at low SOA concentrations. This should be confirmed by a more complex and larger chamber, in which the influence of wall-loss is relatively low.

*Page 9 line 310: Add uncertainties to your data points in Figure 3 and compare to literature data.*

**Reply>** The uncertainty of SOA mass loadings and yields have been included in Fig. 3. Comparisons to literature data at 298 K under neutral seed conditions had been included in Fig. 2, and associated discussions had been integrated in the second paragraph of Sect. 4.1 in the manuscript. Both Fig. 2 and the associated discussions have been updated in the current manuscript.

*Page 10 line 316: Add the uncertainties to the VBS parameters in Table 1 and compare to literature values.*

**Reply>** We think it is difficult to estimate the uncertainties of the VBS parameters arithmetically. Instead, sensitivity analyses of the four-product VBS fitting curves were achieved by changing $\Delta H_{vap}$ while fixing the stoichiometric yields $\alpha_i$, revealing that the effective $\Delta H_{vap}$ could be in the 0 to 70 and 0 to 80 kJ mol$^{-1}$ ranges for neutral and acidic seed conditions, respectively (Page 10 lines 366–368). Conversely, the $\alpha_i$ values were compared with the observations in Fig. 4b and in the text (Page 12 lines 428–432).

Comparisons of $\Delta H_{vap}$ with literature values was presented in Table 2 and discussed in the second paragraph of Sect. 4.2 in the manuscript.

According to Morino et al. (2020), both the root-mean-square errors between the observed and simulated SOA concentrations in the formation experiments and between the observed and simulated volume fraction remainings in the heating experiments were minimized in the case of the $C^*$ distribution reported by Sato et al. (2018). As mentioned in the text, the $C^*$ distribution patterns in this study were confirmed to be similar to those reported by Sato et al. (2018). We have noted this in the text as follows: "According to Morino et al. (2020), both the root-mean-square errors between the observed and simulated SOA concentrations for the formation experiments and between the observed and simulated volume fraction remainings for the heating experiments were minimized in the case of the $C^*$ distribution reported by Sato et al. (2018)." (Page 12 lines 425–427).

*Page 10 line 325: It should read: "…measured intensities of particle phase compounds…".*

**Reply>** The expression "…measured intensities of aerosol phase compounds…" has been modified to "…measured intensities of particle phase compounds…" (Page 12 line 405).

*Page 10 line 331: It should read: "The intensities of both the particle and gas phases were…".*

**Reply>** The expression "The intensity of both the aerosol and gas phases were…" has been modified to "The intensities of both the particle and gas phases were…" (Page 12 line 411).

*Page 10 line 334: Give how much the rate coefficients varies with temperature. Please cite original references like Tillmann et al., PCCP, 11, 2323, 2009.*

**Reply>** The expression "As the α-pinene ozonolysis rate constant does not vary much under the temperature range of 278–298 K (Akimoto, 2016) and…" has been modified to "As the α-pinene ozonolysis rate constant at the temperature range of 278–298 K does not vary considerably (within 15 %; IUPAC Task

Group on Atmospheric Chemical Kinetic Data Evaluation, (http://iupac.pole-ether.fr, last access: 1 February 2021)),…" (Page 12 lines 414–416).

*Page 10 line 335-336: How valid is this assumption as the total amount of SVOCs may be influenced by temperature dependent wall losses, changing product branching ratios, etc..*

**Reply>** The small variation of the rate constant and the currently identified mechanisms (The Master Chemical Mechanism, http://mcm.leeds.ac.uk/MCMv3.3.1/home.htt) of α-pinene ozonolysis reactions indicate that the change of product branching ratios could be insignificant in the studied temperature range. In addition, the temperature-dependence of gas-phase wall-loss was also considered to be low (Text S5). The original expression "As the α-pinene ozonolysis rate constant does not vary much under the temperature range of 278–298 K (Akimoto, 2016) and α-pinene was completely consumed at the reaction time of 90 min, the total amount of formed SVOCs should be similar at the three temperatures." has been modified to: "As the α-pinene ozonolysis rate constant at the temperature range of 278–298 K does not vary considerably (within 15 %; IUPAC Task Group on Atmospheric Chemical Kinetic Data Evaluation, (http://iupac.pole-ether.fr, last access: 1 February 2021)), α-pinene was completely consumed at the reaction time of 90 min, and the temperature-dependence of gas-phase wall-loss was considered insignificant (Text S5), the total amounts of SVOCs formed should be similar at the three temperatures." (Page 12 lines 414–418) .

*Page 11 Figure 4: Replace Aerosol by Particle on the y-axis label and caption. Indicate the uncertainty of your data.*

**Reply>** The phrase "aerosol" in both the y-axis label and the caption have been modified to "particle".

It is difficult to estimate the uncertainty of the volatility distribution because the mass of compounds identified by LC-MS analysis could explain at most 30 % of the total SOA mass (Nozière et al., 2015). However, we note that the calculated distribution includes the uncertainties that result from compound specific sensitivities because the sensitivity of ESI mass spectrometry is compound specific in Sect. 3.3: "As has been noted previously, the sensitivity of ESI mass spectrometry is compound specific, thus the calculated distribution includes the uncertainties that result from compound specific sensitivities." (Page 7 lines 275–276). In addition, the volatility distribution pattern obtained in this study is like those of Sato et al. (2018), in which the distribution obtained with the same method as in this study has been compared with other methods.

*Page 12 line 380: It should read: "…were concentrated in the O:C ratio range…"*

**Reply>** The expression "…were concentratedly distributed in the O:C ratio range…" has been modified to "…were concentrated in the O:C ratio range…" (Page 14 line 465).

*Page 13: I think it would be useful to add a table of major products, their abundances and their tentative chemical identification (compound, structure) either separately or in table S3.*

**Reply>** The abundances (i.e., total area of extracted ion chromatogram) of all the 362 chemicals that were determined in this study have been included in Table S3. The compound names and molecular structures of the tentatively determined major products are now presented in Table S4.

*Page 13 line 429: Please explain the unspecific chromatograms in Figure S5.*

**Reply>** The chromatograms were extracted from the LC-TOF-MS data file for each OS assuming a relative m/z uncertainty of 20 ppm. The peaks that appeared at different retention times in the chromatogram should indicate isomers of the same OS formula. For OS with low signal intensities (i.e., m/z 247, 249, 251, 253, 267, 269, and 283), the variation of the baseline could also be observed from the figure. Note that the baseline wasn't subtracted from the EIC.

The Fig. S5 in the previous manuscript has now been numbered Fig. S11.

*Page 14 line 433: Can you give an estimate of the uncertainties for the OS intensities?*

**Reply>** The phrase "might be of a high uncertainty" has been changed to "might be underestimated" (Page 16 line 518).

*Page 14 Figure 7: Add uncertainties and discuss which differences or trends are significant.*

**Reply>** As has been explained previously, it is difficult to estimate the uncertainty of the volatility distributions. Both trends are significant. Panel (a) indicates the volatility distribution of SOA compounds that presented lower intensity under acidic seed conditions. Panel (b) indicates the volatility distribution of SOA compounds that presented higher intensity under acidic seed conditions. The mechanisms of such differences are interpreted in the manuscript.

*Page 14 line 448-449: For the mechanistic discussion it would be more useful to give here and in the following either the compound name or molecular formulae for each compound identified. State the uncertainties of the molecular yields. Consider adding potential reactions mechanisms e.g. in the supplement.*

**Reply>** Molecular formulae have been included in the text. (Page 16 lines 534–536)

It is difficult to estimate the uncertainties of ethyl-d5-sulfate equivalent (EDSeq.) yields of OS compounds because we have only one dataset and the sensitivity of the ESI mass spectrometry is compound specific. This has been explained in the revised manuscript as follows:

"**3.2 Derivation of the ethyl-d5-sulfate equivalent (EDSeq.) yield of OS**

Before the extraction of filter samples, 20 µL sodium ethyl-d5 sulfate methanol solution (50 µg mL$^{-1}$) was added to each sample filter as an internal standard for the quantification of OS. The ethyl-d5-sulfate equivalent (EDSeq.) masses of OSs were determined by comparing the total chromatographic peak areas

of OSs to that of the EDS standard with known mass. The EDSeq. masses of OSs were divided by the corresponding air volumes collected to obtain the EDSeq. concentrations of OSs. The EDSeq. molecular yield of OS is defined as the ratio between the estimated EDSeq. concentration of OS ($m_{OS}$, µg cm$^{-3}$) and the reacted mass concentration of α-pinene ($\Delta_{VOC}$, µg cm$^{-3}$). Note that the sensitivity of ESI mass spectrometry is compound specific; therefore, the calculated EDSeq. yield includes the uncertainties that result from compound specific sensitivities." (Page 7 lines 258–261).

The potential reaction mechanisms have been included in the supplementary file as Scheme S1, which is referred as "The former is referred hereafter as the alcohol pathway, and the latter the aldehyde pathway (aldehyde + HSO$_4^-$; Surratt et al., 2007) (Scheme S1)." (Page 17 lines 555–556).

*Page 14 line 450-451: Consider revising: "The possible formation mechanisms of the OS compounds were proposed based on literature data combined with the experimental results of this study."*

**Reply>** The original sentence "The possible formation mechanisms of the OS compounds were proposed based on previous literatures combined with the experimental settings of this study." has been revised to "The potential formation mechanisms of the OS compounds were proposed based on literature data in combination with the results of the experiments of this study." (Page 17 line 537–538).

*Page 15 line 461: It should read: "While all eleven OS compounds detected in this study are formed from α-pinene ozonolysis due to the presence of an excess of the OH radical scavenger,…"*

**Reply>** The original expression "While all eleven OS compounds detected in this study were very likely from α-pinene ozonolysis reaction because the presence of excess OH scavenger in this study,…" has been modified to "While all eleven OS compounds detected in this study are formed from α-pinene ozonolysis due to the presence of excess OH radical scavenger,…" (Page 17 line 548–549).

*Page 15 line 467: It should read: "…and one (m/z 265) was said to be from the sulfacation of pinonaldehyde…".*

**Reply>** The expression "…and one (m/z 265) was from the sulfacation of the aldehyde compound, specifically pinonaldehyde (Liggio and Li, 2006; Surratt et al., 2007)." has been changed to "…and one (m/z 265) was said to be from the sulfation of pinonaldehyde (Liggio and Li, 2006; Surratt et al., 2007)." (Page 17 line 554–555).

*Page 15 line 485: "…could only be observed at the lower temperatures."*

**Reply>** The original expression "…could only be observed at relatively low temperatures." has been changed to "…could only be observed at the lower temperatures." (Page 17 line 570).

*Page 15 line 488-489: Consider revising: "When the structure of the OS can be determined and the precursor compound can be assumed, the formation mechanism of the OS in α-pinene ozonolysis with acidic seeds may be confirmed."*

**Reply>** The original sentence "When the structure of the OS can be determined and the precursor compound can be expected, the formation mechanism of the OS in α-pinene ozonolysis with acidic seeds will be confirmed." has been modified to "When the structures of the OSs can be determined and the precursor compounds can be assumed, the formation mechanisms of OSs in α-pinene ozonolysis with acidic seeds may be confirmed." (Page 17 lines 577–578).

*Page 16 Figure 8: Indicate to what the yield is related to. You may add total OS yield vs. temperature which will be positive and dominated by C7H11O6S and C10H15O7S.*

**Reply>** The yield is related to the mass concentration of consumed α-pinene. It is the ethyl-d5-sulfate equivalent (EDSeq.) yield, the derivation of which has been included in the manuscript as **Sect. 3.2 Derivation of the ethyl-d5-sulfate equivalent (EDSeq.) yield of OS**. The original "Sect. 3.2 Volatility distribution analysis" has been moved to Sect. 3.3. The mean EDSeq. OS yields have been included in Fig. 8 and the interpretation of Fig. 8 has been modified to as follows:

"As shown in Fig. 8, the ethyl-d5-sulfate equivalent (EDSeq.) yields of m/z 247, 249, and 265 decreased with the increase of temperature. In fact, the OS at m/z 247 and 249 could only be observed at the lower temperatures. Conversely, the EDSeq. yields of other OS compounds increased with the temperature, or at least the yields at 278 K were the lowest among the three temperature conditions. The temperature dependence of OS EDSeq. yields seems not to be directly related to the formation mechanisms of either the alcohol pathway or the aldehyde pathway. Nevertheless, the mean EDSeq. yields of the eleven OS compounds, which were dominated by mz223 and mz279, increased with the increase of reaction temperature. As eleven OS (including three unreported) were observed in the α-pinene ozonolysis reactions with an OH scavenger and acidic seed particles, the identification of the structures of the OSs using high-resolution ion mobility mass spectrometry is planned as the next step. When the structures of the OSs can be determined and the precursor compounds can be assumed, the formation mechanisms of OSs in α-pinene ozonolysis with acidic seeds may be confirmed." (Page 17 lines 569–578).

*Page 17 Table 3: Try to enlarge the structures as much as possible as well as the letters indicating the various references.*

**Reply>** Modifications have been made accordingly.

*Page 18 line 508: Consider revising: "…with acidic and neutral seed aerosol particles.".*

**Reply>** The expression "…under acidic/neutral seed conditions." has been modified to "…with acidic and neutral seed aerosol particles." (Page 20 line 597).

**Reply>** The expression "Among the 362 identified compounds,…" has been modified to "Among the 362 compounds identified,…" (Page 20 line 605).

*Page 18: In section 5 you should go beyond a summary and add conclusions for our understanding of atmospheric SOA and OS. You may also discuss the relevance of your findings vs. existing literature data.*

**Reply>** We have revised the summary and conclusions, taking the responses to the general comments into consideration.

"Using the compact chamber system, SOA formation from α-pinene ozonolysis was studied with diethyl ether as an OH radical scavenger at temperatures of 278, 288, and 298 K, with acidic and neutral seed aerosol particles. The SOA yields and compounds with a molecular mass of less than 400 Da determined using a LC-TOF-MS were analyzed from the perspectives of temperature and seed particle acidity dependence.

The SOA yield increased slightly with the decrease of chamber temperature. The enthalpies of vaporization under neutral and acidic seed conditions was estimated to be 25 and 44 kJ mol$^{-1}$, respectively. The acidity dependence of the SOA yields at low SOA loadings were comparable to those reported by Gao et al. (2004) and Iinuma et al. (2005). However, the enhancement of the SOA yields under acidic conditions would be limited if the SOA mass loadings are much greater than the amounts of pre-existing particles.

Among the 362 compounds identified, the volatility of 331 was distributed in the VBS bins between −8 and 3. The temperature dependence of the volatility distribution of those identified compounds (particle phase + gas phase) could be consistently explained by the enthalpy of vaporization derived in this study.

The compounds whose intensities under acidic seed conditions were less than 0.9 times those of neutral seed conditions were dominated by monomers, whereas the compounds whose intensities under acidic seed conditions were more than 1.1 times those under neutral seed conditions were dominated by oligomers. The O:C ratios of the former were concentrated in the range of 0.4–0.75. The O:C ratios of the latter were broadly distributed. The compounds with O:C ratios less than 0.4 were all oligomers, which accounted for 61 % of the oligomers with high relative intensity under acidic conditions, whereas those with O:C ratios of greater than 0.75 were highly oxidized molecules, and only contributed to 1 % of the oligomers. In addition, the mean molecular mass of the former compounds (204 ± 4 g mol$^{-1}$) were evidently lower than those of the latter (284 ± 14 g mol$^{-1}$). The differences indicated that the formation of many oligomers, especially with low O:C ratios, was enhanced under acidic seed conditions. The acidity-dependence of certain major compounds could be explained by acid-catalyzed heterogenous reactions (e.g., m/z 171, 185, 343, and 357) or acid-catalyzed decomposition reactions (e.g., m/z 215 and 197), which suggests that little or no enhancement of SOA under acidic conditions in field observations could occur when acid-catalyzed decomposition is dominant.

For the first time, organosulfate compounds were studied for α-pinene ozonolysis reaction in the presence of an OH scavenger and acidic seed particles. Eleven OS compounds were determined from LC-TOF-MS analysis. All of them on average presented higher yields under acidic than under neutral seed conditions.

Six of the OS compounds were potentially formed via the aldehyde + $HSO_4^-$ pathway, which should be confirmed in future studies through high resolution mass spectrometry analyses.

Finally, the organosulfates and the oligomers that increased with an increase in acidity of the seed particles could be indicators of the acidity of pre-existing particles in the field, and the new findings obtained from this study should be confirmed using more complex and larger chambers."

*Tables S1 and S2: Why is there a letter c at the head of the column with ozone concentrations? Add uncertainties for your data. Give only significant digits.*

**Reply>** The letter c at the head of the column with ozone concentrations in Table S1 is not necessary and has been omitted. In Table S2, the letter c is used to indicate that α-pinene was introduced earlier than ozone. The uncertainties of both SOA mass loadings and yields have been added, and the numbers of significant digits have been unified in both Tables S1 and S2.

*Table S3: Please add compound names were possible and mention that C\* values were calculated according to Li et al., 2016.*

**Reply>** Tentative names and molecular structures for some major products are now presented in Table S4. References for the $C^*$ calculation have been added as a footnote in Table S3.

*Figure S3: Add a legend.*

Reply> A legend has been added to the figure. The previous Fig. S3 is now numbered as Fig. S5.

*Figure S4: Are the differences between the yield parameterizations significant?*

**Reply>** The yield parameterizations represent the measured mean SOA mass loadings and yields. Based on these parameterization results, the difference was significant when the SOA loadings were low (< approximately 100 μg m$^{-3}$) but not significant when the SOA loadings were high.

*Is it correct that the temperature dependence of the yields is more significant than the acidity dependence?*

**Reply>** It is difficult to tell which is significant based on the current data. Our aim was to provide information on the influence of particle acidity on SOA yield to SOA models such as CMAQ (Carlton et al., 2010; Pye et al., 2017). Although the temperature dependence of SOA formation in CMAQ is represented by Δ$H_{vap}$, the acidity dependence of monoterpene oxidations hasn't been considered. Our idea is to include the influence of particle acidity on the rate constants of the conversions from SVOC to NVOC in CMAQ model. This will be reported in a separate work.

*Figure S5: How do you explain the rather unspecific chromatograms e.g. for m/z 249, 253, 267, 283 ?*

**Reply>** The chromatograms were extracted from the LC-TOF-MS data file for each OS assuming a relative m/z uncertainty of 20 ppm. The peaks appeared at different retention times in the chromatogram should indicate isomers with the same OS formula. For OSs with low signal intensities (i.e., m/z 247, 249, 251, 253, 267, 269, and 283), the variation in the baseline could also be observed from the figure. Note that the baseline wasn't subtracted from the EIC.

---

## Author Response (AR2)

*Comments to the Author:*

*Dear authors. I tend to share the opinion of the initial reviewers that the amount of new information on ozonolysis of alpha-pinene presented in this study is not very high. However, given the that reviewer #1 agreed with the extensive revisions you have done, and requested no changes I am reasonably comfortable with approving this manuscript for publication in ACP.*

**Reply>** We appreciate the invaluable comments. Our answers to the specific comments are provided below.

*Figures: Some of the color choices are not optimal in my opinion, such as the pale colors in Figures 3, 4 and 7 (as well as S11). I understand the authors want to keep colors consistent but this comes at the expense of a poor contrast. Likewise, the pale symbols in Figure 6 are very hard to see, another color scheme could improve it. Finally, there is no need for using hatched bars in Fig 8, this color choice is distracting attention from the actual data shown in the figure.*

**Reply>** The color choices in Figs. 3, 4, 6, 7, 8, and S11 have been changed to improve contrast. The hatched bards in Fig. 8 have been changed to solid bars. The solid lines in Fig. 3a have been changed to dashed lines and the figure caption has been changed accordingly.

*I have minor corrections listed below.*

*"m/z" is typically italicized, not sure if they do it in ACP*

**Reply>** The expression "m/z" has been italicized through the main text and the SI.

*L12: neutral ((NH4)2SO4)/acidic (H2SO4+((NH4)2SO4)) -> neutral (NH4)2SO4) and acidic (H2SO4+((NH4)2SO4))*

**Reply>** The text has been modified accordingly. (Lines 11–12).

*L19,20: abundance -> peak abundances*

**Reply>** The text has been modified accordingly. (Lines 19, 20)

*L26: complex -> complex set*

**Reply>** The text has been modified accordingly. (Line 26).

*L68: Pre-existing -> Previous*

**Reply>** The text has been modified accordingly. (Line 69).

*L113: consider breaking this long paragraph into several shorter ones, it is hard to read*

**Reply>** The original long paragraph has been divided into five short paragraphs. (Lines 114–175)

*L142: m/zs -> m/z values*

**Reply>** The text has been modified accordingly. (Line 144).

*L142: sec -> s*

**Reply>** The text has been modified accordingly. (Line 144).

*L212: hr -> h*

**Reply>** The text has been modified accordingly. (Line 215).

*:482: be decomposed -> decompose*

**Reply>** The text has been modified accordingly. (Line 485).

*L574: mz223 and mz279 -> m/z 223 and m/z 279*

**Reply>** The text has been modified accordingly. (Line 577).

[revised manuscript text omitted]

**Text S1: Thermostat capacity of the chamber**

10    The thermostat capacity of the cabinet was evaluated by monitoring temperature variation under certain temperature settings under dark conditions. As presented in Figs. S1 and S2, we monitored temperatures at three positions inside the cabinet or Teflon chamber (i.e., T1, T2, and T3) for one hour when the temperature of the cabinet was set to 4.6, 14.6, or 24.7 °C. The temperature inside the cabinet within an hour varied within the 3.9–5.5, 14.0–15.3, and 24.6–25.3 °C ranges, respectively, and the standard deviations are 0.36, 0.27, and 0.18 °C, respectively. The temperature inside the Teflon chamber (T2) varied within

15    the 4.8–5.2, 14.8–15.1, and 24.9–25.2 °C ranges, respectively, with standard deviations of 0.15, 0.10, and 0.11 °C, respectively. The results indicate that the temperatures inside the chamber were well controlled (uncertainties were < 1 °C).

**Text S2: Chamber and filter blank analysis**

Blank chamber experiments have been performed using a Teflon bag that had been used for the α-pinene ozonolysis

20    experiments and had been cleaned. In the experiments, only dry or humidified pure G3 air was introduced into the chamber at 298 K. As a reference, the G3 air was also directly introduced into the PTR-MS instrument to check the tubing and instrument background. The obtained signal intensities of VOCs from PTR-MS measurements were converted into volume mixing ratios using the typical rate constant for ion-molecule reactions (i.e., $2 \times 10^{-9}$ cm$^3$ molecule$^{-1}$ s$^{-1}$) and reaction time (i.e., 100 μs) (see Eq. (1) in Inomata et al. (2010)). The sum of the signal intensities at *m/z* 43, 45, 47, 49, 57, and 60–160 were applied for the

25    calculation. The estimated concentrations of VOCs were 29, 29, and 27 ppbv in the dry chamber air, humid (16.2% RH) chamber air, and pure G3 air, respectively. The results indicate that the chamber background VOC concentrations were low (29 – 27 = 2 ppbv) and humidification of the G3 air had minimal influence on the background VOC concentrations in the chamber. The 2 ppbv value was 7 % of the lowest applied α-pinene concentration (27.0 ppbv, Tables S1 and S2). With regard to particles, when the chamber was filled with only dry or humid (16.2 % RH) G3 air, the particle number concentrations were

30    16 and 16 cm$^{-3}$, respectively (which are less than 0.2 % of the lowest applied seed particle concentration presented in Tables S1 and S2), and the particle mass concentrations were 0.012 and 0.016 μg m$^{-3}$, respectively (density = 1 g cm$^{-3}$). The results indicated that the humidification process of the G3 air did not add contamination to the experimental background, and the low mass concentrations indicated that the chamber background did not considerably influence the amount of SOA formed.

35    In addition, the number and mass concentrations of particles with only α-pinene (109 ppbv) and OH scavenger (51 ppmv) in humid G3 air or only O$_3$ (683 ppbv) and OH scavenger (51 ppmv) in humid G3 air were measured, and the respective results were approximately 11 and 120 cm$^{-3}$ (number concentration), and 0.004 and 0.037 μg m$^{-3}$ (mass concentration, density = 1 g cm$^{-3}$). The results also indicated that the influence of the chamber background on the experiment results was insignificant.

40    One blank filter was analyzed using LC-ToF-MS using a procedure similar to that of the sample filters. The total area of the 362 tentatively determined compounds in the blank filter was on average approximately 10 % of those in the sample filters (Fig. S3). The total areas of the extracted ion chromatogram for each tentatively determined compound in the six samples and the blank are also presented in Table S3.

45    **Text S3: Exploration of the particle-phase wall-loss mechanisms based on model simulations**

To explore the particle-phase wall-loss mechanisms, model simulations of different processes were executed and compared with the measured results (Fig. S4). In Fig. S4, the turbulent deposition simulation based on Lai and Nazaroff (2000) considered the influence of gravitational settling deposition but overlooked the deposition caused by Brownian diffusion. During the experiments, the sampling air velocities based on SMPS and PTR-MS were 300 and 250 ml min$^{-1}$, respectively. Assuming a

50    total sample flow velocity ($u_{sample}$) of $1.2 \times 10^{-5}$ m s$^{-1}$ (i.e., (550 ml min$^{-1}$) / (0.6 × 0.9 m$^2$)) using the deposition rate coefficient

formula for rectangular cavity in Lai and Nazaroff (2000), the deposition rate coefficients of friction velocities ($u^*$) of 5, 10, and 100 % of the $u_{sample}$ were simulated. It turns out that the turbulent deposition can only explain the wall-loss for super-200 nm particles. Simulation results also indicate that Brownian diffusion deposition (Hinds, 1999) contributed little to the particle phase wall-loss as the Brownian deposition rate after 1800 s of an assumed still state began, which corresponds to 30 min after the introduction of seed particles, decreased substantially (Fig. S4). In addition, simulation results indicate that the influence of gravitational settling deposition (Hinds, 1999) on the total particle wall-loss was low (Fig. S4). Excluding the processes that directly cause particle deposition on chamber walls, coagulation can affect the apparent loss rates of sub-100 nm particles (Nah et al., 2017; Wang et al., 2018a) by changing the size distributions of the particles. Therefore, the main reason for the observed wall-loss of sub-100 nm particles could be coagulation.

**Text S4: Derivation of the uncertainties of SOA mass concentrations and SOA yields**

The SOA mass concentrations were derived assuming a constant particle-phase bulk volume wall-loss rate with Eq. (S1). The wall-loss of gas-phase organic vapors was not considered here.

$$m_{\text{SOA,t}} = \left( V_t + k \sum_{i=0}^{t} V_i \Delta t - V_0 \right) \rho_{\text{SOA}}$$

(S1)

where, $m_{\text{SOA,t}}$ is the particle wall-loss corrected SOA mass concentration at reaction time $t$; $V_t$ and $V_i$ are the measured particle volume concentrations at reaction time $t$ and $i$, respectively; $V_0$ is the volume concentration of seed particles; $\Delta t$ is the time interval of each SMPS scan, which equals 5 min in the present study; and $\rho_{\text{SOA}}$ is the density of SOA, which is 1.34 g cm$^{-3}$ (Sato et al., 2018).

Accordingly, the uncertainty of $m_{\text{SOA,t}}$ is derived with Eq. (S2).

$$\delta m_{SOA,t} = \rho_{SOA} \sqrt{(\delta V_t)^2 + \left( \Delta t \sum_{i=0}^{t} V_i \right)^2 (\delta k)^2 + (k \Delta t)^2 \sum_{i=0}^{t} (\delta V_i)^2 + (\delta V_0)^2}$$

(S2)

where, $\delta V_i$ is the product of $V_i$ and the relative uncertainty of the first-order bulk-volume wall-loss constant. The latter equals to the uncertain ($2\sigma$) of the fitting residuals divided by the arithmetic mean of the measured volume concentrations of seed particles, and was in the 2.7–13 % range. The uncertainty of SOA yield at reaction time $t$ was further derived using Eq. (S3).

$$\left( \frac{\delta Y_t}{Y_t} \right)^2 = \left( \frac{\delta m_{SOA,t}}{m_{SOA,t}} \right)^2 + \left( \frac{\delta \Delta_{VOC,t}}{\Delta_{VOC,t}} \right)^2$$

(S3)

where, $Y_t$ and $\Delta_{\text{VOC,t}}$ are the SOA yield and reacted α-pinene, respectively, at reaction time $t$. The value of $\delta \Delta_{\text{VOC,t}}/\Delta_{\text{VOC,t}}$ was determined to be 0.2 through the measurement of α-pinene standard using the PTR-MS combined with a permeater (PD-1B, GASTECH).

**Text S5: Assessment of the influence of gas-phase wall-loss on SOA yields following Krechmer et al. (2016)**

The influence of the gas-wall partitioning on SOA yield was estimated for experimental runs whose volatility distribution are available (Sect. 3.3) according to the method suggested by Krechmer et al. (2016). For the VBS bins of 0, 1, 2, and 3, the equivalent organic mass concentration of the walls ($C_w$) were first derived; afterward, the fraction remaining in the gas phase ($F_g$) at gas-wall partitioning equilibrium was derived, which were then used to estimate the gas-phase loss of SOA mass combining the gas phase mass concentration derived from the particle phase concentration assuming gas-particle portioning equilibrium. With the gas-phase wall-loss correction, the SOA mass concentrations and yields could increase by 32, 25, 34,

38, 31, and 31 % for Exp. No. 2 (298 K, neutral seeds), 11 (288 K, neutral seeds), 16 (278 K, neutral seeds), 22 (298 K, acidic seeds), 31 (288 K, acidic seeds), and 36 (278 K, acidic seeds), respectively, when compared with the results when only particle-phase wall-loss was corrected. Notably, the gas-phase wall-loss was not derived for VBS bins smaller than 0 because little fraction is in the gas-phase for such low volatility compounds; it was also not derived for VBS bins greater than 3 because more than a half of the compounds in the bins are in gas-phase and the derivation of the gas phase concentration from the measured particle phase mass concentration could introduce large uncertainty. Therefore, we consider the estimated influence of gas-phase wall-loss on SOA yield by this method a low limit estimation.

$$R-OH \; + \; HO-\overset{\overset{O}{\|}}{\underset{\underset{O}{\|}}{S}}-OH \; \xrightarrow{-\,H_2O} \; R-O-\overset{\overset{O}{\|}}{\underset{\underset{O}{\|}}{S}}-OH$$

Alcohol

$$\underset{H}{\overset{R_1}{\underset{|}{C}}}=O \; + \; HO-\overset{\overset{O}{\|}}{\underset{\underset{O}{\|}}{S}}-OH \; \longrightarrow \; H-\overset{R_1}{\underset{\underset{OH}{|}}{C}}-O-\overset{\overset{O}{\|}}{\underset{\underset{O}{\|}}{S}}-OH$$

Aldehyde

**Scheme S1: Proposed pathways of organosulfate formation: (up) alcohol pathway, (bottom) aldehyde pathway.**

**Table S1: Summary of α-pinene ozonolysis experiments under neutral seed conditions.**

| Exp. No. | Temp (K) | RH (%) / Humid G3 time[b] | α-Pinene (ppbv) | Δα-Pinene (μg m$^{-3}$) | O$_3$ (ppbv) | Seed (cm$^{-3}$) | $m_{SOA}$±uncertainty (μg m$^{-3}$) | Yield±uncertainty |
|---|---|---|---|---|---|---|---|---|
| 1 | | | 150[c] | 795 | 640[c] | 2.78E+04 | 136 ± 10 | 0.17 ± 0.03 |
| 2[a] | | | 323[c] | 1705 | 698[c] | 2.93E+04 | 595 ± 28 | 0.35 ± 0.06 |
| 3 | | | 150[c] | 797 | 745[c] | 2.64E+04 | 145 ± 9 | 0.18 ± 0.03 |
| 4 | 298 | ~26–27 / 13 min | 252[c] | 1328 | 753[c] | 3.93E+04 | 417 ± 23 | 0.31 ± 0.05 |
| 5 | | | 205[d] | 1085 | >652[d] | 2.66E+04 | 307 ± 17 | 0.28 ± 0.05 |
| 6 | | | 54.4[d] | 278 | >592[d] | 4.46E+04 | 37.0 ± 9.0 | 0.13 ± 0.04 |
| 7 | | | 67.7[d] | 348 | >670[d] | 3.21E+04 | 24.5 ± 7.8 | 0.070 ± 0.026 |
| 8 | | | 110[c] | 603 | 748[c] | 2.09E+04 | 167 ± 15 | 0.28 ± 0.05 |
| 9 | | | 244[c] | 1344 | 749[c] | 1.96E+04 | 524 ± 34 | 0.39 ± 0.07 |
| 10 | | | 220[c] | 1216 | 742[c] | 2.58E+04 | 456 ± 31 | 0.38 ± 0.06 |
| 11[a] | 288 | ~32–34 / 10 min | 307[c] | 1688 | 770[c] | 2.46E+04 | 741 ± 49 | 0.44 ± 0.07 |
| 12 | | | 230[d] | 1276 | >572[d] | 2.31E+04 | 518 ± 31 | 0.41 ± 0.07 |
| 13 | | | 40.2[d] | 2155 | >682[d] | 2.42E+04 | 37.5 ± 10.0 | 0.17 ± 0.06 |
| 14 | | | 41.9[d] | 224 | >654[d] | 2.91E+04 | 19.1 ± 12.7 | 0.085 ± 0.058 |
| 15 | | | 157[c] | 900 | 723[c] | 2.29E+04 | 271 ± 21 | 0.30 ± 0.05 |
| 16[a] | | | 252[c] | 1434 | 710[c] | 2.42E+04 | 608 ± 37 | 0.42 ± 0.07 |
| 17 | | | 118[c] | 672 | 746[c] | 2.30E+04 | 236 ± 16 | 0.35 ± 0.06 |
| 18 | 278 | ~45–55 / 7 min | 165[c] | 941 | 721[c] | 2.27E+04 | 350 ± 21 | 0.37 ± 0.06 |
| 19 | | | 290[d] | 1658 | >593[d] | 1.74E+04 | 862 ± 65 | 0.52 ± 0.11 |
| 20 | | | 27.0[d] | 151 | >649[d] | 4.26E+04 | 14.0 ± 11.8 | 0.093 ± 0.080 |
| 21 | | | 40.0[d] | 221 | >646[d] | 2.88E+04 | 34.0 ± 10.7 | 0.15 ± 0.06 |

[a] Aerosols were sampled onto Teflon filters for LC-TOF-MS analysis.
[b] The time during of the introduction of humid G3 pure air.
[c] α-Pinene was introduced into the chamber after ozone. The presented concentration of α-pinene is the highest recorded concentration, and the presented ozone concentration is the concentration before the start of the ozonolysis reactions.
[d] α-Pinene was introduced into the chamber before ozone. The presented α-pinene concentration is the concentration before the start of the ozonolysis reactions and the ozone concentration is the highest concentration at the start of the reactions.

**Table S2: Summary of α-pinene ozonolysis experiments under acidic seed conditions.**

| Exp. No. | Temp (K) | RH (%) / Humid G3 time[b] | α-Pinene (ppbv)[c] | Δα-Pinene (μg m$^{-3}$) | O$_3$ (ppbv)[d] | Seed (cm$^{-3}$) | $m_{SOA}$±uncertainty (μg m$^{-3}$) | Yield±uncertainty |
|---|---|---|---|---|---|---|---|---|
| 22[a] | | | 236 | 1226 | >499 | 1.73E+04 | 450 ± 41 | 0.37 ± 0.08 |
| 23 | | | 310 | 1645 | >190[e] | 3.02E+04 | 487 ± 40 | 0.30 ± 0.07 |
| 24 | | | 197 | 1050 | >476 | 2.73E+04 | 212 ± 23 | 0.20 ± 0.05 |
| 25 | | | 145 | 753 | >722 | 2.32E+04 | 186 ± 36 | 0.25 ± 0.07 |
| 26 | 298 | ~26–27 / 13 min | 243 | 1262 | >740 | 2.85E+04 | 356 ± 59 | 0.28 ± 0.08 |
| 27 | | | 145 | 760 | >824 | 9.61E+03 | 200 ± 34 | 0.26 ± 0.07 |
| 28 | | | 314 | 1652 | >600 | 1.90E+04 | 441 ± 68 | 0.27 ± 0.07 |
| 29 | | | 55.3 | 289 | >658 | 1.82E+04 | 54.9 ± 15.4 | 0.19 ± 0.07 |
| 30 | | | 51.3 | 266 | >677 | 1.87E+04 | 24.2 ± 12.0 | 0.091 ± 0.049 |
| 31[a] | | | 299 | 1616 | >409 | 2.13E+04 | 661 ± 40 | 0.41 ± 0.09 |
| 32 | | | 126 | 678 | >653 | 1.45E+04 | 213 ± 17 | 0.31 ± 0.07 |
| 33 | 288 | ~32–34 / 10 min | 228 | 1233 | >622 | 1.69E+04 | 428 ± 32 | 0.35 ± 0.07 |
| 34 | | | 291 | 1570 | >596 | 1.71E+04 | 592 ± 43 | 0.38 ± 0.08 |
| 35 | | | 84.8 | 464 | >684 | 1.39E+04 | 80.7 ± 9.5 | 0.17 ± 0.04 |
| 36[a] | | | 312 | 1772 | >373 | 2.18E+04 | 737 ± 56 | 0.42 ± 0.09 |
| 37 | | | 252 | 1375 | >595 | 1.80E+04 | 767 ± 28 | 0.56 ± 0.11 |
| 38 | 278 | ~45–55 / 7 min | 143 | 797 | >645 | 1.65E+04 | 345 ± 13 | 0.43 ± 0.09 |
| 39 | | | 200 | 1112 | >612 | 1.65E+04 | 525 ± 18 | 0.47 ± 0.10 |
| 40 | | | 52.0 | 293 | >666 | 1.48E+04 | 69.5 ± 3.4 | 0.24 ± 0.05 |

[a] Aerosols were sampled onto Teflon filters for LC-TOF-MS analysis.
[b] The time during of the introduction of humid G3 pure air.
[c] Initial α-pinene concentration before the start of the reactions is presented.
[d] α-Pinene was introduced earlier than ozone. Thus, the values recorded here are the highest concentrations at the start of the α-pinene ozonolysis reactions, except Exp. No. 23.
[e] Only the ozone concentration at the end of the experiment was recorded.

**Table S3: SOA compounds identified through LC-ToF-MS analysis: the *m/z*, chemical formula, retention time in LC column, saturation concentration, and total areas of extracted ion chromatograms (including those of the blank filter). The data are presented in an Excel file.**

**Table S4: Tentatively determined molecular formula and structures of some major α-pinene ozonolysis products.**

| Compound (Ref.) | *m/z* [M−H]$^-$ | Formula [M−H]$^-$ | Proposed molecular structure |
|---|---|---|---|
| -- (This study) | 155.070 | C$_8$H$_{11}$O$_3$ |  |
| Terebic acid (Sato et al., 2016) | 157.050 | C$_7$H$_9$O$_4$ |  |
| Pinalic acid isomers (Sato et al., 2016) | 169.086 | C$_9$H$_{13}$O$_3$ |  |
| Terpenylic acid, Norpinic acid (Sato et al., 2016; Gao et al., 2004) | 171.065 | C$_8$H$_{11}$O$_4$ |  |

| Compound (Reference) | m/z | Formula | Structure |
|---|---|---|---|
| Pinonic acid (Sato et al., 2016) | 183.102 | $C_{10}H_{15}O_3$ | |
| Pinic acid (Sato et al., 2016) | 185.081 | $C_9H_{13}O_4$ | |
| Oxopinonic acid (Zhang et a., 2015) | 197.081 | $C_{10}H_{13}O_4$ | |
| 10-Hydroxypinonic acid (Sato et al., 2016) | 199.096 | $C_{10}H_{15}O_4$ | |
| 3-Methyl-1,2,3-butanetricarboxylic acid (Sato et al., 2016) | 203.055 | $C_8H_{11}O_6$ | |
| -- (Zhang et al., 2017) | 215.091 | $C_{10}H_{15}O_5$ | |
| -- (Zhang et al., 2015) | 313.164 | $C_{16}H_{25}O_6$ | |
| -- (This study) | 339.180 | $C_{18}H_{27}O_6$ | |
| Terpenyl-diaterpenyl ester (Zhang et al., 2015) | 343.139 | $C_{16}H_{23}O_8$ | |
| - (Zhang et al., 2015; Kristensen et al., 2017) | 355.175 | $C_{18}H_{27}O_7$ | |
| Pinyl-diaterpenyl ester (Zhang et al., 2015) | 357.154 | $C_{17}H_{25}O_8$ | |

[Figure]

 **Figure S1: A schematic of the compact chamber system. Arrows in light blue indicate the general air circulation inside the HCLP-1240. PTR-MS, proton transfer reaction mass spectrometry; DMA, differential mobility analyser; CPC, condensation particle counter; T. Controller, temperature controller; T1, temperature near the floor of the HCLP-1240; T2, temperature inside the Teflon bag; T3, temperature measured by the T. Controller.**

[Figure]

135    **Figure S2: Temperature variations at the top (T3) and the bottom (T1) of the cabinet, and inside the Teflon chamber (T2) based on initial temperature settings of 4.6, 14.6, and 24.7 °C, respectively.**

[Figure]

140    **Figure S3: Comparisons of the signal intensities (i.e., total area of extracted ion chromatogram) between sample filters (broad color bars) and the blank filter (thin black bars).**

[Figure]

**Figure S4: Size-resolved wall loss rates measured under a 17–23 % RH range at a temperature of 298 K (color markers) and model simulations of wall-loss caused by different mechanisms (black and grey curves). The legend of the measured data indicates the mean particle diameter during the wall loss rate measurement ($D_{p,mean}$), the initial total particle number ($N_0$) and volume ($V_0$) concentrations, and the bulk volume wall-loss constant ($k_{V,bulk}$). Model simulations of turbulent deposition based on Lai and Nazaroff (2000) considered the influence of gravitational settling deposition but neglected the influence of Brownian diffusion; simulations when the friction velocity ($u^*$) was 5, 10, and 100 % of the sample velocity ($u_{sample}$, $1.2 \times 10^{-5}$ m s$^{-1}$) are presented. Simulations of Brownian diffusion deposition and settling deposition assuming still state of Brownian diffusion deposition and settling deposition. For the former, the deposition rate coefficients at 100, 1800, 3600, and 5400 s after an assumed still state have begun are presented.**

[Figure]

**Figure S5: Number-, surface-, and volume-size distributions of aerosol particles before the start (i.e., seed particles, dashed curves) and at the end (i.e., 90 min after the start of the α-pinene ozonolysis reaction, solid curves) of a typical experimental run (Exp. No. 5).**

[Figure]

**Figure S6: The evolution of particle number-size distributions (image plot) and total particle number concentrations ($N_t$) during a typical experiment run (Exp. No. 27).**

[Figure]

**Figure S7: Comparison of $\Delta H_{vap}$ in the present study with previous studies in the $\Delta H_{vap}$ versus $C^*$ plot.**

[Figure]

165 **Figure S8: Sensitivity analyses of the four-product VBS fitting curves by changing $\Delta H_{vap}$ (unit: kJ mol$^{-1}$) while fixing the stoichiometric yields $\alpha_i$. In each panel, markers represent measured data, curves in dark colors represent the four-product VBS fitting results, and curves with light colors represent the four-product VBS fitting results if a different $\Delta H_{vap}$ other than the fitted one is applied.**

[Figure]

170

**Figure S9: SOA yields versus SOA mass loadings under neutral and acidic seed conditions at different temperatures based on four-product VBS model fittings.**

[Figure]

175 **Figure S10: ESI MS spectrum of pinic acid standard.**

[Figure]

[Figure]

180

**Figure S11: Extracted ion chromatograms of 11 organosulfates under different experimental conditions. The retention time range (RTR) used for the derivation of the signal intensity of each OS compound is also presented.**